**Competition alters predicted forest carbon cycle responses to nitrogen availability and**
**elevated CO$_2$: simulations using an explicitly competitive, game-theoretic vegetation**
**demographic model**
Ensheng Weng[1,2], Ray Dybzinski[3], Caroline E. Farrior[4], Stephen W. Pacala[5]
[1]Center for Climate Systems Research, Columbia University, New York, NY 10025
[2]NASA Goddard Institute for Space Studies, 2880 Broadway, New York, NY 10025
[3]Institute of Environmental Sustainability, Loyola University Chicago, Chicago, IL 60660
[4] Department of Integrative Biology, University of Texas at Austin, Austin, TX 78712
[5]Department of Ecology & Evolutionary Biology, Princeton University, Princeton, NJ 08544
**Corresponding author:** Ensheng Weng (wengensheng@gmail.com; phone: 212-678-5585)
**Key words:** Allocation; Biome Ecological strategy simulator (BiomeE); Competitively-optimal
strategy; Game theory; Nitrogen cycle
**Abstract:** Competition is a major driver of carbon allocation to different plant tissues (e.g.
wood, leaves, fine roots), and allocation, in turn, shapes vegetation structure. To improve their
modeling of the terrestrial carbon cycle, many Earth system models now incorporate vegetation
demographic models (VDMs) that explicitly simulate the processes of individual-based
competition for light and soil resources. Here, in order to understand how these competition
processes affect predictions of the terrestrial carbon cycle, we simulate forest responses to
elevated atmospheric $CO_2$ concentration [$CO_2$] along a nitrogen availability gradient using a
VDM that allows us to compare fixed allocation strategies versus competitively-optimal
allocation strategies. Our results show that competitive and fixed strategies predict opposite
fractional allocation to fine roots and wood, though they predict similar changes in total NPP
along the nitrogen gradient. The competitively-optimal allocation strategy predicts decreasing
fine root and increasing wood allocation with increasing nitrogen, whereas the fixed strategy
predicts the opposite. Although simulated plant biomass at equilibrium increases with nitrogen
due to increases in photosynthesis for both allocation strategies, the increase in biomass with
nitrogen is much steeper for competitively-optimal allocation due to its increased allocation to
wood. The qualitatively opposite fractional allocation to fine roots and wood of the two
strategies also impacts the effects of elevated [$CO_2$] on plant biomass. Whereas the fixed
allocation strategy predicts an increase in plant biomass under elevated [$CO_2$] that is
approximately independent of nitrogen availability, competition leads to higher plant biomass
response to elevated [$CO_2$] with increasing nitrogen availability. Our results indicate that the
VDMs that explicitly include the effects of competition for light and soil resources on allocation
may generate significantly different ecosystem-level predictions of carbon storage than those that
use fixed strategies.

## 1 Introduction

Allocation of assimilated carbon to different plant tissues is a fundamental aspect of plant growth and profoundly affects terrestrial ecosystem biogeochemical cycles (Cannell and Dewar, 1994; Lacointe, 2000). Ecologically, allocation represents an evolutionarily-honed "strategy" of plants that use limited resources and compete with other individuals and consequently drives successional dynamics and vegetation structure (De Kauwe et al., 2014; DeAngelis et al., 2012; Haverd et al., 2016; Tilman, 1988). Biogeochemically, allocation links plant physiological processes, such as photosynthesis and respiration, to biogeochemical cycles and carbon storage of ecosystems (Bloom et al., 2016; De Kauwe et al., 2014). Thus, correctly modeling allocation patterns is critical for correctly predicting terrestrial carbon cycles and Earth system dynamics.

In current Earth System Models (ESMs), the terrestrial carbon cycle is usually simulated by pool-based compartment models that simulate ecosystem biogeochemical cycles as lumped pools and fluxes of plant tissues and soil organic matter (Fig. 1: A) (Emanuel and Killough, 1984; Eriksson, 1971; Parton et al., 1987; Randerson et al., 1997; Sitch et al., 2003). In these models, the dynamics of carbon can be described by a linear system of equations (Koven et al., 2015; Luo et al., 2001; Luo and Weng, 2011; Sierra and Mueller, 2015; Xia et al., 2013):

$$\frac{dX}{dt} = AX + BU \qquad \text{(Eq. 1)}$$

where $X$ is a vector of ecosystem carbon pools, $U$ is carbon input (i.e., Gross Primary Production, GPP), $B$ is the vector of allocation parameters to autotrophic respiration and plant carbon pools (e.g., leaves, stems, and fine roots), and $A$ is a matrix of carbon transfer and turnover. In this system, carbon dynamics are defined by carbon input ($U$), allocation ($B$), and residence time and transfer coefficients ($A$). The allocation schemes ($B$) are thus embedded in a linear system, or

quasi-linear system if the allocation parameters in *B* are a function of carbon input (*U*) or plant
carbon pools (*X*).
The modeling of allocation in this system (i.e., the parameters in vector *B*) is usually based
on plant allometry, biomass partitioning, and resource limitation (De Kauwe et al., 2014;
Montané et al., 2017). The allocation parameters are either fixed ratios to leaves, stems, and
roots, which may vary among plant functional types (e.g., CENTURY, Parton et al., 1987; TEM,
Raich et al., 1991; CASA, Randerson et al., 1997) or are responsive to climate and soil
conditions as a way to phenomenologically mimic the shifts in allocation that are empirically
observed or hypothesized (e.g., CTEM, Arora and Boer, 2005; ORCHIDEE, Krinner et al., 2005;
LPJ, Sitch et al., 2003). These modeling approaches either assume that vegetation is equilibrated
(fixed ratios) or average the responses of plant types to changes in environmental conditions as a
collective behavior. Thus, the carbon dynamics in these models can be constrained by selecting
appropriate parameters of allocation, turnover rates, and transfer coefficients to fit the
observations (Friend et al., 2007; Hoffman et al., 2017; Keenan et al., 2013).

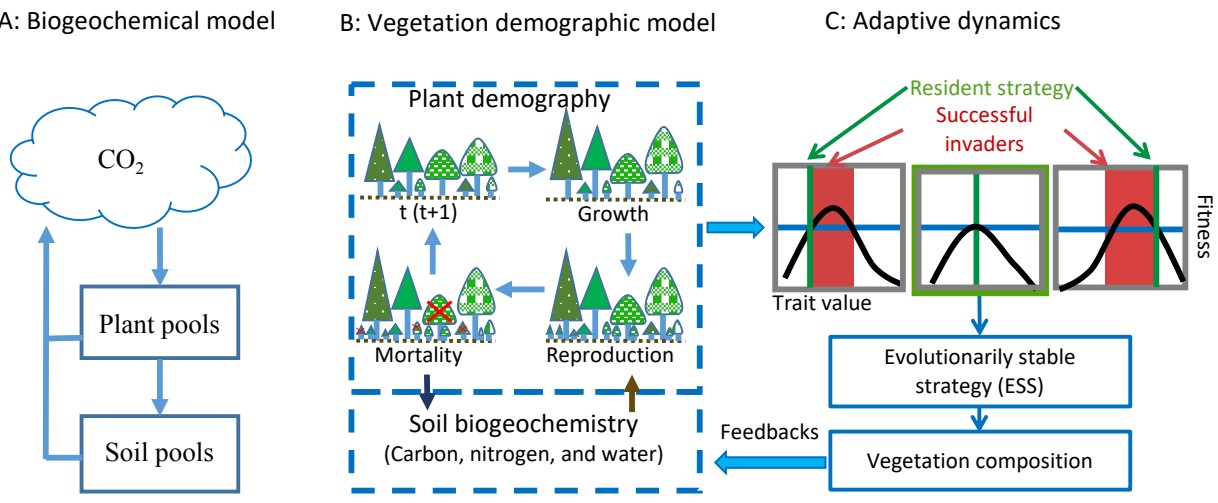


78                      **Figure 1 Hierarchical structure of vegetation models**


To predict transient changes in vegetation structure and composition in response to climate
change, vegetation demographic models (VDMs) that are able to simulate transient population
dynamics are being incorporated into ESMs (Fisher et al., 2018; Scheiter and Higgins, 2009).
Generally, VDMs explicitly simulate demographic processes, such as plant reproduction, growth,
and mortality, to generate the dynamics of populations (Fig. 1: B). To speed computations and
minimize complexity, groups of individuals are usually modeled as cohorts. With multiple
cohorts and PFTs, VDMs can bring plant functional diversity and adaptive dynamics into the
system when explicitly simulating individual-based competition for different resources and
vegetation succession and thus predict dominant plant traits changes with environmental
conditions and ecosystem development (Scheiter et al., 2013; Scheiter and Higgins, 2009; Weng
et al., 2015).
The combinations of plant traits represent the competition strategies at different stages of
ecosystem development. Evolutionarily, a strategy that can outcompete all other strategies in the
environment created by itself will be dominant. This strategy is called an evolutionarily stable
strategy or a competitively-optimal strategy (McGill and Brown, 2007). In VDMs,
competitively-optimal strategies can therefore be reasonably predicted based on the costs and
benefits of different strategies (i.e., combinations of plant traits) through their effects on
demographic processes (i.e., fitness) and ecosystem biogeochemical cycles (Fig. 1:C) (e.g.,
Farrior et al., 2015; Weng et al., 2015).
The dynamics of plant traits can substantially change predictions of ecosystem
biogeochemical dynamics since they change the key parameters of vegetation physiological
processes and soil organic matter decomposition (e.g., Dybzinski et al., 2015; Farrior et al.,
2015; Weng et al., 2017). Therefore, the key parameters that are used to estimate carbon
dynamics in the linear system model (Eq. 1), such as allocation ($B$) and residence times in
different carbon pools (matrix $A$, which includes coefficients of carbon transfer and turnover
time) become functions of competition strategies that vary with environment and carbon input. In
addition, the turnover of vegetation carbon pools becomes a function of allocation, leaf
longevity, fine root turnover, and tree mortality rates, which change with vegetation succession
and the most competitive plant traits. These changes make the system nonlinear and can lead to
large biases within the framework of the compartmental pool-based models as represented by Eq.
(1) (Sierra et al., 2017; Sierra and Mueller, 2015). Because of  the high complexity associated
with demographic and competition processes, the model predictions are usually sensitive to the
parameters in these processes and are of high uncertainty (e.g., Pappas et al., 2016).

In contrast to their implementation in the more complicated VDMs discussed above,

models of competitively-dominant plant strategies using much simpler model structures and
assumptions can sometimes be solved analytically (Dybzinski et al., 2011, 2015; Farrior et al.,
2013, 2015). Although simplified, such models can pin-point the key processes that improve the
predictive power of simulation models (Dybzinski et al., 2011; Farrior et al., 2013, 2015),
allowing them to help researchers formulate model processes and understand the simulated
ecosystem dynamics in ESMs. For example, the analytical model derived by Farrior et al. (2013)
that links interactions between ecosystem carbon storage, allocation, and water stress at elevated
atmospheric $CO_2$ concentration [$CO_2$]  sheds light on the otherwise inscrutable processes leading
to varied soil water dynamics in a land model coupled with an VDM (Weng et al., 2015).
Recognizing the benefit, Weng et al. (2017) included both a simplified analytical model and a
more complicated VDM to understand competitively optimal leaf mass per area, competition
between evergreen and deciduous plant functional types, and the resulting successional patterns.

In this study, we use a stand-alone simulator derived from the LM3-PPA model (Weng et

al., 2017, 2015) to show how forests respond to elevated $[CO_2]$ and nitrogen availability via
different competitively-optimal allocation strategies. The demographic processes of this model
have been coupled into the land model of the Geophysical Fluid Dynamical Laboratory's Earth
System Model (Shevliakova et al., 2009; Weng et al., 2015) and are being added to NASA
Goddard Institute for Space Study's Earth system model, ModelE (Schmidt et al., 2014). Using
this model, we simulate the shifts in competitively optimal allocation strategies in response to
elevated $[CO_2]$ at different nitrogen levels based on insights from the analytical model derived by
Dybzinski et al. (2015). Dybzinski et al.'s (2015) model predicts that increases in carbon storage
at elevated $[CO_2]$ relative to storage at ambient $[CO_2]$ are largely independent of total nitrogen
because of an increasing shift in carbon allocation from long-lived, low-nitrogen wood to short-
lived, high-nitrogen fine roots under elevated $[CO_2]$ with increasing nitrogen availability. Here,
we analyze the simulated ecosystem carbon cycle variables (gross and net primary production,
allocation, and biomass) of separate mono- and polyculture model runs. In the monoculture runs,
ecosystem properties are the result of the prescribed allocation strategies of a given PFT. In the
polyculture runs, competition between the different allocation strategies results in succession and
the eventual dominance of the most competitive allocation strategy for a given nitrogen
availability and $[CO_2]$ level. Since everything else in the model is identical, we are able to
compare the predictions of single **fixed strategies** with **competitively-optimal allocation**
**strategies** by comparing the ecosystem properties of these two types of runs.
**2 Methods and Materials**
**2.1 BiomeE model overview**

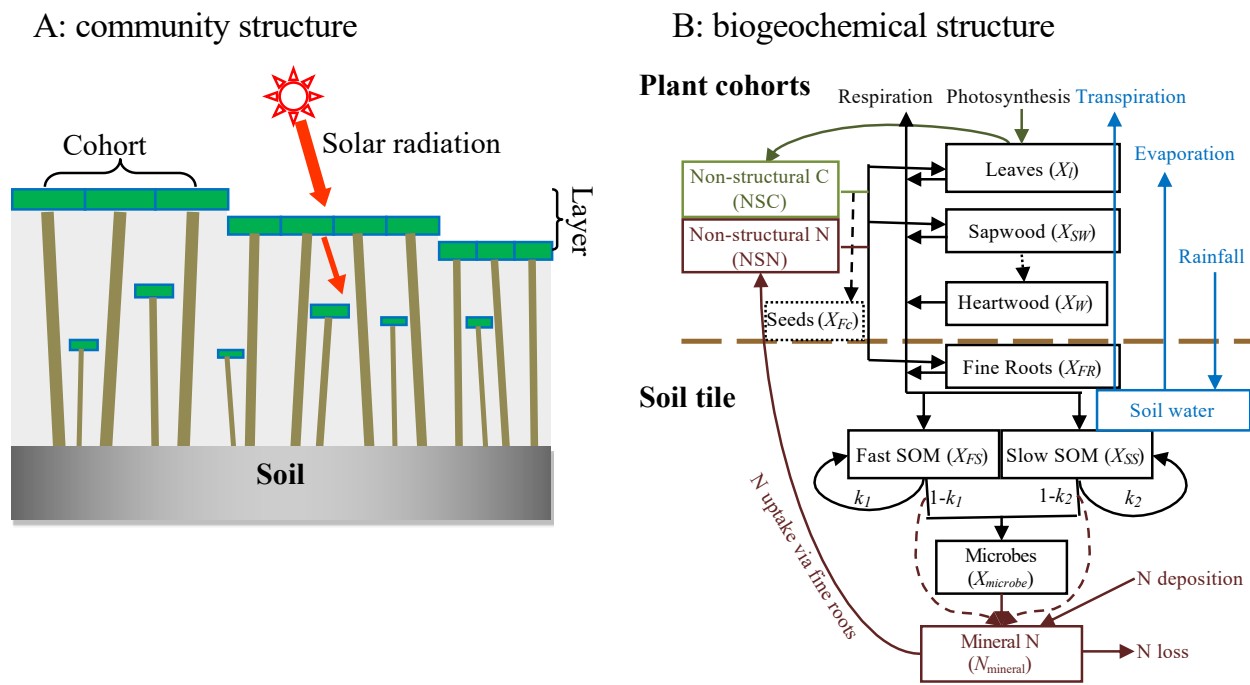


**Figure 2. Structure of BiomeE**
Panel A: vegetation structure: trees organize their crowns into canopy layers according to both
their height and their crown area following the rules of the PPA model, which mechanistically
models light competition. Panel B: Biogeochemical structure and compartmental pools. The
green, brown, and black lines are the flows of carbon, nitrogen, and coupled carbon and nitrogen,
respectively. The green box is for carbon only. The brown boxes are nitrogen pools. The black
boxes are for both carbon and nitrogen pools, where $X$ can be C (carbon) and N (nitrogen). The
C:N ratios of leaves, fine roots, seeds, and microbes are fixed. The C:N ratios of woody tissues,
fast soil organic matter (SOM), and slow SOM are flexible. Only one tree's C and N pools are
shown in this figure. The blue box and arrows are for water storage in soil and fluxes of rainfall,
evaporation, and transpiration. The model can have multiple cohorts of trees, which share the
same pool structure. The dashed line separates the aboveground and belowground processes.

We used a stand-alone ecosystem simulator (Biome Ecological strategy simulator,
BiomeE) to conduct simulation experiments. BiomeE is derived from the version of LM3-PPA
used in Weng *et al*. (2017), and its code is available at Github
(https://github.com/wengensheng/BiomeESS). In this version, we simplified the processes of
energy transfer and soil water dynamics of LM3-PPA (Weng et al., 2015) but still retained the
key features of plant physiology and individual-based competition for light, soil water, and, via
the decomposition of soil organic matter, nitrogen (Fig. 2 and Supplementary Information I for
details). In this model, individual trees are represented as sets of cohorts of similar size trees and
are arranged in different vertical canopy layers according to their height and crown area
following the rules of the Perfect Plasticity Approximation (PPA) model (Strigul et al., 2008).
Sunlight is partitioned into these canopy layers according to Beer's law. Thus, a key parameter
for light competition, critical height, is defined; all the trees above this context-dependent height
get full sunlight and all trees below this height are shaded by the upper layer trees.
Each tree consists of seven pools: leaves, fine roots, sapwood, heartwood, fecundity
(seeds), and non-structural carbohydrates and nitrogen (NSC and NSN, respectively) (Fig. 2: b).
The carbon and nitrogen in plant pools enter the soil pools with the mortality of individual trees
and the turnover of leaves and fine roots. There are three soil organic matter (SOM) pools for
carbon and nitrogen: fast-turnover, slow-turnover, and microbial pools, along with a mineral
nitrogen pool for mineralized nitrogen in soil. The simulation of SOM decomposition and
nitrogen mineralization is based on the models of Gerber *et al*. (2010) and Manzoni *et al*. (2010)
and described in detail in Weng et al. (2017). The decomposition rate of a SOM pool is
determined by the basal turnover rate together with soil temperature and moisture.  The nitrogen
mineralization rate is a function of decomposition rate and the C:N ratio of the SOM. Microbes
must consume more carbon in the high C:N ratio SOM pools to get enough nitrogen and must
release excessive nitrogen in the low C:N ratio SOM pools to get enough carbon for energy
(Weng *et al*. 2017).

**Table 1 Model parameters**

| Symbol | Definition | Unit | Default value | Reference |
|---|---|---|---|---|
| $\alpha_Z$ | Parameter of tree height | m m$^{-0.5}$ | 36 | Farrior et al., 2013 |
| $\theta_Z$ | Diameter exponent of tree height | - | 0.5 | Farrior et al., 2013 |
| $\Lambda$ | Taper factor | - | 0.75 | Weng et al. 2015 |
| $\rho_W$ | Wood density | Kg C m$^{-3}$ | 300 | Jenkins et al., 2003 |
| $\alpha_C$ | Parameter of crown area | m m$^{-1.5}$ | 150 | Farrior et al., 2013 |
| $\theta_C$ | Diameter exponent of crown area | - | 1.5 | Farrior et al., 2013 |
| $l^*$ | Target crown leaf area layers (crown leaf area index) | m$^2$ m$^{-2}$ | 3.5 | - |
| $\sigma$ | Leaf mass per unit area | kg C m$^{-2}$ | 0.14 | Wright et al., 2004 |
| $\gamma$ | Specific root area, calculated from root radius and density | m$^2$ kg C$^{-1}$ | 34.5 | Pregitzer et al., 2002 |
| $\varphi_{RL}$ | Ratio of target fine root area to target leaf area | m$^2$ m$^{-2}$ | Varied with PFTs | - |
| $\alpha_{CSA}$ | ratio of target sapwood cross-sectional area to target leaf area | m$^2$ m$^{-2}$ | 0.2x10$^{-4}$ | McDowell et al., 2002 |
| $f_{U,max}$ | Maximum mineral nitrogen absorption rate | hour$^{-1}$ | 0.5 | - |
| $K_{FR}$ | Root biomass at which the N-uptake rate is half of the maximum | kg C m$^{-2}$ | 0.3 | - |
| $CN_{L,0}$ | Target C:N ratio of leaves | kg C kgN$^{-1}$ | 76.5 (Function of LMA) | Wright et al., 2004 |
| $CN_{FR,0}$ | Target C:N ratio of fine roots | kg C kgN$^{-1}$ | 60 | Magill et al., 2004 |
| $CN_{W,0}$ | Target C:N ratio of wood | kg C kgN$^{-1}$ | 350 | Martin et al., 2015 |
| $CN_{F,0}$ | Target C:N ratio of seeds | kg C kgN$^{-1}$ | 20 | Soriano et al., 2011 |
| $f_1$ | Supply rate of NSC and NSN at normal growth | - | 1/(3*365) | - |
| $f_2$ | Maximum fraction of NSC and NSN used for growth in a day | - | 0.02 | - |
| $f_{LFR,max}$ | Maximum fraction of available carbon allocated to leaves and fine roots | - | 0.85 | - |
| $v$ | Fraction of carbon converted to seeds | - | 0.1 | - |

| | | | | |
|---|---|---|---|---|
| $r_{D/S}$ | Nitrogen-limiting factor | - | Solved by the model (Eqs 9 and 10) | - |


Plant growth and reproduction are driven by the carbon assimilation of leaves via
photosynthesis, which is in turn dependent on water and nitrogen uptake by fine roots. The
photosynthesis model is identical to that of LM3-PPA (Weng et al., 2015), which is a simplified
version of Leuning model (Leuning et al., 1995). This model first calculates photosynthesis rate,
stomatal conductance, and water demand of the leaves of each tree (cohort) in the absence of soil
water limitation. Then, it calculates available water supply as a function of fine root surface area
and soil water content. The demand-based assimilation rate and stomatal conductance are
adjusted if soil water supply is less than plant water demand. Soil water content is calculated
based on the fluxes of precipitation, soil surface evaporation, and plant water update
(transpiration) in three layers of soil to a depth of 2 meters. (Please see Supplementary
Information I for details).
Assimilated carbon enters into the NSC pool and is subsequently used for respiration,
growth, and reproduction. Empirical allometric equations relate woody biomass (including
coarse roots, bole, and branches), crown area, and stem diameter. The individual-level
dimensions of a tree, *i.e.*, height ($Z$), biomass ($S$), and crown area ($A_{CR}$) are given by empirical
allometries (Dybzinski et al., 2011; Farrior et al., 2013):

$$Z(D) = \alpha_Z D^{\theta_Z}$$
$$S(D) = 0.25\pi\Lambda\rho_W\alpha_Z D^{2+\theta_Z} \qquad \text{(Eq. 2)}$$
$$A_{CR}(D) = \alpha_c D^{\theta_c}$$

where $Z$ is tree height, $D$ is tree diameter, $S$ is total woody biomass carbon (including bole,
coarse roots, and branches) of a tree, $\alpha_c$ and $\alpha_Z$ are PFT-specific constants, $\theta_c=1.5$ and $\theta_Z=0.5$
(Farrior et al., 2013) (although they could be made PFT-specific if necessary), $\pi$ is the circular
constant, $\Lambda$ is a PFT-specific taper constant, and $\rho_W$ is PFT-specific wood density (kg C m$^{-3}$)
(Table 1).
We set *targets* for leaf ($L^*$), fine root ($FR^*$), and sapwood cross-sectional area ($A_{SW}^*$) that
govern plant allocation of non-structural carbon and nitrogen during growth. These *targets* are
related by the following equations based on the assumption of the pipe model (Shinozaki,
Kichiro et al., 1964):

$$
\begin{aligned}
L^*(D,p) &= l^* \cdot A_{CR}(D) \cdot \sigma \cdot p(t) \\
FR^*(D) &= \varphi_{RL} \cdot l^* \cdot \frac{A_{CR}(D)}{\gamma} \\
A_{SW}^*(D) &= \alpha_{CSA} \cdot l^* \cdot A_{CR}(D)
\end{aligned}
\qquad \text{(Eq. 3)}
$$

where $L^*$(D, p), $FR^*(D)$, and $A_{SW}^*(D)$ are the targets of leaf mass (kg C/tree), fine root biomass
(kg C/tree), and sapwood cross sectional area (m$^2$/tree), respectively, at tree diameter $D$; $l^*$ is the
target leaf area per unit crown area of a given PFT; $A_{CR}$(D) is the crown area of a tree with
diameter $D$; $\sigma$ is PFT-specific leaf mass per unit area (LMA); and $p(t)$ is a PFT-specific function
ranging from zero to one that governs leaf phenology (Weng et al., 2015); $\varphi_{RL}$ is the target ratio
of total root surface area to the total leaf area; $\gamma$ is specific root area; and $\alpha_{CSA}$ is an empirical
constant (the ratio of sapwood cross-sectional area to target leaf area). The phenology function
$p(t)$ takes values 0 (non-growing season) or 1 (growing season) following the phenology model
of LM3-PPA (Weng et al., 2015). The onset of a growing season is controlled by two variables,
growing degree days (GDD), and a weighted mean daily temperature ($T_{pheno}$), while the end of a
growing season is controlled by $T_{pheno}$. (Please see Supplementary Information I for details of the
phenology model)
**Nitrogen uptake**
The rate of nitrogen uptake ($U$, g N m$^{-2}$ hour$^{-1}$) from the soil mineral nitrogen pool is an
asymptotically increasing function of fine root biomass density ($C_{FR,total}$, kg C m$^{-2}$), following
McMurtrie *et al*. (2012)

$$U = f_{U,max} \cdot N_{mineral} \cdot \frac{C_{FR,total}}{C_{FR,total}+K_{FR}} , \qquad \text{(Eq. 4)}$$

where, $N_{mineral}$ is the mineral nitrogen in soil (g N m$^{-2}$), $f_{U,max}$ is the maximum rate of nitrogen
absorption per hour when $C_{FR,total}$ approaches infinity, $K_{FR}$ is a shape parameter (kg C m$^{-2}$) at
which the nitrogen uptake rate is half of the parameter $f_U$,max.  The nitrogen uptake rate of an
individual tree ($U_{tree}$, kg N hour$^{-1}$ tree$^{-1}$) is calculated as follows:

$$U_{tree} = U \cdot \frac{C_{FR,tree}}{C_{FR,total}} , \qquad \text{(Eq. 5)}$$

where, $C_{FR,tree}$ is the fine root biomass of a tree (kg C tree$^{-1}$). The nitrogen absorbed by roots
enters into the NSN pool and then is allocated to plant tissues through plant growth.

**Allocation and plant growth**

The partitioning of carbon and nitrogen into the plant pools (*i.e.*, leaves, fine roots, and

sapwood) is limited by the allometric equations, targets of leaves, fine roots, and sapwood cross-
sectional area, and the stoichiometry (i.e., C:N ratios) of these plant tissues. At a daily time step,
the model calculates the amount of carbon and nitrogen that are available for growth according
to the total NSC and NSN and current leaf and fine root biomass. Basically, the available NSC
($G_C$) is the summation of a small fraction ($f_1$) of the total NSC in an individual plant and the
differences between the targets of leaf and fine roots and their current biomass capped by a larger
fraction ($f_2$) of NSC (Eq. 6.1). The available NSN ($G_N$) is analogous to that of the NSC and
meets approximately the stoichiometrical requirement of plant tissues (Eq. 6.2).

$$G_C = \min\left(f_1 NSC + L^* + FR^* - L - FR, f_2 NSC\right) \qquad \text{(Eq. 6.1)}$$

$$G_N = \min\left(f_1 NSN + N_L^* + N_{FR}^* - N_L - N_{FR}, f_2 NSN\right) \qquad \text{(Eq. 6.2)}$$

where $L^*$ and $FR^*$ are the targets of leaves and fine roots, respectively (see Eq. 3); $L$ and $FR$ are
current leaf and fine roots biomass, respectively; $N_L^*$ and $N_{FR}^*$ are nitrogen of leaves and fine
roots at their targets according to their target C:N ratios. The parameter $f_1$ is the fraction of NSC
(or NSN) for normal growth after leaves and fine roots approach their targets and $f_2$ caps the
maximum daily availability of NSC (or NSN) during the period of leaf flush at the beginning of
a growing season. The parameter $f_1$ is much smaller than $f_2$. We let $f_1 = 1/(365 \times 3)$ and $f_2 = 0.02$ in
this study.

The allocation of the available NSC (i.e., $G_C$) to wood ($G_W$), leaves ($G_L$), fine roots ($G_{FR}$),

and seeds ($G_F$) follows the equations below (Eq. 7). These equations describe the mass growth of
plant tissues with nitrogen effects on the carbon allocation between high-nitrogen tissues and
low-nitrogen tissues (wood) for maximizing leaves and fine roots growth ($G_L$ and $G_{FR}$,
respectively), optimizing carbon usage at given nitrogen supply ($G_N$), and keeping the tissues at
their target C:N ratios.

$$G_C \geq G_W + G_L + G_{FR} + G_F \qquad \text{(Eq. 7.1)}$$

$$G_N \geq \frac{G_L}{CN_{L,0}} + \frac{G_{FR}}{CN_{FR,0}} + \frac{G_F}{CN_{F,0}} + \frac{G_W}{CN_{W,0}} \qquad \text{(Eq. 7.2)}$$

$$\frac{(FR + G_{FR})\gamma}{(L + G_L)/\sigma} = \varphi_{RL} \qquad \text{(Eq. 7.3)}$$

$$G_L + G_{FR} = Min \begin{pmatrix} L^* + FR^* - L - FR, \\ f_{LFR,max} \ G_C \end{pmatrix} \cdot r_{S/D} \qquad \text{(Eq. 7.4)}$$

$$G_F = \left[ G_C - Min \left( \frac{L^* + FR^* - L - FR,}{f_{LFR,max}\, G_C} \right) r_{S/D} \right] \cdot v \cdot r_{S/D} \qquad \text{(Eq. 7.5)}$$

$$G_W = \left[ G_C - Min \left( \frac{L^* + FR^* - L - FR,}{f_{LFR,max}\, G_C} \right) r_{S/D} \right] \cdot (1 - v \cdot r_{S/D}) \qquad \text{(Eq. 7.6)}$$

where, $CN_{L,0}$, $CN_{FR,0}$, $CN_{F,0}$, and $CN_{W,0}$ are the target C:N ratios of leaves, fine roots, seeds, and
sapwood, respectively; $\gamma$ is specific root area ($m^2$ kg $C^{-1}$); $\sigma$ is leaf mass per unit area (kg C $m^{-2}$);
$f_{LFR,max}$ is the maximum fraction of $G_C$ for leaves and fine roots (0.85 in this study); $v$ is the
fraction of left carbon for seeds (0.1 in this study); $r_{S/D}$ is a nitrogen-limiting factor ranging from
0 (no nitrogen for leaves, fine roots, and seeds) to 1 (nitrogen available for full growth of leaves,
fine roots, and seeds). The parameter $r_{S/D}$ controls the allocation of $G_C$ and $G_N$ to the four plant
pools (Eq. 7.1). It can be analytically solved (Eqs. 8 and 9).

$$r_{S/D} = Min\left[1, Max\left(0, \frac{G_N - G_C/CN_W}{N' - G_C/CN_W}\right)\right], \qquad \text{(Eq. 8)}$$

where, $N'$ is defined as the potential nitrogen demand for plant growth at $r_{S/D}=1$ (i.e., no nitrogen
limitation).

$$N' \equiv \frac{\gamma\sigma\left[FR + Min\left(\frac{L^* + FR^* - L - FR,}{f_{LFR,max}\, G_C}\right)\right] - \varphi_{RL}L}{(\gamma\sigma + \varphi_{RL})CN_L} + \frac{\varphi_{RL}\left[L + Min\left(\frac{L^* + FR^* - L - FR,}{f_{LFR,max}\, G_C}\right)\right] - \gamma\sigma L}{(\gamma\sigma + \varphi_{RL})CN_{FR}} +$$
$$\frac{v\left[G_C - Min\left(\frac{L^* + FR^* - L - FR,}{f_{LFR,max}\, G_C}\right)\right]}{CN_F} + \frac{(1-v)\left[G_C - Min\left(\frac{L^* + FR^* - L - FR,}{f_{LFR,max}\, G_C}\right)\right]}{CN_W}.$$

$$\text{(Eq. 9)}$$

When $G_N \geq N'$ ($r_{S/D} = 1$), there is no nitrogen limitation, and all the $G_C$ will be used for plant
growth and the allocation follows the rules of the carbon only model (Eqs 7.4~7.6 as $r_{S/D} = 1$).
The excessive nitrogen ($G_N - N'$) will be returned to the NSN pool (as if they were never taken
out). When $G_C/CN_{W,0} < G_N < N'$ (i.e., $0 < r_{S/D} < 1$), all $G_C$ and $G_N$ will be used in new tissue growth;
however, the leaves and fine roots cannot reach their targets at this step (i.e. they are down-
regulated). When $G_N \leq G_C/CN_{W,0}$ ($r_{S/D} = 0$), all the $G_N$ will be allocated to sapwood and the
excessive carbon ($G_C - G_N CN_{W,0}$) will be returned to NSC pool. This is a very rare case since a
low $G_N$ leads to low leaf growth, reducing $G_C$ before the case $G_N < G_C/CN_{W,0}$ happens. Therefore,
in most cases, Eq. 7.1 is: $G_C = G_W + G_L + G_{FR} + G_F$. Overall, this strategy down-regulates leaf
production under low nitrogen conditions while making use of assimilated carbon in height-
structured competition for light.

Allocation to wood tissues ($G_W$) drives the growth of tree diameter, height, and crown

area and thus increases the targets of leaves and fine roots (Eq. 3). By differentiating the stem
biomass allometry in Eq. 2 with respect to time, using the fact that $dS/dt$ equals the carbon
allocated for wood growth ($G_W$), we have the diameter growth:

$$\frac{dD}{dt} = \frac{G_W}{0.25\pi\Lambda\rho_w\alpha_z(2+\theta_z)D^{1+\theta_Z}} \tag{Eq. 10}$$

This equation transforms the mass growth to structural changes in tree architecture. With an
updated tree diameter, we can calculate the new tree height and crown area using allometry
equations (Eq. 2) and targets of leaf and fine root biomass (Eq. 3) for the next growth step.

Overall, this is a flexible allocation scheme and still follows the major assumptions in the

previous version of LM3-PPA (Weng, et al., 2015, 2017). This allocation scheme prioritizes the
allocation to leaves and fine roots, maintains a minimum growth rate of stems, and keeps the
constant area ratio of fine roots to leaves. Based on these allocation rules, the average allocation
of carbon and nitrogen to leaves, fine roots, and wood over a growing season are governed by the
targets for the leaf area per unit crown area (i.e., crown leaf area index, $l^*$) and fine root area per
unit leaf area ($\varphi_{RL}$). Since the crown leaf area index, $l^*$, is fixed in this study, $\varphi_{RL}$ is the key
parameter determining the relative allocation of carbon to fine roots and stems. A high $\varphi_{RL}$
means a high relative allocation to fine roots and therefore low relative allocation to stems, and
*vice versa*. Note, here $\varphi_{RL}$ is fixed for each PFT and will remain so for all the model runs.

The process of choosing a context-dependent competitively dominant $\varphi_{RL}$ will take place

after finding the fitness of each $\varphi_{RL}$ in monoculture and in competition with other PFTs (*i.e.*,
different values of $\varphi_{RL}$). The competitively optimal strategy is the one that can successfully
exclude all others in the processes of competition and succession, but it is not necessarily the one
that maximizes production in monoculture. For example, each $\varphi_{RL}$ creates an environment of
light profile and soil nitrogen in its monoculture. Other $\varphi_{RL}$ PFTs may have higher fitness in this
environment than the one that creates it. Only the competitively dominant strategy has the
highest fitness in the environment it creates (Fig. 1: C).
**2.2 Site and Data**
Data pertaining to vegetation, climate, and soil at Harvard Forest (Aber et al., 1993; Hibbs, 1983;
Urbanski et al., 2007) were used to design the plant functional types (PFTs) and ecosystem
nitrogen levels used in the simulation experiments, to drive the model, and to calibrate model
parameters.  Harvard Forest is located in Massachusetts, USA (42.54°, -72.17°). The climate of
Harvard Forest is cool temperate with annual precipitation 1050 mm, distributed fairly evenly
throughout the year. The annual mean temperature is 8.5 °C with a high monthly mean
temperature of 20°C in July and a low of -7°C in January. The soils are mainly sandy loam with
average depth around 1 m and are moderately well drained in most areas. In forest sites, soil
carbon is around 8 kg C m$^{-2}$ and nitrogen 300 g N m$^{-2}$ (Compton and Boone, 2000).  The
vegetation is deciduous broadleaf/mixed forest with major species red oak (*Quercus rubra*), red
maple (*Acer rubrum*), black birch (*Betula lenta*), white pine (*Pinus strobus*), and hemlock (*Tsuga*
*canadensis*) (Compton and Boone, 2000; Savage et al., 2013). The data used to drive our model
runs are gap-filled hourly meteorological data at Harvard Forest from 1991 to 2006, obtained
from North American Carbon Program (NACP) Site-Level Synthesis datasets (Barr et al., 2013).

**2.3 Simulation experiments**

We set two atmospheric $CO_2$ concentration ($[CO_2]$) levels: 380 ppm and 580 ppm, and

eight ecosystem total nitrogen levels (ranging from 114.5 g N m$^{-2}$ to 552 g N m$^{-2}$ at the interval
of 62.5 g N m$^{-2}$) by assigning the initial content of the slow SOM pool for our simulation
experiments (Table 2). This range covers the soil nitrogen contents across the plots at Harvard
Forest with different species compositions and land use history (200~300 g N m$^{-2}$) (Compton and
Boone, 2000; Melillo et al., 2011), and represents the range from infertile to fertile soils in
temperate forests (Post et al., 1985; Yang et al., 2011). The nitrogen cycles through the plant and
soil pools and is redistributed among them via plant demographic processes, soil carbon
transfers, and plant uptake. In all the simulation experiments, we assume the ecosystem has no
nitrogen inputs and no outputs for convenience since we already have eight total nitrogen levels
to represent the consequences of different nitrogen input and output processes at an equilibrium
state.  The PFTs were based on an evergreen needle-leaved tree PFT with different leaf to fine
root area ratios, $\varphi_{RL}$, in the range from 1 to 8 (Table 2). Simply stated, the PFTs we investigate
only differ in parameter $\varphi_{RL}$.

We define the model runs started with only one fixed-$\varphi_{RL}$ PFT as "monoculture runs"

although the actual allocation of carbon to different plant tissues varies with $[CO_2]$ and
ecosystem nitrogen availability. The model runs started with multiple PFTs are called
"polyculture runs" (eight PFTs with different $\varphi_{RL}$ at the beginning, although many are driven to
extinction during a given model run). We conducted one set of monoculture runs and two sets of
polyculture runs (Table 2).

**Table 2 Simulation experiments**

| Type | Model runs | Initial PFT(s) $\varphi_{RL}$ | Ecosystem total nitrogen levels | $CO_2$ concentration $[CO_2]$ |
|---|---|---|---|---|
| Monoculture runs | One model run per combination of PFT ($\varphi_{RL}$), nitrogen level, and $CO_2$ concentration | One of the following PFTs: $\varphi_{RL}$= 1, 2, 3, 4, 5, 6, 7, or 8 | Eight levels ranging from 114.5 g N m$^{-2}$ to 552 g N m$^{-2}$ at the interval of 62.5 g N m$^{-2}$: (i.e., 114.5, 177, 239.5, 302, 364.5, 427, 489.5, and 552 g N m$^{-2}$) | Ambient: 380 ppm |
| Polyculture runs I | One model run per combination of nitrogen level and $CO_2$ concentration | All the PFTs ($\varphi_{RL}$= 1~ 8) used in the monoculture runs | | |
| Polyculture runs II | One model run per combination of nitrogen level and $CO_2$ concentration | Eight PFTs with $\varphi_{RL}$ ranging from 4.5-0.5$i$ to 8.5-0.5$i$ at the interval of 0.5, where $i$ denotes the eight nitrogen levels from 114.5 to 552 gN m$^{-2}$. | | Elevated: 580 ppm |


In the monoculture runs, we run the full combinations of eight PFTs with root/leaf area
ratios ($\varphi_{RL}$) from 1 to 8, eight ecosystem total nitrogen levels, and two $CO_2$ concentrations (380
ppm and 580 ppm) (Table 2). For the eight PFTs, only those with $\varphi_{RL}$ <=6 survived at ambient
[$CO_2$] (380 ppm) because the carbon assimilated by leaves could not meet the demand by plant
tissues at $\varphi_{RL}$>6. The monoculture runs are for exploring the model predictions of gross primary
production (GPP), net primary production (NPP), allocation, and biomass at equilibrium with
fixed $\varphi_{RL}$ at different total nitrogen levels.
In polyculture runs I, we used the same PFTs as in those monoculture runs, where their $\varphi_{RL}$
varied from 1 to 8 at the interval of 1.0 and the ecosystem total nitrogen levels were the same as
those used in the monoculture runs (Table 2). This set of polyculture runs was used to explore
successional patterns at both ambient and elevated [$CO_2$] (380 ppm and 580 ppm, respectively).
However, this set of model runs could not show the details of equilibrium plant biomass and
allocation patterns along the nitrogen gradient because of the large intervals between the $\varphi_{RL}$
values.
To achieve greater resolution in our competition predictions, we designed the polyculture
runs II using a dynamic PFT combination scheme according to the ranges of $\varphi_{RL}$ obtained from
the polyculture runs I that could survive at a particular nitrogen level at both $CO_2$ concentrations.
For each nitrogen level, we set eight PFTs with $\varphi_{RL}$ that varied in a range 3.5 (e.g., $x \sim x+3.5$) at
the interval of 0.5, starting with the highest $\varphi_{RL}$ of 8.0 at the lowest N level (114.5 g N m$^{-2}$) and
decreasing 0.5 per level of increase in ecosystem total N. We used $i$=1, 2, …, 8 to denote the
eight N levels from 114.5 to 552 g N m$^{-2}$. The $\varphi_{RL}$ of the eight PFTs at each level were 5.0-0.5$i$,
5.5-0.5$i$, …, 8.5-0.5$i$ (Table 2). For example, at the nitrogen of 114.5 g N m$^{-2}$ ($i = 1$), the $\varphi_{RL}$ of
the eight PFTs were 4.5, 5.0, …, 8.0 and at 177 g N m$^{-2}$ ($i = 2$), they were 4.0, 4.5, …, 7.5.
For both monoculture and polyculture runs, visual inspection indicated that stands had
reached equilibrium after ~1200 years. To be conservative, we present equilibrium data by
averaging model properties between years 1400 and 1800. We compared simulated equilibrium
GPP, NPP, allocation (both absolute amount of carbon and fractions of the total NPP), and plant
biomass of the polyculture runs II with those from the monoculture runs. We used the results
from one PFT ($\varphi_{RL}$=4) to highlight the differences of plant responses with competitively optimal
allocation strategies obtained from the polyculture runs II.

**3 Results**

In the monoculture runs, GPP and NPP increase by a factor of three along the gradient of
nitrogen used in this study (114.5 - 552 g N m$^{-2}$) at both ambient (Fig. 3) and elevated [$CO_2$]
(Figs. S1). The magnitude of differences in GPP and NPP due to differences in fixed allocation
within a given nitrogen level is comparable to the magnitude of differences in GPP and NPP due
to nitrogen level within a given fixed allocation strategy (Fig. 3: a and b) when $\varphi_{RL}$ is in the
range that allows plants to grow normally (1~5 in the case of ambient [$CO_2$]). As prescribed by
the definition of $\varphi_{RL}$, allocation of NPP to fine roots increases with $\varphi_{RL}$ in monoculture runs (Fig.
3: c). As a consequence, allocation of NPP to wood decreases as $\varphi_{RL}$ increases (Fig. 3: d).
Allocation to leaves does not change much with $\varphi_{RL}$. (Fig. 3: e, note differences in scale).
Correspondingly, plant biomass at equilibrium decreases with $\varphi_{RL}$ (Fig. 3: f).  The effects of
nitrogen on the allocation of carbon to fine roots and wood follow our allocation model
assumptions because more carbon is allocated to low-nitrogen woody tissues in our model when
nitrogen is limited. However, the amplitude of changes in GPP and NPP induced by nitrogen
availability is lower than the amplitude of changes resulting from different values of $\varphi_{RL}$ in the
monoculture runs.

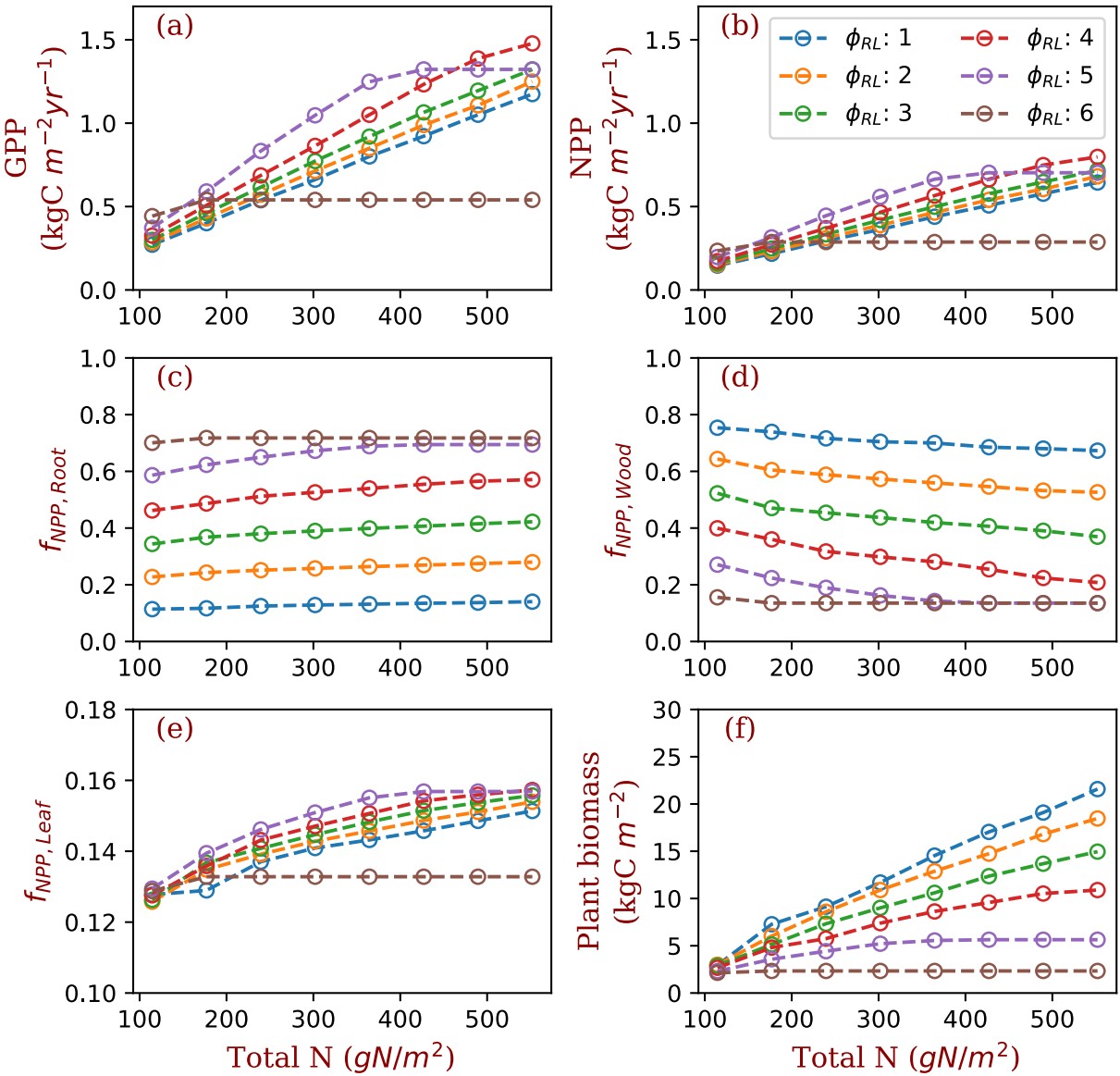


**Figure 3. GPP, NPP, Allocation and Plant biomass at equilibrium state simulated by monoculture runs**. GPP: Gross primary production; NPP: Net primary production; $f_{NPP,x}$: the fraction of NPP allocated to $x$, where $x$ is Root (fine roots), Leaf (leaves in crown), or Wood (including tree trunk, stems, and coarse roots). The data are from the averages of the model run years from 1400 and 1800. Each model run is initiated with one PFT with fixed ratio of fine root area to leaf area ($\varphi_{RL}$).

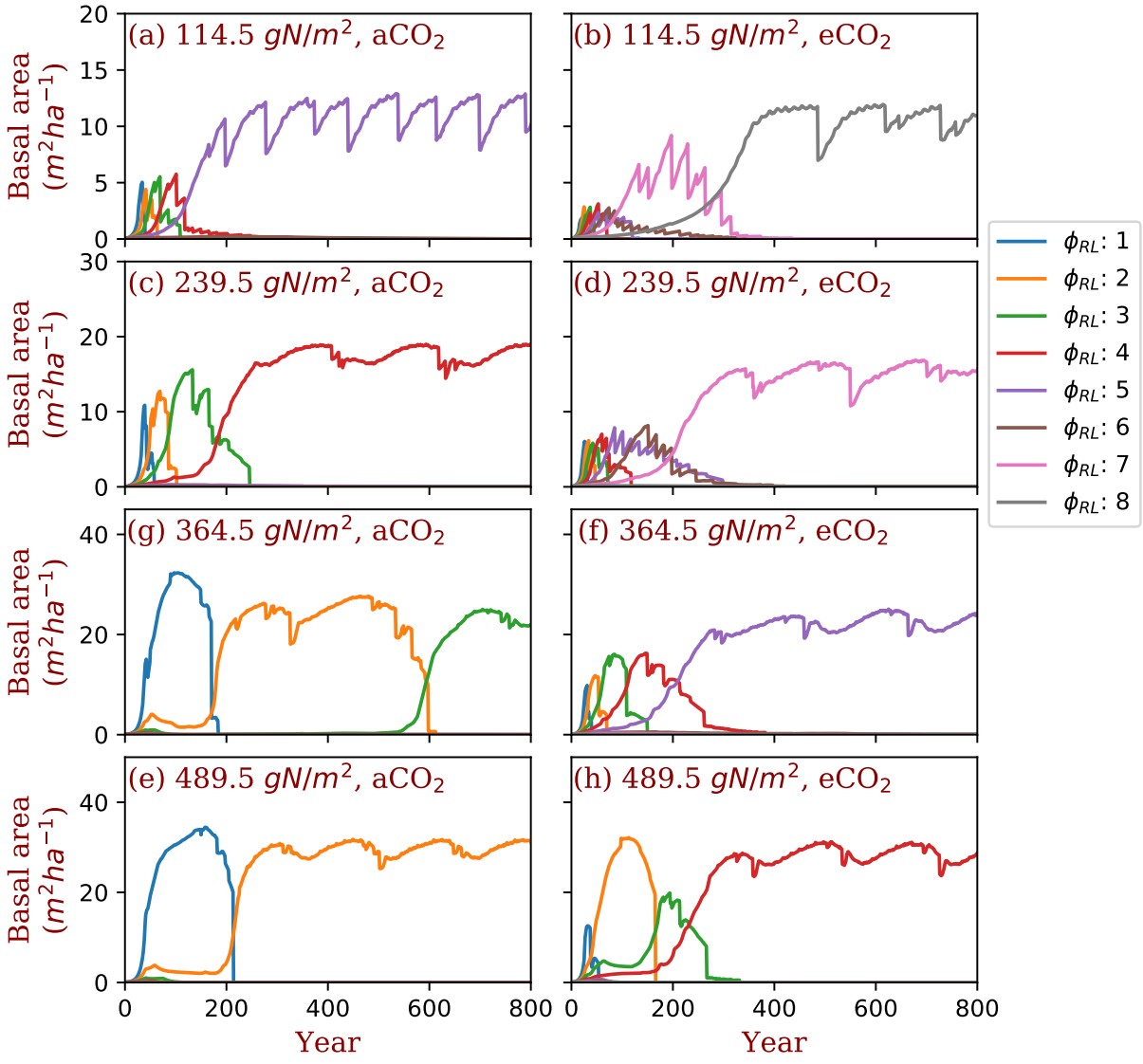

**Figure 4 Successional patterns of polyculture runs I at ambient and elevated [CO₂]**

**concentrations**. $\varphi_{RL}$ is the fixed ratio of fine root area to leaf area of a particular strategy.

We used two sets of polyculture runs to look for the $\varphi_{RL}$ that is closest to competitively

optimal. In the polyculture runs I, where $\varphi_{RL}$ ranges from 1 to 8 at all nitrogen levels, the

winning strategy ($\varphi_{RL}$) increases from 5 to 2 as the total nitrogen increases from 114.5 g N m⁻² to

489.5 g N m⁻² at ambient [CO₂] (380 ppm) (Fig. 4: a, c, g, e). Elevated [CO₂] (580 ppm) shifts

the winning strategy to higher ($\varphi_{RL}$) at all the total nitrogen levels. As shown in Fig. 4, the
winning strategy shifts from $\varphi_{RL}$=5 to $\varphi_{RL}$=8 at 114.5 g N m$^{-2}$ and from $\varphi_{RL}$ =2 to $\varphi_{RL}$=4 at 489.5
g N m$^{-2}$. In some situations (e.g., Fig. 4: g and Figs. S2 and S3), it takes a long time for the most
competitive PFTs to out-compete the previously dominant PFTs because of the sequential
replacement of dominant PFTs during the course of succession and the slow growth rate of trees
in understory.

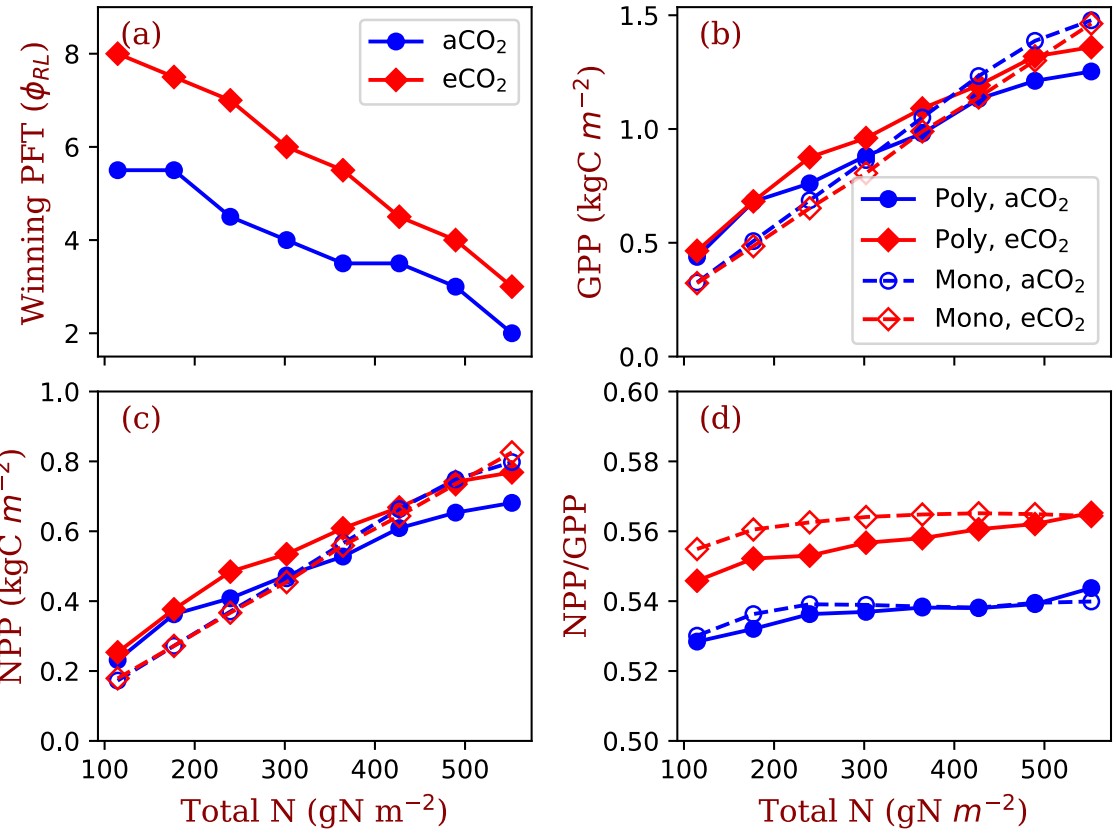


**Figure 5 Winning PFTs ($\varphi_{RL}$, a) in polyculture runs II and equilibrium Gross Primary**
**Production (GPP, b), Net Primary Production (NPP, c), and Carbon Use Efficiency**
**(NPP/GPP, d) at two CO$_2$ concentrations (aCO$_2$: 380 ppm; eCO$_2$: 580 ppm).** The closed
symbols with solid line represent polyculture runs. The open symbols with dashed lines represent
monoculture runs (only $\varphi_{RL}$=4 shown in this figure). $\varphi_{RL}$ is the fixed ratio of fine root area to leaf
area of a particular strategy.

Based on the shifts of the winning $\varphi_{RL}$ from ambient [$CO_2$] to elevated [$CO_2$] at the eight
nitrogen levels, we designed the polyculture runs II with high resolution of $\varphi_{RL}$ and calculated
their GPP, NPP, allocation, and plant biomass at equilibrium state. The of $\varphi_{RL}$ of the winning
PFTs decreases from 5.5 to 2 at ambient [$CO_2$] and from 8.0 to 3.0 at elevated [$CO_2$] as total
nitrogen increases from 114.5 g N m$^{-2}$ to 552.0 g N m$^{-2}$. The equilibrium GPP and NPP increase
with total nitrogen at values similar to those of the monoculture runs (Fig. 5: b and c). However,
the $CO_2$ stimulation of NPP increases with total nitrogen in the polyculture runs more than it in
the monoculture runs. Elevated [$CO_2$] increases carbon use efficiency (defined as the ratio of
NPP to GPP in this study, NPP/GPP) in both the monoculture and polyculture runs (Fig. 5: d).
Also, the dependence of NPP:GPP ratio on nitrogen is higher in the polyculture runs than it in
the monoculture runs (Fig. 5:c).

Allocation of NPP to leaves increases with nitrogen in all conditions, i.e. both competition

and monoculture at both ambient [$CO_2$] and elevated [$CO_2$] (Fig. 6: a). Foliage NPP is similar in
these four model runs when nitrogen is low. At high nitrogen (>400 g N m$^{-2}$), polyculture runs
have higher foliage NPP than the monoculture runs generally. Allocation to leaves is relatively
stable across the nitrogen gradient at the two [$CO_2$] levels (Fig. 6: b). The fraction of NPP
allocated to leaves changes little with nitrogen (Fig. 6: b) and it is universally higher at ambient
[$CO_2$] than it at elevated [$CO_2$].

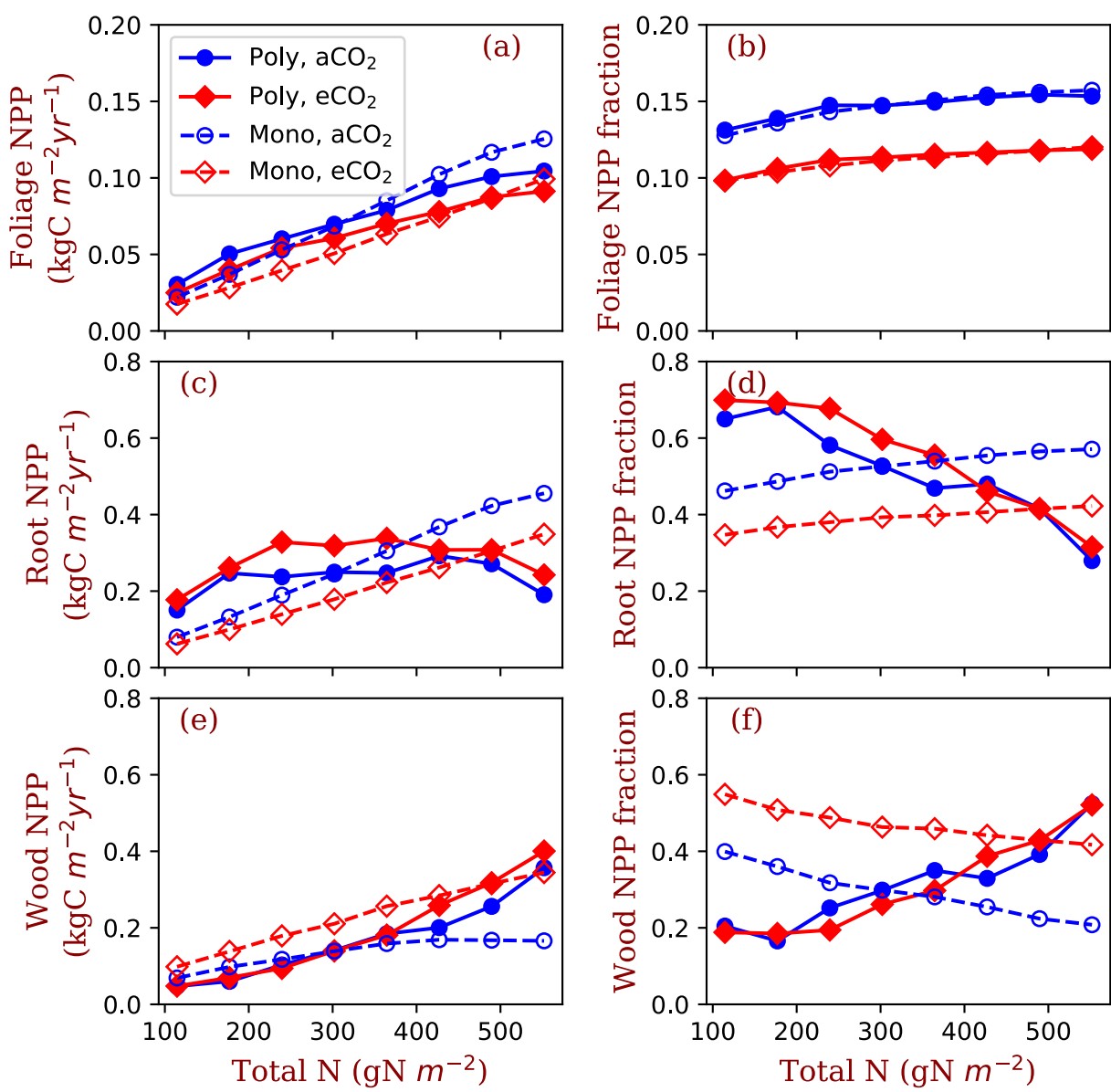


**Figure 6** Allocation to leaves, fine roots, and wood tissues of the competition and monoculture runs at the eight total nitrogen levels and two $CO_2$ concentrations (a$CO_2$: 380 ppm; e$CO_2$: 580 ppm). The panels a, c, and e show the NPP allocated to the tissues and the panels b, d, and f show the fractions of the allocation in total NPP. The closed symbols with solid line represent polyculture runs (Poly). The open symbols with dashed lines represent monoculture runs (only $\varphi_{RL}$=4 shown in this figure, Mono). $\varphi_{RL}$ is the fixed ratio of fine root area to leaf area of a particular strategy.


Fine root NPP does not significantly change with ecosystem total nitrogen in polyculture
runs, whereas it increases monotonically with increasing nitrogen in monoculture runs (Fig. 6: c).
Elevated $[CO_2]$ increases fine root allocation at low nitrogen in polyculture runs but decreases
root allocation irrespective of nitrogen in monoculture runs (Fig. 6: c). The fraction of NPP
allocated to fine roots decreases with nitrogen at both $CO_2$ concentrations in polyculture runs but
it increases slightly in monoculture runs (Fig. 6: d). In monoculture runs, elevated $[CO_2]$ reduces
the fraction of NPP allocated to fine roots at all nitrogen levels. In polyculture runs, fractional
allocation to fine roots increases at elevated $[CO_2]$ when nitrogen is low (e.g., 114.5-302 g N m$^-$
$^2$) and decrease at elevated $[CO_2]$ when nitrogen is high (e.g., 364-552 g N m$^{-2}$).
In the reverse of the fine root response, NPP allocation to woody tissues increases with
total nitrogen in both competition and monoculture runs (Fig. 6: e). In polyculture runs, the
fraction of allocation to woody tissues decreases at elevated $[CO_2]$ when ecosystem total
nitrogen is low (e.g., $114 - 245$ g N m$^{-2}$) and increases at elevated $[CO_2]$ when ecosystem total
nitrogen is high (e.g., $302 - 552$ g N m$^{-2}$).
As a result of the changes in competitively-optimal $\varphi_{RL}$, plant biomass increases
dramatically with ecosystem total nitrogen in polyculture runs compared with that in
monoculture runs (Fig. 7: a). The effects of elevated $[CO_2]$ on plant biomass increase with
nitrogen in polyculture runs but are constant overall in monoculture runs (Fig. 7: b). Compared
with the full spread of monoculture runs with $\varphi_{RL}$ ranging from 1 to 6, polyculture runs have
high root allocation at low nitrogen and low root allocation at high nitrogen due to changes in the
dominant competitive allocation strategy, which amplifies plant biomass responses to elevated
$[CO_2]$ with increasing nitrogen (Fig. 7: c and d).

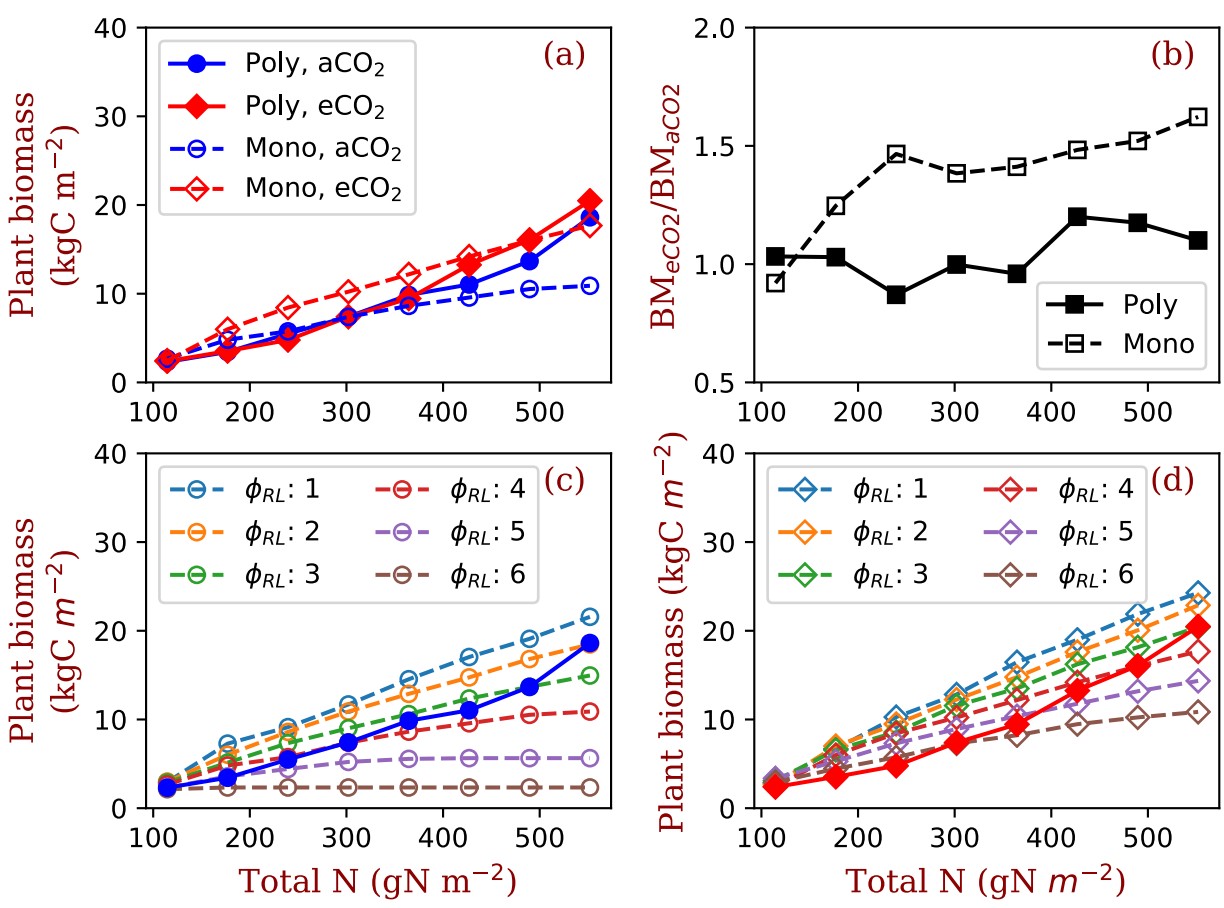

**Figure 7 Plant biomass responses to elevated [CO₂] and nitrogen**

Panel a shows the equilibrium plant biomass (means of simulated plant biomass from model run year 1400 to 1800) in polyculture runs and monoculture runs (only $\varphi_{RL}$=4 shown as an example). Panel b shows the ratio of simulated plant biomass at elevated [CO₂] to ambient [CO₂] for both competition and monoculture runs. Panels c and d show the comparisons with monoculture runs with $\varphi_{RL}$ increasing from 1 to 6 at ambient (c) and elevated [CO₂] (d). The closed symbols with solid line represent polyculture runs. The open symbols with dashed lines represent monoculture runs ($\varphi_{RL}$ ranges from 1 to 6). $\varphi_{RL}$ is the fixed ratio of fine root area to leaf area of a particular strategy. aCO₂: 380 ppm; eCO₂: 580 ppm.


## 4 Discussion

Our simulations show that the predicted responses of individual plants to elevated $[CO_2]$
can be significantly changed by explicit inclusion of competition processes. Here, the major
tradeoff for light- and N-limited trees is the relative allocation between stems and fine roots
(Dybzinski et al. 2011). Although the wood allocation (and thus carbon sequestration potential)
of every PFT used in this study increases under elevated $[CO_2]$ at all nitrogen levels (e.g. Fig. 6:
e dashed lines), only those PFTs that allocate more to fine roots (with lower carbon sequestration
potential) can survive competition under elevated $[CO_2]$ (Fig. 6: c solid lines). Put together,
explicit inclusion of competition processes reduces the expected increase in biomass (and thus
carbon sequestration potential) under elevated $[CO_2]$ compared with simulations that do not
include competition processes (Fig. 7: b).
Since there is a lack of direct observations or experiments to quantitatively validate the
long-term patterns predicted by our model, we did not calibrate it to fit observations at Harvard
Forest. In the following section, we analyze the model processes in detail and validate our
modeling approach by comparing the general patterns from observations and experiments with
model predictions. These comparisons also shed light on the modeling of allocation and
vegetation responses to elevated $[CO_2]$.

### 4.1 Mechanisms of game-theoretic allocation modeling and simulation results validation

In our model, the allocation of carbon and nitrogen within an individual tree is based on
allometric scaling (Eq. 2), functional relationships (Eq. 3), and optimization of resource usage
(Eq. 7). Generally, the allometric scaling relationships define the maximum leaf and fine root
surface area at a given tree size, and the functional relationships define the ratios of leaf area to
sapwood cross-sectional area and fine root surface area. These rules are commonly used in
ecosystem models (Franklin et al., 2012) and have been shown to generate reasonable
predictions (De Kauwe et al., 2014; Valentine and Mäkelä, 2012). These rules implicitly define
the priority of allocation to leaves and fine roots but allow for structurally-unlimited stem growth
when resources (carbon and nitrogen in this study) are available (i.e., the remainder goes to
stems after leaf and fine root growth) and NSC is not accumulated exaggeratedly when
ecosystem nitrogen is limited (Fig. S6).

We used a tuning parameter, maximum leaf and fine root allocation, $f_{LFR,max}$, to constrain

the maximum allocation to leaves and fine roots  in order to maintain a minimum growth rate of
wood in years of low productivity. This is consistent with wood growth patterns in temperate
trees, where new wood tissues must be continuously produced (especially early in the growing
season) to maintain the functions of tree trunks and branches (Cuny et al., 2012; Michelot et al.,
2012; Plomion et al., 2001). This parameter does not change the fact that leaves and fine roots
are the priority in allocation, since allocation ratios to stems are around 0.4~0.7 in temperate
forests (Curtis et al., 2002; Litton et al., 2007). With a value of 0.85, parameter $f_{LFR,max}$ seldom
affects the overall carbon allocation ratios of leaves, fine roots, and stems. If $f_{LFR,max} = 1$ (i.e., the
highest priority for leaf and fine root growth), simulated trunk radial growth would have
unreasonably high interannual variation because leaf and fine root growth would use all carbon
to approach to their targets, leaving nothing for stems in some years of low productivity.

The simulation of competition for light and soil resources is based on two fundamental

mechanisms: 1) competition for light is based on the height of trees according to the PPA model,
which assumes trees have perfectly plastic crown to capture light via stem (trunk) and branch
phototropism (Strigul et al., 2008); and 2) individual soil N uptake is linearly dependent on the
fine root surface area of an individual tree relative to that of its neighbors (Dybzinski et al., 2019;
McMurtrie et al., 2012; Weng et al., 2017). These two mechanisms define an allocational
tradeoff between wood and fine roots for carbon and nitrogen investment in different $CO_2$
concentrations and nitrogen environments. Including explicit competition for these resources to
determine the dominant strategies results in very different predicted allocation patterns – and
thus ecosystem level responses – than those of strategies in the absence of competition. For
example, fractional wood allocation increases with increasing nitrogen availability under
competitive allocation but decreases – *the opposite qualitative response* – under a fixed strategy
(Fig. 6: f). Consequently, equilibrium plant biomass is predicted to increase much more with
increasing nitrogen availability under a competitive strategy (Fig. 4: c, d). In nature, the effects
of competition on dominant plant traits may occur through species replacement or community
assembly (akin to the mechanism in our model) (e.g., Douma et al., 2012), but it may also occur
through adaptive plastic responses or in-place sub-population evolution of ecotypes (Grams and
Andersen, 2007; McNickle and Dybzinski, 2013; Smith et al., 2013).

Generally, the predictions from competitively-optimal allocation strategies predicted by

our model can be found in large scale forest censuses and site-level experiments, such as: 1) high
nitrogen environments (i.e., productive environments) favor high wood allocation and low root
allocation (Litton et al., 2003; Poorter et al., 2012); 2) elevated [$CO_2$] increases root allocation
(Drake et al., 2011; Iversen, 2010; Jackson et al., 2009; Nie et al., 2013; Smith et al., 2013); 3)
low nitrogen availability limits vegetation biomass responses to elevated [$CO_2$] as a result of
high root allocation or root exudation (Jiang et al., 2019a; Norby and Zak, 2011); and 4)
increases in vegetation biomass at elevated $[CO_2]$ are largely due to high wood allocation (Norby
and Zak, 2011; Walker et al., 2019). These predictions emerge from the fundamental
assumptions of our model without tuning parameters to fit the data, providing some confidence
in the robustness of our approach.

The literature on experimental responses of plant community to elevated $[CO_2]$ shows

that the responses vary with site characteristics, forest composition, stand age, plant
physiological responses, and soil microbial feedbacks (Norby and Zak, 2011; Terrer et al., 2016,
2018). For example, in Duke Free Air $CO_2$ Enhancement (FACE) experiment, where the major
trees are loblolly pine (*Pinus taeda*), increases in root production at elevated $[CO_2]$ stimulated
increased nitrogen supply that allowed the forest to sustain higher productivity (Drake et al.,
2011). However, in Oak Ridge FACE, where the major trees are sweetgum (*Liquidambar*
*styraciflua*), increased fine-root production under elevated $[CO_2]$ did not result in increased net
nitrogen mineralization and increases in root production declined after eight years of $CO_2$
enhancement (Iversen, 2010; Norby and Zak, 2011). In EucFACE (Jiang et al., 2019a), where the
major trees are *Eucalyptus tereticornis* and the soil is infertile, trees significantly increased their
root exudation under limited nutrient supplies but had no significant increase in biomass in
response to elevated $[CO_2]$. The BangorFACE experiment (Smith et al., 2013) found that
interspecific competition (*Alnus glutinosa*, *Betula pendula* and *Fagus sylvatica*) resulted in
greater increases in root biomass at elevated $[CO_2]$. Leaf area index (LAI) responses to elevated
$[CO_2]$ are also highly varied. As summarized by Norby and Zak (2011), low LAI (in this case,
open canopy) sites showed significant increases in LAI and high LAI (in this case, closed
canopy) sites showed low increases or even decreases in LAI. They concluded that LAI in
closed-canopy forests is not responsive to elevated [$CO_2$] (Norby et al., 2003; Norby and Zak,

2011).

The nature of developing a model with generic assumptions and balanced processes
reduces its capability to predict all of these responses. For example, plants have a variety of
physiological mechanisms to deal with excessive carbon supply when plant demand (i.e., "sink")
is relatively low (Fatichi et al., 2019; Körner, 2006), such as down-regulating leaf photosynthesis
rate by the accumulated assimilates (Goldschmidt and Huber, 1992) or respiring excessive
carbohydrates to regenerate substrates for photosynthesis (Atkin and Macherel, 2009). But these
mechanisms are short-term physiological responses (minutes to hours, sometimes days) for
plants in situations of temporary nitrogen shortage, high irradiation, or drought stress. It is not
"economically" sustainable in an infertile environment to maintain highly productive leaves but
often suppress their photosynthesis or respire a large portion of their assimilated carbon.
Root exudation is a critical process for plants. It can stimulate soil organic matter
decomposition and nitrogen mineralization to facilitate soil nitrogen supply at the expense of
carbon (Cheng, 2009; Cheng et al., 2014; Drake et al., 2011; Phillips et al., 2011). The process of
root exudation has been adopted by many models to couple with microbial processes in the
determination of soil organic matter decomposition (Sulman et al., 2014; Wieder et al., 2014,
2015). Some carbon-only models, e.g., LM3 (Shevliakova et al., 2009), the parent model of this
one, and TECO (Luo et al., 2001), incorporate root exudation to put extra carbon into the soil in
order to avoid down-regulating canopy photosynthesis or overestimating vegetation biomass,
both of which had been tuned against data. However, in a demographic competition model like
this one, individual plants cannot reap a reward from root exudation as they do in nature when
the microbial activities are not fully coupled and the nitrogen in soil is assumed fully accessible
by roots of all individuals. Therefore, root exudation is not a competitive strategy in the system
defined by the assumptions of this model.

Since the purpose of this study is to explore long-term ecological strategies in different

but relatively stable environments, we did not include these processes, especially since they
present additional challenges in balancing the complexity of the tradeoffs between modeled
demographic processes and plant traits. However, the lack of these processes does limit the
predictions of instantaneous responses to variation in environmental conditions or resource
supply and possibly of some long-term vegetation characteristics as well. For example, our
model predicts reduced LAI under nitrogen limitation (Fig. S7) based on first principles, but it is
incidentally the only mechanism that reduces the whole-canopy photosynthesis rate in our
model. There are mechanisms that increase nitrogen use efficiency at the expense of carbon by
increasing LMA and therefore leaf longevity to maintain high LAI and high canopy-level
photosynthesis rates (Aerts, 1995, 1999; Aerts and Chapin, 1999; Givnish, 2002). We did not
include these mechanisms in our simulations, although they are well-developed in this model
(Weng et al. 2017), because we wished to focus on the strategy of allocation. The clear
descriptions of our model's assumptions, its traceable processes, and inclusion of the tradeoffs
involved in aboveground and belowground competition provide a useful benchmark from which
to incorporate additional mechanisms and tradeoffs.

**4.2 Root overproliferation vs. wood allocation**

The allocation strategy that maximizes site vegetation biomass allocates very little to fine

roots (Figs. 3 and S1). In contrast, the competitively optimal strategy allocates more carbon to
fine roots, termed "fine-root overproliferation" in the literature (Gersani et al., 2001; McNickle
and Dybzinski, 2013; O'Brien et al., 2005). It is the result of a competitive "arms race": while
increasing fine root area under elevated [$CO_2$] does not result in more nitrogen for an individual,
failing to do so would cede some of that individual's nitrogen to its neighbors. Because most
nitrogen uptake is via mass flow and diffusion (Oyewole et al., 2017) and because both of these
mechanisms depend on sink strength, individuals with *relatively* greater fine root mass than their
neighbors take a greater share of nitrogen, as was recently demonstrated empirically (Dybzinski
et al., 2019; Kulmatiski et al., 2017). Thus, fine roots may overproliferate for competitive
reasons relative to lower optimal fine root mass in the hypothetical absence of an evolutionary
history of competition (Craine, 2006; McNickle and Dybzinski, 2013). This may also explain
why root C:N ratio is highly variable (Dybzinski et al., 2015; Luo et al., 2006; Nie et al., 2013): a
high density of fine roots in soil may be more important than the high absorption ability of a
single root in competing for soil nitrogen in the usually low mineral nitrogen soils.

Root overproliferation is still controversial in experiments.  For example, Gersani et al.

(2001) and O'Brien (2005) found that competing plants generated more roots than those
growing in isolation; whereas McNickle and Brown (2014) found that competing plants
generated comparable roots to those growing in isolation. Compared to modeled roots, real roots
are far more adaptive and complex at modifying their growth patterns in response to soil nutrient
and water dynamics (Hodge, 2009). The root growth strategies in response to competition also
vary with species (Belter and Cahill, 2015).  The mechanisms of self-recognition of inter- and
intra- roots also can lead to varied behavior of root growth (Chen et al., 2012). However, all of
the aforementioned studies considered only *plastic* root overproliferation, where individuals
produce more roots in the presence of other individuals than they do in isolation, analogous to
stem elongation of crowded seedlings (Dudley and Schmitt, 1996). A portion of root
overproliferation may also be *fixed*, analogous to trees that still grow tall even when grown in
isolation. Dybzinski et al. (2019) showed that plant community nitrogen uptake rate was
independent of fine root mass in seedlings of numerous species, suggesting a high degree of
fixed fine root overproliferation. To improve root competition models, more detailed
experiments that control root growth should be conducted to quantify the marginal benefits of
roots in isolated, monoculture, and polyculture environments.

At high soil nitrogen, height-structured competition for light (also a game-theoretic

response, Falster and Westoby, 2003; Givnish, 1982) prevails, and trees with greater *relative*
allocation to trunks prevail.  The balance between these two competitive priorities (fine roots vs.
stems) can be observed in our model predictions as a shift from fine root allocation to wood
allocation as soil nitrogen increases. The increases in the critical height (i.e. the context-
dependent height of the shortest tree in canopy layer in the PPA) from low nitrogen to high
nitrogen indicates a shift from the importance of competition for soil nitrogen to the importance
of competition for light as ecosystem nitrogen increases (Fig. S8). Because the most competitive
type shifts from high fine root allocation to low fine root allocation as ecosystem total nitrogen
increases, increases in NPP and plant biomass across the nitrogen gradient are greater than the
increases in NPP and plant biomass assuming allocational strategies in the absence of
competition (Fig. 3). This greatly reduces the carbon cost of belowground competition as
ecosystem total nitrogen increases. The decrease in the fraction of NPP allocated to leaves at
elevated $[CO_2]$ (Fig. 6: b) occurs because of increases in total NPP and nearly constant absolute
NPP allocation to foliage (Fig. 6: a).

**4.3 Model complexity and uncertainty**

Compared with the conventional pool-based vegetation models that use pools and fluxes

to represent plant demographic processes at a land simulation unit (e.g., grid or patch), VDMs
add two more layers of complexity. The first is the inclusion of stochastic birth and mortality
processes of individuals (i.e., demographic processes). These processes allow the models to
predict population dynamics and transient vegetation structure, such as size-structured
distribution and crown organization (e.g., Moorcroft et al., 2001; Strigul et al., 2008). With
changes in vegetation structure, allocation and mortality rates can change, generating a different
carbon storage accumulation curve compared with those predicted by pool-based models where
vegetation structure is not explicitly represented (e.g., Weng et al., 2015). The second is the
simulated shift in dominant plant traits during succession due to the shifting of competitive
outcomes among different PFTs, which changes the allocation between fast- and slow-turnover
pools and thus the parameters of allocation and the residence time of carbon in the ecosystem.

Together, these mechanisms may alter long-term predictions of terrestrial carbon cycl due

to changes in PFT-based parameters (Dybzinski et al., 2011; Farrior et al., 2013; Weng et al.,
2015). As described in Introduction, current pool-based models can be described by a linear
system of equations characterized by the key parameters of allocation, residence time, and
transfer coefficients (Eq. 1) with the rigid assumption of unchangeable plant types (Luo et al.,
2012; Xia et al., 2013). In VDMs however, allocation, residence time, leaf traits, phenology,
mortality, plant forms, and their responses to climate change are all strategies of competition
whose success varies with the environmental conditions and the traits of the individuals they are
competing against.

Many tradeoffs between plant traits can shift in response to environmental and biotic

changes, limiting the applicability of varying a single trait, as we have in this study. For example,
allocation, leaf traits, mycorrhizal types, and nitrogen fixation can all change with ecosystem
nitrogen availability (Menge et al., 2017; Ordoñez et al., 2009; Phillips et al., 2013; Vitousek et
al., 2013). The unrealistic effects of model simplification can be corrected by adding important
tradeoffs that are missing. For example, the positive feedback between root allocation and SOM
decomposition plays a role in mitigating the effects of tragedies of the commons of root over-
proliferation (e.g., Gersani et al., 2001; Zea-Cabrera et al., 2006) due to a negative feedback
induced by root turnover. High root allocation increases the decomposition rate of SOM and the
supply of mineral nitrogen because of the high turnover rate of root litter, which favors a strategy
of high wood allocation and reduces the competitive optimal fine root allocation. This negative
feedback indicates that the model structure is flexible and that we can incorporate correct
mechanisms step by step to improve model prediction skills. Testing single strategies is still a
necessary step to improving our understanding of the system and prediction skills of the models,
though it could lead to unrealistic responses sometimes.

We found that model predictions can differ significantly in response to seemingly-small

variations in basic assumptions or quantitative relationships. For example, our model predicts
that the ratio of plant biomass under elevated $[CO_2]$ relative to plant biomass under ambient
$[CO_2]$ should increase with increasing nitrogen due to the shift of carbon allocation from fine
roots to woody tissues. In contrast, the analytic model of Dybzinski *et al*. (2015) predicts that the
ratio of plant biomass under elevated $[CO_2]$ relative to plant biomass under ambient $[CO_2]$
should be largely independent of total nitrogen because of an increasing shift in carbon allocation
from long-lived, low-nitrogen wood to short-lived, high-nitrogen fine roots under elevated $[CO_2]$
and with increasing nitrogen. This significant difference between these two predictions traces
back to differences in how fine root stoichiometry is handled in the two models. In the model of
Dybzinski *et al*. (2015), the fine root C:N ratio is flexible and the marginal nitrogen uptake
capacity per unit of carbon allocated to fine roots depends on its nitrogen concentration. Like the
model presented here, the model of Dybzinski *et al*. (2015) predicts decreasing fine root mass
with increasing nitrogen availability. *Unlike* the model presented here (which has constant fine
root nitrogen concentration), the model of Dybzinski et al. (2015) predicts increasing fine root
nitrogen concentration with increasing nitrogen availability. As a result, there is less nitrogen to
allocate to wood as nitrogen increases in the model of Dybzinski *et al*. (2015) than there is in the
model presented here. These countervailing factors even out the ratio of plant biomass under
elevated [$CO_2$] relative to plant biomass under ambient [$CO_2$] across the nitrogen gradient in
Dybzinski *et al*. (2015), whereas their absence amplifies this ratio with increasing nitrogen in the
model presented here. Our ability to diagnose and understand this discrepancy highlights the
utility of deploying closely-related analytical and simulation models (Weng et al., 2017).
We conducted simulations only at one site for the purpose of exploring the general
patterns of competitively optimal allocation strategies and their responses to elevated [$CO_2$] at
different nitrogen availabilities. We can speculate about shifts in the competitively optimal
allocation strategy in different forest biomes by considering the effects of temperature on soil
nitrogen supply via the SOM's decomposition rate and its positive effect on net nitrogen
mineralization. For example, the SOM decomposition rate is usually high in warm regions and
low in cold regions (Davidson and Janssens, 2006) assuming there are no water limitations and
SOM is equilibrated with carbon input. According to our model, allocation to roots is high in low
nitrogen supply conditions (cold regions) and low in high nitrogen supply conditions (warm
regions). This pattern can be found from temperate to boreal forest zones (Cairns et al., 1997;
Gower et al., 2001; Reich et al., 2014; Zadworny et al., 2016). Temperature also alters NPP, i.e.,
carbon supply: as temperature goes down, NPP decreases and nitrogen demand decreases,
alleviating nitrogen limitation and leading to shifts of allocation to stems. So, the differences in
temperature effects on photosynthesis and SOM decomposition will determine competitive
allocation strategy. Since SOM decomposition is more sensitive to temperature than gross
primary production is at long-temporal and large-spatial scales (Beer et al., 2010; Carey et al.,
2016; Crowther et al., 2016), our model suggests that allocation will shift to wood in a warming
world. Whether the carbon stored in that wood is enough to offset the carbon released from
increasing soil respiration is a critical question.

Water is also a critical factor affecting allocation and its responses to elevated $[CO_2]$.

Low soil moisture usually leads to high allocation to roots (Poorter et al., 2012). Elevated $CO_2$
can reduce transpiration (as found in our study as well, Figs. S9~S11) and therefore increase soil
moisture, resulting in increases in allocation to stems and aboveground biomass (Walker et al.,
2019). A game-theoretic modeling study using the PPA framework shows that the competitively
optimal allocation strategy shifts to high wood allocation at elevated $[CO_2]$ in environments with
water limitation (Farrior et al., 2015). This is opposite to the elevated $[CO_2]$ effects on allocation
in nitrogen-limited environments as simulated in this study. According to field experiments, fine
root allocation is more responsive to nitrogen changes than it to soil moisture changes (Canham
et al., 1996; Poorter et al., 2012). Poorter et al. (2012) attribute the mechanisms to the optimal
strategies in response to the relative stable nitrogen supply and stochastic water input in soil. The
vertical distribution of roots and the contributions of roots in different layers to water and
nitrogen uptake also suggest that the uptake of soil nutrients are dominant in shaping root system
architecture (Chapman et al., 2012; Morris et al., 2017), though root growth and turnover are
flexible and sensitive to nitrogen and water supply (Deak and Malamy, 2005; Linkohr et al.,
2002; Pregitzer et al., 1993).

**4.4 Common principles for allocation modeling and implications**

As shown in model inter-comparison studies, the mechanisms of modeling allocation

differ very much, leading to high variation in their predictions (e.g., De Kauwe et al. 2014).
Calibrating model parameters to fit data may not increase model predictive skill because data are
often also highly variable. Franklin et al. (20`12) suggest that in order to build realistic and
predictive allocation models, we should correctly identify and implement fundamental principles.
Our model predicts similar patterns to those predicted by the model of Valentine and Mäkelä
(2012), which has very different processes of plant growth and allocation. However, these two
models share fundamental principles, including 1) evolutionary- or competitive-optimization, 2)
capped leaves and fine roots at given tree sizes, 3) structurally unlimited stem allocation (i.e.,
optimizing carbon use) because the woody tissues can serve as unlimited sink for surplus carbon,
and 4) height-structure competition for light and root-mass-based competition for soil resources.
The principles 2 and 3 are commonly used in models (De Kauwe et al., 2014; Jiang et al.,
2019b). However, the different rules of implementing them (e.g., allometric equation, functional
relationships, etc. ) lead to highly varied predictions (as shown in De Kauwe et al., 2014), though
model formulations may be very similar.

In competitively-optimal models, such as this study and also Valentine and Mäkelä (2012),

the competition processes generate similar emergent patterns by selecting those that can survive
in competition, regardless the details of those differences. The competition processes also make
the details of allocation settings for a single PFT and their direct responses to elevated [$CO_2$] less
important, because competition processes will select out the most competitive strategy from
diverse strategies in response to changes in [$CO_2$] and nitrogen. Our study and Valentine and
Mäkelä (2012),  posit a fundamental tradeoff between light competition and nitrogen competition
via allocation based on insights gained from simpler models (e.g., Dybzinski et al., 2015; Mäkelä
et al., 2008) for predicting allocation as an emergent property of competition. One advantage of
building a model in this way is that the vegetation dynamics are predicted from first principles,
rather than based on the correlations between vegetation properties and environmental
conditions. With these first principles, the models can produce reasonable predictions, though the
details of physiological and demographic processes vary among models.
For vegetation models designed to predict the effects of climate change, the important
operational distinction is that the fundamental rules cannot or will not change as climate changes.
Nor, presumably, will the underlying ecological and evolutionary processes change as climate
changes. The emergent properties can change as climate changes however, and the models built
on the "scale-appropriate" unbreakable constraints and ecological and evolutionary processes
will be able to accurately predict changes in emergent ecosystem properties (Weng et al., 2017).
In our opinion, the scientific effort to build better models is better served by understanding
unrealistic predictions than by "fixing" them with unreliable mechanisms when there is a lack of
data or theory to make them consistent with observations. Validating assumptions and initial
responses are critical, and the long-term responses can be validated via spatial patterns.
This modeling approach also demands improvement in model validation and benchmarking
systems (Collier et al., 2018; Hoffman et al., 2017). As shown in this study, allocation responses
to elevated $CO_2$ at different nitrogen levels in monoculture runs are opposite to those in
competitive-allocation runs. For example, in monoculture runs, elevated [$CO_2$] increases wood
allocation and decreases fine root allocation at low nitrogen; whereas in competitive-allocation
runs elevated $[CO_2]$ leads to low wood allocation and high fine root allocation. Simply
calibrating our model against short-term observational data may improve the agreement with
observations but would not change the model's predictions because the model's predictions
emerge from its fundamental assumptions.

**811 5 Conclusions**

Our study illustrates that including the competition processes for light and soil resources in
a game-theoretic vegetation demographic model can substantially change the prediction of the
contribution of ecosystems to the global carbon cycle. Allowing the model to explicitly track the
competitive allocation strategies can generate significantly different ecosystem-level predictions
(e.g., biomass and ecosystem carbon storage) than those of strategies in the absence of explicit
competition. Building such a model requires differentiating between the unbreakable tradeoffs of
plant traits and ecological processes from the emergent properties of ecosystems. Drawing on
insights from closely-related analytical models to develop and understand more complicated
simulation models seems, to us, indispensable. Evaluating these models also requires an updated
model benchmarking system that includes the metrics of competitive plant traits during the
development of ecosystems and their responses to global change factors.

**824 Acknowledgements**

This work was supported by NASA Modeling, Analysis, and Prediction (MAP) Program
(NNH16ZDA001N-MAP), USDA Forest Service Northern Research Station (Agreement 13-JV-
11242315-066), and the Carbon Mitigation Initiative at Princeton University. C.E.F
acknowledges support from the University of Texas at Austin.

**830    Codes and data availability**

The model codes used in this study, simulated data, and Python scripts used in this study are in
Github (https://github.com/wengensheng/BiomeE-Allocation).

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
