# Peer review of "Competition alters predicted forest carbon cycle responses to nitrogen availability and"

_Biogeosciences, 2019_

## Referee Comment (RC1) · Anonymous Referee #1 · 7 Mar 2019

The current paper aims to compare the predictions of biomass allocation within a vegetation demographic model (VDM) with explicit competition versus a model without competition, under elevated CO$_2$ across a nitrogen availability gradient. To this end, the authors use a derivation of an existing VDM, where the only process that varies is the biomass allocation. The authors then present comparisons between the two model versions at equilibrium for one site. The topic of predicting vegetation allocation pattern for different nutrient limitation states is an important one, and one that existing vegetation model often have trouble with.The approach of having one model with two different process representations is also very valuable as it can pinpoint model differences to the exact processes in question.

However, the value of this study is largely obscured by the way the model is presented and discussed, making it very difficult for the reader to link between model assumptions, results and model implications.

**Major comments**

It is unclear to me if this analysis actually shows a difference between a model with and one without competition or simply a difference between a model with fixed and one with flexible allocation. As the authors themselves point out in the introduction, pool-based vegetation models often incorporate a flexible allocation scheme based on nutrient and water availability. It is currently unclear if a model with such a scheme would perform differently from the competition model included here,

One of the key assumptions of the model is the order of allocation (l 245): first a fixed fraction allocated to the sapwood, then allocation to the leaves and roots, then if there is available C and N left, to sapwood and fruit. And, most importantly, any carbon left in excess because of N limitation is allocated to sapwood. This last step could have some interesting implications for light competition under N limitation, and can maybe explain the different wood allocation patterns observed for the competition models. This model assumption needs to be justified and discussed.

While I understand that this is a theoretical study, and such studies are valuable and note every study needs to show a comparison with data, there is a complete lack of model reality checking. Especially when the two model versions show contrasting allocation patterns, there should be a way to determine what the reality is. There is a wealth of data from FACE experiments, N addition experiments, long-term measurements, soil N gradients etc. I believe it would be very interesting to have a section in the discussion comparing the current model predictions with general observed trends.

As it is, the discussion mostly contains comparisons with previous models from the same model family, which while I think is probably relevant to the authors for model development purposes, is of little interest to the general audience.

**Detailed comments**

L 52 I'm not sure there are any ESM's that just simulate the nitrogen cycle, this sentence might need rephrasing

L85 the last sentence in this paragraph ("Competitively-optimal...") does not follow directly from the rest of the paragraph, there seems to be a logic jump. What is competitively-optimal? How does such a model result in allocation strategies?

L99 There is a lot of information packed into this equation which is not appropriately explained. Also I am not sure if this equation is relevant to the rest of the paper.

L111 The turnover of vegetation carbon pools is generally not only driven by mortality but also tissue senescence

L 207 Are the C:N ratios of all pools considered fixed?

L238, eq. 7 It would help here if the first term and the second term in the minimum function were explained in words - I think it is start of growing season available NSC and during growing season available NSC?

L245 I'm not sure I understand why step 1 is needed given eq 6c

L254 Does step 4 here imply that the sapwood has variable C:N? Can this increase indefinitely under N limitation?

L279 Is there a justification for the range of soil N availability?

L355 Generally, I would say 'hump-shaped' is a curve that goes up then down, which is not the case here.

L445 Are there no observational studies showing this behaviour? L482 Are there no

measurements in the literature of fine root C:N ratios?

---

## Referee Comment (RC2) · Anonymous Referee #2 · 8 Mar 2019

This study presents a modeling comparison where a single model was altered with fixed allocation and competition-driven allocation scheme along a nitrogen availability gradient and under ambient and elevated CO2. The competition scheme that the author considered are nutrient availability and light availability. The authors found that competition-driven allocation scheme predicted different fractional allocation to fine-root and wood as compared to fixed-allocation scheme. While the results are generally supported by their study, I do have several issues that I would like to bring to author's

attention.

Major comments:

It appears that the allocation assumptions made in L 254 – 256 are key to their predicted results. In particular, it appears to me that the exact order of step 1 and step 2 may have a profound effect on the competition dynamics. I wonder what will happen if plant prioritize NSC allocation to leaf and root first, and chuck the remaining C to wood next? In the current text, I think the author did not provide sufficient discussion or justification to these potentially fundamental assumptions. Moreover, what happen if the extra C under step 4 is respired rather than allocated to wood? This could potentially match with some existing model treatment with the extra C, which deserves some discussion.

Furthermore, while the results indicate a reversed fractional allocation pattern to fine-root and wood under competition-driven allocation scheme, there is no "data" to actually prove that this new allocation scheme is an improvement to the fixed allocation scheme. Many models already consider "dynamic" allocation based on nutrient availability and water, but the author did not make any comparison against those model behaviors. I'd suggest the authors at least to bridge their modeling results with some observations to make a more convincing argument that their scheme has some advantages.

Moreover, the author highlighted that competition for light and nutrient drives successional dynamics (e.g. L 46, 83, 105-107, etc.), which left me with the impression that successional dynamics is a key component of the paper. But in fact, it surprises me that the authors only included results on successional dynamics in the supplementary materials, and there's little discussion around this topic. I'd suggest tightening up Figure S4 and S5, and move them into the main text, with more thorough discussions around them.

Minor comments:

[Figure]

L 24-26: question: does this mean fixed allocation performs similarly in predicted NPP when compared to those based on competitive-allocation strategy? So the change in allocation pattern does not result in any changes in predicted NPP?

L35-36. It's a bit unclear what the author trying to illustrate here.

L38: "ecosystem-level predictions" of what? You indicated earlier that the predicted NPP was similar, right?

L230. The symbol $\varphi$RL was described here, which appears to be too late. Suggest to define it in its first appearance.

Figure 2. The author showed how competition runs compared differently to the fixed allocation runs, based on $\varphi$RL =4. Since you are talking about succession and competition, it remains unclear what is the community response?

Figure 3. Missing unit on y-axis.

---

## Referee Comment (RC3) · Benjamin Stocker (Referee) · 12 Mar 2019

This paper describes a model and its prediction for competitively optimal allocation (ratio of root to foliage surface area) and how it varies with a range of nitrogen levels and two CO2 levels. The same (or a similar version of the same? See point 12 below) model has been described and applied previously to investigate optimal phenological strategies in Weng et al. (2016, GCB, doi: 10.1111/gcb.13542) and a carbon only

version was presented in Weng et al. (2015, BG, doi:10.5194/bg-12-2655-2015).

The present manuscript addresses allocation as a single variable parameter, although multiple traits affect plant functioning in the face of N availability and $CO_2$ concentrations. However, the focus on allocation is justified, in my opinion, for two reasons: First, allocation warrants particular attention as it is a key process that is known to be responsive to changes in above and belowground resource availabilities and affects the carbon cycling in ecosystems when an allocation shifts occur between long-lived wood and short-lived foliage or fine roots. Second, most vegetation models simulate allocation either based on fixed parameters, or based on empirical relationships. However, as pointed out in the manuscript (l. 528), predicting allocation from first principles is key to realistically and robustly simulating vegetation changes in response to multiple changing environmental factors. The present model embodies a promising way forward to resolve this challenge, determining a competitively optimal allocation strategy, based on height-structured competition for light in the form of a Vegetation Demographics Modelling approach (cohort-based, not average individual-based as is common in Dynamic Vegetation Models). In this respect, the present model takes a pioneering path, that only less than a handful (or even less?) other models can follow.

I see two main weaknesses of the present study. First, predictions are not tested against observational data. What data needs to be used as a test (see comment 10)? However, I don't expect much additional insight from a comparison to observational data at this stage, and consider the theoretical predictions to deserve to be presented as "naked predictions". However, for a paper that deals with just the theoretical side of the problem, some aspects may warrant additional attention (additional figures for results results and extended discussion) in the manuscript (see comments 6, 8, 10, 11). Second, the modelling approach with regards to the excess-C-sapwood allocation (see l.245) raised some questions for me. Is it compatible with our understanding for what controls sapwood area (or what determines the Huber value, defined as the ratio of sapwood area:leaf area)? I worry that this model formulation may cause unrealistic model behaviour in some instances. Anyways, the conclusions need to be drawn carefully with regards to this aspect (see comment 5 below).

In the "SPECIFIC POINTS" described below, I am suggesting some modifications in the description of the model and to improve readability, and some changes in the presentation to distill the most relevant points and most interesting insights from this research. GENERAL POINTS may warrant particular attention. The paper is generally written very well and the presentation of results is clear and clean. If these points can be addressed satisfactorily, I may recommend a revised version of this manuscript for publication in Biogeosciences.

GENERAL POINTS

1. Dynamic adaptation (adaptive plastic responses) of allocation occurs also within species. The present model embodies the assumption that species follow a fixed allocation strategy, and changes in allocation are induced by replacement of species with different allocation strategies. Please add a discussion for the assumption of fixed within-species allocation may affect results.

2. A competitively optimal strategy is determined for stationary boundary conditions. Does this inhibit such a modelling approach to be applicable for global change simulations (transiently changing boundary conditions)? See also comment 11.

3. Allocation and the plant-level C and N budgets, assume fixed tissue C:N ratios and flexible allocation to sapwood to make C and N budgets work. This does not correspond to the known controls on sapwood area and may induce unwanted model behaviour. However, it is difficult to think understand to what degree this affects the results and conclusions. See also comment 5.

4. Total N that is circulating in the system is manipulated for the different simulations, with zero N loss and inputs. This sounds like a rather pragmatic than realistic solution. In reality, losses are never zero, and N levels are manipulated in experiments by

fertilisation. Why is it not implemented like this? Would you expect any systematic differences between your implementation and one with non-zero losses and manipulated inputs?

5. The abstract mentions "opposite fractional allocation to fine roots and wood" in competitive and fixed-allocation runs. Relatively more allocation to fine roots at high N levels in fixed allocation runs sounds like a result that runs counter to the common understanding of the controls on fine root allocation (e.g. Poorter et al., 2012 New Phytologist, doi: 10.1111/j.1469-8137.2011.03952.x), and appears confusing in view of the fact that the model is based on a fixed root:leaf area ratio. I suspect that the increase in relative root allocation at high N levels in the fixed-allocation run is a result of the excess-C-sapwood allocation in this model. See also my comment 3. If this is indeed the case, I would recommend not to present results in the abstract that are contingent on this, arguably unrealistic, model behaviour. I guess the authors don't want to challenge the common understanding of fine root allocation controls with their results. It would suffice to point out that in the competition runs, relative fine root allocation decreases with increasing N levels; and present this in the light of the common modelling approach whereby the root:leaf area ratios (and effective relative allocation ratios) are fixed.

6. Presentation of results for a fixed $\varphi$: In my understanding, the essence of this research is to predict how stand-level relative allocation changes in response to N availability and CO2. The approach to predict it is to derive a competitively optimal allocation strategy at the individual-level. In view of the main aim (essence) of this research, I would expect as a (first) figure something that shows competitively optimal $\varphi$ for each N/CO2 level, derived from the competition runs.

7. A question: Does each point in Fig. 3 show the wood/root/foliage NPP (fraction) for the competitively optimal $\varphi$ at the respective N and CO2 levels (for the competition runs)? Comparing this to values from fixed-allocation runs with a pre-specified $\varphi$ is maybe not the most interesting. This leads to the next point. . .

8. [see Addendum to this comment] In view of my comment 1 (dynamic adaptation of allocation also within species), readers may want to know if an optimality criterion that is defined for some instantaneous individual-level metric (like foliage NPP, or height growth, etc.) leads to the (qualitatively) same predictions as an optimality criterion that is funded in an evolutionarily stable strategy (ESS, like the one applied here). It might be interesting to evaluate the fixed-allocation runs to determine which $\varphi$ maximises some optimality criterion (like foliage NPP, or height growth, etc.). A (first) figure could then compare this individual-level derived optimal $\varphi$ to the ESS-derived optimal $\varphi$. Furthermore, points in Fig. 3 for the fixed-allocation runs could then be taken to represent the $\varphi$ level that maximises the optimality criterion for each N/CO2 level respectively. This would also enable a direct comparison of the the two optimality approaches (ESS vs. instantaneous). I understand if this suggestion is beyond the scope of this paper, or not feasible. Anyways, I would be interested to learn more about such a comparison. Addendum: I see now (upon reading the Discussion, l. 425), that this point is addressed by Supplementary figures S1 and S2. Maybe this is too much of my own personal interest, but I think to generally enhance this point in the presentation of the results (possibly with additional/modified figures as suggested above) would improve the manuscript.

9. A description of how the competitively optimal allocation is determined (description of the algorithm), would be helpful. May be added before current Section 2.2.

10. The differences in predictions based on different allocation schemes are interesting, but the missing comparison to observational data prevents conclusions to be drawn about which is more realistic or leads to better model performance. The question is: What is the key observation that can be used to test predictions? And of course: What is the key prediction that authors want to test? In my view, it is a viable option to remain with theoretical predictions, not actually using data from observations. However, this may require an extended discussion in the light of generally observed patterns with additional references to the literature. The challenge remains that the overproliferation

in root growth predicted by the competitively-optimal allocation scheme may not directly be testable. How much is over-proliferation in reality? What would be a suitable observation to evaluate this prediction?

11. Regarding implications for Earth system modelling: From reading this manuscript, it's not entirely clear whether the approach for determining the competitively-optimal allocation strategy is applicable for typical Earth system simulations, where boundary conditions change transiently. In my understanding, the approach chosen here determines a system steady state, formed by a monospecific stand with a certain allocation strategy, that cannot be invaded by any species with a different allocation strategy. As explained in the manuscript, this requires the model to be run into steady state for 1200 simulation years. How would this be implemented for a typical Earth system simulation setup? I think it would be very informative to complement Section 4.3 with a discussion on this point.

12. The description of how the present model version differs from model versions used in Weng et al., 2016, 2015, could be made clearer.

SPECIFIC POINTS

l.96: Should be a gap in 'trait (s)'?

Eq. 2: To solve the model, ds/dt has to be set to zero, I guess. Shouldn't this be reflected in Eq. 2? Or how exactly is the competitively optimal strategy determined?

l.148: Apart from variations in across-species allocation patterns (e.g., oak species tending to invest more into roots), there are also clear patterns in within species and within-individuals (flexible adaptation) variations in allocation when subjected to shifts in resource availability. In my understanding, such fast allocation responses are not captured by the modelling approach here. This should be clarified. Connects to Comment 1 above.

l.159: The simulation experiments are described in the abstract and intro to be done

along a "nitrogen availability gradient". How did you manipulate N? This is described at a later stage, but could already be made clear here.

Section 2.1: A separate paragraph on how $CO_2$ assimilation is simulated, would be helpful.

Eq. 3: To be consistent with Fig. 1, I would suggest to use the symbol $X\_FR$ as the pool size (or $C\_FR$ in this case), distinguish community- and individual-level variables for example using a bar over the variable for the community-level, and use a separate letter for the parameter 'Root_0' (e.g., $K\_FR$).

Eq. 4: Should be clarified that this is the community-level total root biomass (if I am correct). Clarification is needed to understand Eq. 4.

Eq. 5: D is not defined. Diameter?

l. 215: Add bracket: "... targets for leaf, fine root, and sapwood cross-sectional area ($L*\_k$, $FR*\_k$, and $A*\_SW$)" here for a better overview of the description.

Eq. 5/6: How is D incremented? The way the system is described now, the tree doesn't grow in D or am I missing something?

l. 238: Can you add $f\_1$ and $f\_2$ to the description in this sentence? E.g., "capped by a larger fraction of NSC ($f\_1$)"?

l. 241: Are Eq. 7 and its parameters $f\_1$ and $f\_2$ identical throughout the year? The description here suggests that something is different during leaf flush versus the period of "normal growth". Or maybe I'm just misunderstanding it the way it's formulated now.

l. 246: Units of 0.15? g C?

l. 254: Since sapwood production requires N as well (although relatively less than production of other tissues), and "excess C" sounds like this is the amount of NSC left, after NSN is used up (hence zero), I would assume that some iteration is necessary to perfectly match the use of NSC and NSN in the allocation procedure. How is this

solved? Either more detail should be given here, or the description should be modified to avoid misunderstanding.

l. 260: I would welcome a summarising sentence on the mechanisms determining C:N stoichiometry. The connection between sapwood allocation and the NSC:NSN budget implies that a plant that acquires relatively little N in comparison to assimilated C (in other words: an "N-limited tree") would produce relatively more sapwood. Does this mean that Eq. 6 (the A_SW sub-equation) is "over"-satisfied? What are the implications of this in the model? Does it affect the relationship between height growth vs. crown area expansion?

Section 2.1: A description of how the competitively optimal allocation is determined (description of the algorithm) would be helpful. May be added before current Section 2.2.

l. 292: "Full factorial" suggests that all combinations of treatment factors are applied to force the runs. But here, this is a mix of a treatment factor (N levels) and model parameter (phi). I suggest to rephrase this.

Table 1: If i=(114.5, . . ., 552) g N m-2, then 4.5-0.5*i is a negative number. Is this correct? Maybe N levels in units of kg N m-2 are used here?

l. 362-364: The decrease in fractional allocation to fine roots with elevated CO2 at high N levels is surprising. May it be a result of the excess-C sapwood allocation approach implemented in this model? May warrant a brief discussion of this aspect.

l. 386-388: In my reading, this is a main result and should be shown in a separate figure, shown at the very beginning of the results section.

l. 418-419: See my comment 5.

l.425: See my comment 8. This is an interesting point, but is dealt with rather briefly here. How is "maximising growth rate" implemented exactly? NPP? NPP of a specific pool? "Allocating very little" is vague. The crucial aspect is that for a given N level and

uptake half-saturation constant, the plants allocate much less to fine roots in the best-performing (by what measure?) fixed-allocation run than in the competitively optimal-allocation run.

l. 430-433: Sentence is hard to follow. Is the height at transition into the canopy (reaching critical height) increasing or decreasing with increasing N?

l. 493 ("succession"): Discussing competitively optimal strategy shifts during succession confused me here. I understood, that the the competitively optimal strategy is determined for a *steady-state*, and (based on my understanding from reading previous papers of this group of authors) an ESS is determined from competition upon invasion into a mono-specific stand. But now I realise that the algorithm for determining competitive optimality has never been described in the present paper. A gap that shold be filled (see also comments above).

l. 569: I had a great laugh when I read the short description of that repository on github ("BiomeESS: for simulating multiple plant forms, on-going, unpublished, with ridiculous processes and many bugs.") Maybe the author wants to change that upon publication of this manuscript (and if necessary resolve some known bugs). If not, I appreciate the honesty.

---

## Author Comment (AC1) · 4 May 2019

**Response to Anonymous Referee #1**

Thanks for your comments and suggestions. We will revise this manuscript according to your review comments. Below are our responses and plan for revision. Your comments are in italics, and our responses are in roman.

*The current paper aims to compare the predictions of biomass allocation within a vegetation demographic model (VDM) **with explicit competition versus a model without competition**, under elevated $CO_2$ across a nitrogen availability gradient. To this end, the authors use a derivation of an existing VDM, where the only process that varies is the biomass allocation. The authors then present comparisons between **the two model versions** at equilibrium for one site. The topic of predicting vegetation allocation pattern for different nutrient limitation states is an important one, and one that existing vegetation model often have trouble with. The approach of having one model with two different process representations is also very valuable as it can pinpoint model differences to the exact processes in question.*

*However, the value of this study is largely obscured by the way the model is presented and discussed, making it very difficult for the reader to link between model assumptions, results and model implications.*

Thanks for the comments. We will revise the manuscript following the comments and suggestions of all the three reviewers.

**Major comments**

*It is unclear to me if this analysis actually shows **a difference between a model with and one without competition or simply a difference between a model with fixed and one with flexible allocation**. As the authors themselves point out in the introduction, pool-based vegetation models often incorporate a flexible allocation scheme based on nutrient and water availability. It is currently unclear if a model with such a scheme would perform differently from the competition model included here.*

It is a model with one PFT (no competition) vs. multiple PFTs (with competition). We will clarify this and change the notations of the figures (from "fixed vs. comp." to "single vs. multiple").

*One of the key assumptions of the model **is the order of allocation** (l 245): first a fixed fraction allocated to the sapwood, then allocation to the leaves and roots, then if there is available C and N left, to sapwood and fruit. And, most importantly, any carbon left in excess because of N limitation is allocated to sapwood. This last step could have some interesting implications for light competition under N limitation, and can maybe explain the different wood allocation*

*patterns observed for the competition models. This model assumption needs to be justified and discussed.*

I agree with the reviewer that our description of allocation scheme is confusing, and leads to misunderstanding of the simulation experiments. We used four steps to describe the technical implementation of the ideas of the allocation model in the codes. It failed to convey the major principles of the allocation scheme itself.

The allocation scheme is robust and has been used in many Perfect Plasticity Approximation model-based studies. The carbon assimilated by leaves via photosynthesis enters into the non-structural carbon (NSC) pool first and then is used for respiration, growth, and reproduction. The nitrogen absorbed by roots enters into the non-structural nitrogen (NSN) pool and then is allocated to plant pools (*i.e.*, leaves, fine roots, seeds, and sapwood) following plant growth. The partitioning of carbon and nitrogen into the plant pools is constrained by allometric equations, targets of leaves and fine roots, and the target C:N ratios of these plant pools. The plant growth (and therefore allocation) is simulated at two steps: 1) calculating the amount of carbon and nitrogen that will be used for plant growth at this time step (which can be hourly, daily, weekly, etc.); 2) allocating the available carbon and nitrogen to leaves, fine roots, seeds, and sapwood following rules of first principles. We let the plant growth follow the rules below as they are in the first version (Weng et al. 2015, Biogeosciences. carbon only model) and second version of LM3-PPA (Weng et al. 2017, Global Change Biology. carbon and nitrogen):

1. Plants maintain their leaves and fine roots as close as possible to their targets as defined by allometry equations during the growing season (i.e., leaf and fine roots priority over stems)

2. The ratio of fine root area to leaf area is constant (pipe model) when there is no disturbance to abruptly change leaves and fine roots.

3. Plants must keep some carbon storage (i.e., NSC) for respiration (*i.e.*, they don't kill themselves by using up their NSC for tissue growth).

4. C:N ratios of plant tissues must be close to their target C:N ratios, though they can have daily variations due to numerical issues in matching carbon and nitrogen allocation in daily growth.

5. Plants are able to use available carbon and nitrogen in the most efficient way from the perspective of competition.

   Following these rules, our model numerically calculates the amount of carbon and nitrogen that are available for growth (i.e., building new tissues) at a daily time step. Basically, the available NSC ($G_C$) is the summation of a small fraction ($f_2$) of the total NSC in an individual plant and the differences between the targets of leaf and fine roots and their current biomass capped by a larger fraction ($f_1$) of NSC (Eq. 1.1). The available NSN ($G_N$) is analogous to that of the NSC and meets approximately the stoichiometrical requirement of plant tissues (Eq. 1.2).

$$G_C = \min (f_1 NSC, f_2 NSC + L^* + FR^* - L - FR) \qquad \text{(Eq. 1.1)}$$

$$G_N = \min (f_3 NSN, f_4 NSN + N_L^* + N_{FR}^* - N_L - N_{FR}) \qquad \text{(Eq. 1.2)}$$

where $L^*$ and $FR^*$ are the targets of leaves and fine roots, respectively; $L$ and $FR$ are current leaf and fine roots biomass, respectively; $N_L^*$ and $N_{FR}^*$ are nitrogen of leaves and fine roots at their targets according to their default C:N ratios. The parameter $f_1$ gives the daily availability of NSC during periods of leaf flush at the beginning of a growing season and $f_2$ for normal growth of stems after plant leaves and fine roots approach their targets. Usually, parameter $f_1$ is much larger than $f_2$. We let $f_1$=0.05 and $f_2$= 1/(365×3) in this study. The parameter $f_2$ is used to keep a certain amount of NSC. Likewise, the parameters $f_3$ and $f_4$ are defined the same way as $f_1$ and $f_2$. We let $f_3$= $f_1$ and $f_4$= $f_2$ for convenience in this study. Compared to carbon availability, nitrogen availability is relatively stable because SOM cannot vary wildly from day to day (thought temperature can). Plants thus always have a stable supply of nitrogen from these SOM. The mean nitrogen available for growth ($G_N$) is actually equal to mean daily mineralization rate. The parameters $f_3$ and $f_4$ are only used to smooth nitrogen supply and get proper seasonal patterns.

The allocation of the available NSC ($G_C$) to wood ($G_W$), leaves ($G_L$), fine roots ($G_{FR}$), and seeds ($G_F$) is following the equations below (Eqs. 2). This allocation scheme coordinates the supply of carbon and nitrogen for growth by adjusting the allocation between high-nitrogen tissues and low-nitrogen tissues to maximize leaves and fine roots growth ($G_L$ and $G_{FR}$, respectively) to maximize nitrogen usage at given nitrogen supply (i.e., $G_N$) and keep the tissues at their default C:N ratios.

$$G_C = G_W + G_L + G_{FR} + G_F \qquad \text{(Eq. 2.1)}$$

$$G_L + G_{FR} = Min \begin{bmatrix} [L^* + FR^* - L - FR], \\ (1 - f_{W,min}) \, G_C \end{bmatrix} \cdot r_{D/S} \qquad \text{(Eq. 2.2)}$$

$$G_F = v[G_C(t) \cdot r_{D/S} - (G_L + G_{FR})] \qquad \text{(Eq. 2.3)}$$

$$G_W = (1 - v \cdot r_{D/S})G_C - (1 - v)(G_L + G_{FR}) \qquad \text{(Eq. 2.4)}$$

$$\frac{(FR + G_{FR})SRA}{(L + G_L)/LMA} = \varphi_{RL} \qquad \text{(Eq. 2.5)}$$

$$\frac{G_L}{CN_L} + \frac{G_{FR}}{CN_{FR}} + \frac{G_F}{CN_F} + \frac{G_W}{CN_W} \leq G_N \qquad \text{(Eq. 2.6)}$$

$$L + G_L \leq L^* \qquad \text{(Eq. 2.7)}$$

$$FR + G_{FR} \leq FR^* \qquad \text{(Eq. 2.8)}$$

where, $r_{D/S}$ is a nitrogen-limited allocation factor to be solved numerically each step; $f_{W,min}$ is the minimum fraction of $G_C$ for stems (0.15 in this study); $v$ is the fraction of carbon for seeds (0.1 in

this study); $CN_L$, $CN_{FR}$, $CN_F$, and $CN_W$ are the default C:N ratios of leaves, fine roots, seeds, and wood (including sapwood), respectively. Parameter $r_{D/S}$ ranges from 0 (highest nitrogen limitation; no nitrogen for leaves, fine roots, and seeds at this step) to 1 (nitrogen is sufficient for all tissues).

The allocation scheme itself is flexible and can generate variable allocation patterns even with one fixed scheme of allocation (i.e., fixed $\varphi_{RL}$) at different environments because of variable $r_{D/S}$. The key step in solving this set of equations is to solve $r_{D/S}$ in each growth step (daily in this model). This parameter changes with relative nitrogen availability. When there is no nitrogen limitation, $r_{D/S}$ equals to 1 and the allocation follows the conditions defined by Eqs. 2.1~2.5 (carbon only model). When $r_{D/S}$ equals to 0, $G_N$ does not meet the nitrogen demand even if all the $G_C$ is allocated to wood tissues and we have to return the excessive carbon to the NSC pool (this is a very rare case because of low carbon input long before $r_{D/S}$ approaches to 0 due to nitrogen limitation, though our codes must be able to deal with all possible cases.). When $r_{D/S}$ is in between ($0 < r_{D/S} < 1$), the leaves and fine roots cannot reach to their targets after this step of growth (i.e., plants maintain a low LAI in low nitrogen environments). At low nitrogen availability, the parameter $r_{D/S}$ keeps a low LAI, a relatively constant NPP/GPP ratio, and a stable NSC for each single PFT. The low LAI reduces carbon supply, and therefore reduces nitrogen demand for plant growth, making $r_{D/S}$ larger than zero.

Overall, this is a flexible allocation scheme and still follows the major assumptions in the previous versions of LM3-PPA (Weng, et al., 2015, 2017). It prioritizes the allocation to leaves and fine roots, maintains a minimum growth rate of stems, keeps the constant area ratio of fine roots to leaves, and optimizes resource usage by relocating carbon and nitrogen to wood tissues when nitrogen is not sufficient for full growth of leaves and fine roots. In normal growth, for each time step, leaves and fine roots get ($L^* + FR^* - L - FR$), stems $(1-v)f_2$NSC, and seeds $vf_2$NSC. In the early days of a growing season when leaves are much lower than its target, leaves and fine roots get a large portion of $G_C$ (maximum is 0.85 in this study, 1- $f_{W,min}$). When nitrogen is limited, leaves and fine roots are lower than their targets, reducing photosynthesis and carbon supply. So, this allocation scheme will not result in over-growth of stems because of the reduced leaves at nitrogen limitation.

Based on these allocation rules, the mean of allocations of carbon and nitrogen to leaves, fine roots, and wood over a growing season are governed by the targets for the leaf area per unit crown area (i.e., crown leaf area index, $l^*$) and fine root area per unit leaf area ($\varphi_{RL}$). Since the crown leaf area index, $l^*$, is fixed in this study, $\varphi_{RL}$ is the key parameter determining the relative allocation of carbon to fine roots and stems. A high $\varphi_{RL}$ means a high relative allocation to fine roots and therefore low relative allocation to stems, and *vice versa*.

The parameter $f_{W,min}$ is for maintaining a relatively stable growth rate of tree trunk in the highly variable years. Since allocation ratios to stems are around 0.4~0.7 in temperate forests, with a value of 0.15, it does not affect the overall allocation ratios of carbon among leaves, fine roots, and stems, and still keep trunk grow in bad years, though at a very low rate. If we let $f_{W,min}$

= 0 (i.e., completely leaf growth priority), trees would have unreasonably high variation of trunk growth because leaf and fine root growth would use all carbon for approaching to their targets and leave nothing for stems in bad years.

The parameter $f_2$ represents another strategy of growth: *conservative* vs. *progressive*. At a small $f_2$, trees keep a large NSC pool for bad years (conservative); at a big $f_2$, trees grow fast and take the whole site at a short period of time, but risks starvation in bad years. Since we don't explore this strategy in this study, we let it be constant for all PFTs.

We will add a section in discussion to justify these assumptions.

*While I understand that this is a theoretical study, and such studies are valuable and note **every study needs to show a comparison with data**, **there is a complete lack of model reality checking**. Especially when the two model versions show contrasting al- location patterns, there should be a way to determine what the reality is. There is a wealth of data from FACE experiments, N addition experiments, long-term measurements, soil N gradients etc. I believe it would be very interesting to have a section in the discussion comparing the current model predictions with general observed trends.*

We agree with this comment. We will synthesize the data from FACE and nitrogen fertilization experiments and add a section for reality checking of the model and discussing our simulation results.

*As it is, the discussion mostly contains comparisons with previous models from the same model family, which while I think is probably relevant to the authors for model development purposes, is of little interest to the general audience.*

We will add a section in discussion to compare with other model predictions.

**Detailed comments**

*L 52 I'm not sure there are any ESM's that just simulate the nitrogen cycle, this sentence might need rephrasing*

We will rephrase this sentence to "that simulate ecosystem biogeochemical cycles as lumped pools and fluxes …".

*L85 the last sentence in this paragraph ("Competitively-optimal...") does not follow directly from the rest of the paragraph, there seems to be a logic jump. What is competitively-optimal? How does such a model result in allocation strategies?*

We will add a paragraph to explain "competitively optimal" and allocation strategies.

*L99 There is a lot of information packed into this equation which is not appropriately explained. Also I am not sure if this equation is relevant to the rest of the paper.*

We will remove this equation and add a figure to show the idea of three levels of model processes in this model.

[Figure]

*L111 The turnover of vegetation carbon pools is generally not only driven by mortality but also tissue senescence*

We will rephrase this sentence to include senescence of leaves and turnover of fine roots.

*L 207 Are the C:N ratios of all pools considered fixed?*

Only leaves and fine roots are strictly fixed. Wood C:N can be variable in numerically solving the allocation patterns for convenience, but the allocation scheme makes it only variate in a very small range.

*L238, eq. 7 It would help here if the first term and the second term in the minimum function were explained in words - I think it is start of growing season available NSC and during growing season available NSC?*

We explained it within the new description of the allocation scheme (Eq. 1 in the response to allocation description).

*L245 I'm not sure I understand why step 1 is needed given eq 6c*

We explained it within the new description of the allocation scheme. It is a minimum growth ($f_{W,min}$). In Eq. 6, we only define the target sapwood cross sectional area.

*L254 Does step 4 here imply that the sapwood has variable C:N? Can this increase indefinitely under N limitation?*

Yes, sapwood has variable C:N ratio during the numerical iteration. However, it does not lead to indefinite increase in wood C:N because of reduced GPP and relatively stable supply of mineral nitrogen.

*L279 Is there a justification for the range of soil N availability?*

AU: We set these levels according to Harvard Forest soil nitrogen content. We will update the manuscript with description of soil nitrogen and references.

*L355 Generally, I would say 'hump-shaped' is a curve that goes up then down, which is not the case here.*

We will rephrase this sentence.

*L445 Are there no observational studies showing this behaviour?*

This study is experimental. We will look for other empirical studies.

*L482 Are there no measurements in the literature of fine root C:N ratios?*

There are many measurements of root C:N. However, here, we are talking about its ESS responses to ecosystem nitrogen. We will rephrase this sentence and clarify it.

---

## Author Comment (AC2) · 4 May 2019

**Response to Anonymous Referee #2**

Thanks for your comments and suggestions. We will revise this paper according to your review comments. Below are our responses and plan for revision. Your comments are in italics, and our responses are in roman.

*This study presents a modeling comparison where a single model was altered with fixed allocation and competition-driven allocation scheme along a nitrogen availability gradient and under ambient and elevated $CO_2$. The competition scheme that the author considered are nutrient availability and light availability. The authors found that competition-driven allocation scheme predicted different fractional allocation to fine root and wood as compared to fixed-allocation scheme. While the results are generally supported by their study, I do have several issues that I would like to bring to author's attention.*

I realized the terms "fixed allocation scheme" is really confusing from reviewers' comments. It's a "fixed scheme" of allocation, while "allocation" is flexible. We will change to "single vs. multiple" in the revised version.

**Major comments:**

*It appears that the allocation assumptions made in L 254 – 256 are key to their predicted results. In particular, it appears to me that the exact order of **step 1 and step 2** may have a profound effect on the competition dynamics. I wonder what will happen if plant prioritize NSC allocation to leaf and root first, and chuck the remaining C to wood next? In the current text, I think the author did not provide sufficient discussion or justification to these potentially fundamental assumptions. Moreover, what happen if the extra C under step 4 is respired rather than allocated to wood? This could potentially match with some existing model treatment with the extra C, which deserves some discussion.*

Our description of allocation scheme is confusing, and leads to misunderstanding of the simulation experiments. We used four steps to describe the technical implementation of the ideas of the allocation model in the codes. However, it turns out this description failed to convey the major principles of the allocation scheme in our model.

In our model, the carbon assimilated by leaves via photosynthesis enters into the non-structural carbon (NSC) pool first and then is used for respiration, growth, and reproduction. The nitrogen absorbed by roots enters into the non-structural nitrogen (NSN) pool and then is allocated to plant pools (*i.e.*, leaves, fine roots, seeds, and sapwood) following plant growth. The partitioning of carbon and nitrogen into the plant pools is constrained by allometric equations, targets of leaves and fine roots, and the C:N ratios of these plant pools. The plant growth (and therefore allocation) is simulated at two steps: 1) calculating the amount of carbon and nitrogen that will be used for plant growth at this time step (which can be hourly, daily, weekly, etc.); 2) allocating the available carbon and nitrogen to leaves, fine roots, seeds, and sapwood following rules of first principles. This allocation scheme is robust and has been used in many Perfect Plasticity Approximation model-based studies. We let the plant growth follow the rules below as they are in the first version (Weng et al. 2015, Biogeoscieces. carbon only model) and second version of LM3-PPA (Weng et al. 2017, Global Change Biology. carbon and nitrogen):

1. Plants maintain their leaves and fine roots as close as possible to their targets as defined by allometry equations during the growing season (i.e., leaf and fine roots priority over stems)

2. The ratio of fine root area to leaf area is constant (pipe model) when there is no disturbance to abruptly change leaves and fine roots.

3. Plants must keep some carbon storage (i.e., NSC) for respiration (i.e., they don't suicide by using up their NSC in tissue growth).

4. C:N ratios of plant tissues must be close to their target C:N ratios, though they can have daily variations due to numerical issues in matching carbon and nitrogen allocation in daily growth.

5. Plants are able to use available carbon and nitrogen in the most efficient way from the perspective of competition.

Following these rules, our model numerically calculates the amount of carbon and nitrogen that are available for growth (i.e., building new tissues) at a daily time step. Basically, the available NSC ($G_C$) is the summation of a small fraction ($f_2$) of the total NSC in an individual plant and the differences between the targets of leaf and fine roots and their current biomass capped by a larger fraction ($f_1$) of NSC (Eq. 1.1). The available NSN ($G_N$) is analogous to that of the NSC and meets approximately the stoichiometrical requirement of plant tissues (Eq. 1.2).

$$G_C = \min\left(f_1 NSC, f_2 NSC + L^* + FR^* - L - FR\right) \qquad \text{(Eq. 1.1)}$$

$$G_N = \min\left(f_3 NSN, f_4 NSN + N_L^* + N_{FR}^* - N_L - N_{FR}\right) \qquad \text{(Eq. 1.2)}$$

where $L^*$ and $FR^*$ are the targets of leaves and fine roots, respectively; $L$ and $FR$ are current leaf and fine roots biomass, respectively; $N_L^*$ and $N_{FR}^*$ are nitrogen of leaves and fine roots at their targets according to their default C:N ratios. The parameter $f_1$ gives the daily availability of NSC during periods of leaf flush at the beginning of a growing season and $f_2$ for normal growth of stems after plant leaves and fine roots approach their targets. Usually, parameter $f_1$ is much larger than $f_2$. We let $f_1$=0.05 and $f_2$= 1/(365x3) in this study. The parameter $f_2$ is used to keep a certain amount of NSC. Likewise, the parameters $f_3$ and $f_4$ are defined the same way as $f_1$ and $f_2$. We let $f_3$=$f_1$ and $f_4$=$f_2$ for convenience in this study. Compared to carbon availability, nitrogen availability is relatively stable because SOM cannot vary wildly from day to day. Plants thus always have a stable supply of nitrogen from these SOM. The mean nitrogen available for growth ($G_N$) is actually equal to mean daily mineralization rate. The parameters $f_3$ and $f_4$ are only used to smooth nitrogen supply.

The allocation of the available NSC ($G_C$) to wood ($G_W$), leaves ($G_L$), fine roots ($G_{FR}$), and seeds ($G_F$) is following the equations below (Eqs. 2). This allocation scheme coordinates the supply of carbon and nitrogen for growth by adjusting the allocation between high-nitrogen tissues and low-nitrogen tissues to maximize leaves and fine roots growth ($G_L$ and $G_{FR}$, respectively) to maximize nitrogen usage at given nitrogen supply (i.e., $G_N$) and keep the tissues at their default C:N ratios.

$$G_C = G_W + G_L + G_{FR} + G_F \qquad \text{(Eq. 2.1)}$$

$$G_L + G_{FR} = Min \begin{bmatrix} [L^* + FR^* - L - FR], \\ (1 - f_{W,min}) \, G_C \end{bmatrix} \cdot r_{D/S} \qquad \text{(Eq. 2.2)}$$

$$G_F = v[G_C(t) \cdot r_{D/S} - (G_L + G_{FR})] \qquad \text{(Eq. 2.3)}$$

$$G_W = (1 - v \cdot r_{D/S})G_C - (1 - v)(G_L + G_{FR}) \qquad \text{(Eq. 2.4)}$$

$$\frac{(FR + G_{FR})SRA}{(L + G_L)/LMA} = \varphi_{RL} \qquad \text{(Eq. 2.5)}$$

$$\frac{G_L}{CN_L} + \frac{G_{FR}}{CN_{FR}} + \frac{G_F}{CN_F} + \frac{G_W}{CN_W} \leq G_N \qquad \text{(Eq. 2.6)}$$

$$L + G_L \leq L^* \qquad \text{(Eq. 2.7)}$$

$$FR + G_{FR} \leq FR^* \qquad \text{(Eq. 2.8)}$$

where, $r_{D/S}$ is a nitrogen-limited allocation factor to be solved numerically each step; $f_{W,min}$ is the minimum fraction of $G_C$ for stems (0.15 in this study); $v$ is the fraction of carbon for seeds (0.1 in this study); $CN_L$, $CN_{FR}$, $CN_F$, and $CN_W$ are the default C:N ratios of leaves, fine roots, seeds, and wood (including sapwood), respectively. Parameter $r_{D/S}$ ranges from 0 (highest nitrogen limitation; no nitrogen for leaves, fine roots, and seeds at this step) to 1 (nitrogen is sufficient for all tissues).

The allocation scheme itself is flexible and can change with environment even with one fixed scheme of allocation (i.e., fixed $\varphi_{RL}$) because of variable $r_{D/S}$. The key step in solving this set of equations is to solve $r_{D/S}$ in each growth step (daily in this model). This parameter changes with relative nitrogen availability. When there is no nitrogen limitation, $r_{D/S}$ equals to 1 and the allocation follows the conditions defined by Eqs. 2.1~2.5 (carbon only model). When $r_{D/S}$ equals to 0, $G_N$ does not meet the nitrogen demand even if all the $G_C$ is allocated to wood tissues and we have to return the excessive carbon to the NSC pool (this is a very rare case because of low carbon input long before $r_{D/S}$ approaches to 0 due to nitrogen limitation, though our codes must be able to deal with all possible cases.). When $r_{D/S}$ is in between ($0 < r_{D/S} < 1$), the leaves and fine roots cannot reach to their targets after this step of growth (i.e., plants maintain a low LAI in low nitrogen environments). At low nitrogen availability, the parameter $r_{D/S}$ keeps a low LAI, a relatively constant NPP/GPP ratio, and a stable NSC for each single PFT. The low LAI reduces carbon supply, and therefore reduces nitrogen demand for plant growth, making $r_{D/S}$ larger than zero.

Overall, this is a flexible allocation scheme and still follows the major assumptions in the previous versions of LM3-PPA (Weng, et al., 2015, 2017). It prioritizes the allocation to leaves and fine roots, maintains a minimum growth rate of stems, keeps the constant area ratio of fine roots to leaves, and optimizes resource usage by relocating carbon and nitrogen to wood tissues when nitrogen is not sufficient for full growth of leaves and fine roots. In normal growth, for each time step, leaves and fine roots get $(L^* + FR^* - L - FR)$, stems $(1-v)f_2$NSC, and seeds $vf_2$NSC. In the early days of a growing season when leaves are much lower than its target, leaves and fine roots get a large portion of $G_C$ (maximum is 0.85 in this study, $1 - f_{W,min}$). When nitrogen is limited, leaves and fine roots are lower than their targets, reducing photosynthesis and carbon

supply. So, this allocation scheme will not result in over-growth of stems because of the reduced leaves at nitrogen limitation.

Based on these allocation rules, the mean of allocations of carbon and nitrogen to leaves, fine roots, and wood over a growing season are governed by the targets for the leaf area per unit crown area (i.e., crown leaf area index, $l^*$) and fine root area per unit leaf area ($\varphi_{RL}$). Since the crown leaf area index, $l^*$, is fixed in this study, $\varphi_{RL}$ is the key parameter determining the relative allocation of carbon to fine roots and stems. A high $\varphi_{RL}$ means a high relative allocation to fine roots and therefore low relative allocation to stems, and *vice versa*.

The parameter $f_{W,min}$ quantifies the priority of leaves and fine roots allocation. Since allocation ratios to stems are around 0.4~0.7 in temperate forests, with a value of 0.15, it does not affect the overall allocation ratios of carbon among leaves, fine roots, and stems, and still keep trunk grow in bad years, though at a very low rate. A small $f_{W,min}$ can maintain a relatively stable growth rate of tree trunk. If we let $f_{W,min} = 0$ (i.e., completely leaf-fine root growth priority), trees would have unreasonably high inter-annual variation of trunk growth because leaf and fine root growth would use all carbon for approaching to their targets and leave nothing for stems in bad years.

*Furthermore, while the results indicate a reversed fractional allocation pattern to fine- root and wood under competition-driven allocation scheme, there is no "data" to actually prove that this new allocation scheme is an improvement to the fixed allocation scheme. Many models already consider "dynamic" allocation based on nutrient avail- ability and water, but the author did not make any comparison against those model behaviors. I'd suggest the authors at least to bridge their modeling results with some observations to make a more convincing argument that their scheme has some advantages.*

We will synthesize the data from FACE experiments and compare our results with other models.

*Moreover, the author highlighted that competition for light and nutrient drives successional dynamics (e.g. L 46, 83, 105-107, etc.), which left me with the impression that successional dynamics is a key component of the paper. But in fact, it surprises me that the authors only included results on successional dynamics in the supplementary materials, and there's little discussion around this topic. I'd suggest tightening up Figure S4 and S5, and move them into the main text, with more thorough discussions around them.*

We will add a succession pattern figure (below) into the paper.

[Figure]

**Minor comments:**

*L 24-26: question: does this mean fixed allocation performs similarly in predicted NPP when compared to those based on competitive-allocation strategy? So the change in allocation pattern does not result in any changes in predicted NPP?*

Yes, though NPP changes a little bit. Changes in allocation pattern mainly affect the relative allocation between stems and fine roots. Leaves are similar at the same nitrogen level.

*L35-36. It's a bit unclear what the author trying to illustrate here.*

We will rephrase this sentence.

*L38: "ecosystem-level predictions" of what? You indicated earlier that the predicted NPP was similar, right?*

Yes, it's ecosystem carbon dynamics. We will clarify it.

*L230. The symbol $\varphi_{RL}$ was described here, which appears to be too late. Suggest to define it in its first appearance.*

We have moved it to the equation where it is used first time and reorganized this section.

*Figure 2. The author showed how competition runs compared differently to the fixed allocation runs, based on $\varphi_{RL}$ =4. Since you are talking about succession and competition, it remains unclear what is the community response?*

We will add the successional figures (above in response to major comments).

*Figure 3. Missing unit on y-axis.*

We will replace it with the new figure below:

[Figure]

---

## Author Comment (AC3) · 4 May 2019

Thanks for your comments and suggestions. We will revise this paper according to your review comments. Below are our responses and plan for revision. Your comments are in italics, and our responses are in roman.

*This paper describes a model and its prediction for competitively optimal allocation (ratio of root to foliage surface area) and how it varies with a range of nitrogen levels and two CO2 levels. The same (or a similar version of the same? See point 12 below) model has been described and applied previously to investigate optimal phenological strategies in Weng et al. (2016, GCB, doi: 10.1111/gcb.13542) and a carbon only version was presented in Weng et al. (2015, BG, doi:10.5194/bg-12-2655-2015).*

Thanks for the summary.

*The present manuscript addresses allocation as a single variable parameter, although multiple traits affect plant functioning in the face of N availability and CO2 concentrations. However, the focus on allocation is justified, in my opinion, for two reasons: First, allocation warrants particular attention as it is a key process that is known to be responsive to changes in above and belowground resource availabilities and affects the carbon cycling in ecosystems when an allocation shifts occur between long-lived wood and short-lived foliage or fine roots. Second, most vegetation models simulate allocation either based on fixed parameters, or based on empirical relationships. However, as pointed out in the manuscript (l. 528), predicting allocation from first principles is key to realistically and robustly simulating vegetation changes in response to multiple changing environmental factors. The present model embodies a promising way forward to resolve this challenge, determining a competitively optimal allocation strategy, based on height-structured competition for light in the form of a Vegetation Demographics Modelling approach (cohort-based, not average individual-based as is common in Dynamic Vegetation Models). In this respect, the present model takes a pioneering path, that only less than a handful (or even less?) other models can follow.*

Thanks for the comments.

*I see two main weaknesses of the present study.* **First, predictions are not tested against observational data**. *What data needs to be used as a test (see comment 10)? However, I don't expect much additional insight from a comparison to observational data at this stage, and consider the theoretical predictions to deserve to be presented as "naked predictions". However, for a paper that deals with just the theoretical side of the problem, some aspects may warrant additional attention (additional figures for results and extended discussion) in the manuscript (see comments 6, 8, 10, 11).* **Second, the modelling approach with regards to the excess-C-sapwood allocation** *(see l.245) raised some questions for me. Is it compatible with our understanding for what controls sapwood area (or what determines the Huber value, defined as the ratio of sapwood area: leaf area)? I worry that this model formulation may cause unrealistic model behaviour in some instances. Anyways, the conclusions need to be drawn carefully with regards to this aspect (see comment 5 below).*

I agree with Dr. Stocker. We will update the description of allocation scheme (Please see responses to referees 1 and 2) and discuss the comparison of our simulations with observations and simulations of other models if invited to submit a revised version.

Specifically for the Huber value (sapwood area/leaf area) here, it is kept constant for a PFT, following Eq. 6 in the main text, where Huber value is a PFT-specific parameter ($\alpha_{CSA}$). Since the allometry equations are correct (Eq. 2 in the main text) and the model doesn't have processes of plant hydraulics, Huber value is just used to differentiate sapwood and heartwood and does not affect the functions of the trees. Please see a revised description of this equation below:

"We set *targets* for leaf ($L^*$), fine root ($FR^*$), and sapwood cross-sectional area ($A_{SW}^*$) that govern plant allocation of non-structural carbon and nitrogen during growth. These *targets* are related by the following equations based on the assumption of the pipe model (Shinozaki, Kichiro et al., 1964):

$$L^*(D,p) = l^* \cdot A_{CR}(D) \cdot LMA \cdot p(t)$$
$$FR^*(D) = \varphi_{RL} \cdot l^* \cdot \frac{A_{CR}(D)}{SRA} \qquad\qquad \text{(Eq. 1)}$$
$$A_{SW}^*(D) = \alpha_{CSA} \cdot l^* \cdot A_{CR}(D)$$

where $L^*$ (D, p), $FR^*(D)$, and $A_{SW}^*(D)$ are the targets of leaf mass (kg C/tree), fine root biomass (kg C/tree), and sapwood cross sectional area (m²/tree), respectively, at tree diameter *D*; $l^*$ is the target leaf area per unit crown area of a given PFT; $A_{CR}$(D) is the crown area of a tree with diameter *D*; LMA is PFT-specific leaf mass per unit area; and *p(t)* is a PFT-specific function ranging from zero to one that governs leaf phenology (Weng et al., 2015); $\varphi_{RL}$ is the target ratio of total root surface area to the total leaf area; SRA is specific root area; and $\alpha_{CSA}$ is an empirical constant (the ratio of sapwood cross-sectional area to target leaf area). The phenology function *p*(t) takes values 0 (non-growing season) or 1 (growing season) following the phenology model of LM3-PPA (Weng et al., 2015). The onset of a growing season is controlled by two variables, growing degree days (GDD), and a weighted mean daily temperature ($T_{pheno}$), while the end of a growing season is controlled by $T_{pheno}$."

The "excess-C-sapwood allocation" is a numerical step to adjust the growth of leaves, fine roots, wood, and seeds according to available carbon and nitrogen. The model uses these steps to numerically solve the following growth/allocation equations at given carbon and nitrogen:

$$G_C = G_W + G_L + G_{FR} + G_F \qquad\qquad \text{(Eq. 2.1)}$$

$$G_L + G_{FR} = Min\begin{bmatrix} [L^* + FR^* - L - FR], \\ (1 - f_{W,min})\, G_C \end{bmatrix} \cdot r_{D/S} \qquad\qquad \text{(Eq. 2.2)}$$

$$G_F = v[G_C(t) \cdot r_{D/S} - (G_L + G_{FR})] \qquad\qquad \text{(Eq. 2.3)}$$

$$G_W = (1 - v \cdot r_{D/S})G_C - (1 - v)(G_L + G_{FR}) \qquad\qquad \text{(Eq. 2.4)}$$

$$\frac{(FR+G_{FR})SRA}{(L+G_L)/LMA} = \varphi_{RL} \tag{Eq. 2.5}$$

$$\frac{G_L}{CN_L} + \frac{G_{FR}}{CN_{FR}} + \frac{G_F}{CN_F} + \frac{G_W}{CN_W} \leq G_N \tag{Eq. 2.6}$$

$$L + G_L \leq L^* \tag{Eq. 2.7}$$

$$FR + G_{FR} \leq FR^* \tag{Eq. 2.8}$$

where, $r_{D/S}$ is a nitrogen-limited allocation factor to be solved in this set of equations; $f_{W,min}$ is the minimum fraction of $G_C$ for stems (0.15 in this study); $v$ is the fraction of carbon for seeds (0.1 in this study); $CN_L$, $CN_{FR}$, $CN_F$, and $CN_W$ are the default C:N ratios of leaves, fine roots, seeds, and sapwood, respectively. Parameter $r_{D/S}$ ranges from 0 (Highest nitrogen limitation; no nitrogen for leaves, fine roots, and seeds at this step) to 1 (nitrogen is sufficient for all tissues).

*In the "SPECIFIC POINTS" described below, I am suggesting some modifications in the description of the model and to improve readability, and some changes in the presentation to distill the most relevant points and most interesting insights from this research. GENERAL POINTS may warrant particular attention. The paper is generally written very well and the presentation of results is clear and clean. If these points can be addressed satisfactorily, I may recommend a revised version of this manuscript for publication in Biogeosciences.*

*GENERAL POINTS*

*1. Dynamic adaptation (adaptive plastic responses) of allocation occurs also within species. The present model embodies the assumption that species follow a fixed allocation strategy, and changes in allocation are induced by replacement of species with different allocation strategies.* **Please add a discussion for the assumption of fixed within-species allocation may affect results.**

Our allocation scheme (Eq. 2) itself is flexible and can change with environment even with one fixed scheme of allocation (i.e., fixed $\varphi_{RL}$) by numerically maximizing carbon usage at nitrogen limitation. The key step in solving this set of equations is to numerically solve $r_{D/S}$ in each growth step (daily in this simulator). The parameter $r_{D/S}$ changes with relative nitrogen availability. When there is no nitrogen limitation, $r_{D/S}$ equals to 1 and the allocation follows the conditions defined by Eqs. 2.1~2.5 (carbon only model). When $r_{D/S}$ equals to 0, $G_N$ does not meet the nitrogen demand even if all the $G_C$ is allocated to wood tissues and the model has to return the excessive carbon to the NSC pool (this is a very rare case because of low carbon input long before $r_{D/S}$ approaches to 0 due to nitrogen limitation. However, our codes must be able to deal with all possible cases.). When $r_{D/S}$ is in between (0< $r_{D/S}$<1), the leaves and fine roots cannot reach to their targets after this step of growth (i.e., plants maintain a low LAI in low nitrogen environments). At low nitrogen availability, the parameter $r_{D/S}$ keeps a low LAI, a relatively constant NPP/GPP ratio, and a relatively stable NSC for each single PFT.

We will add a paragraph to discuss this allocation scheme following this suggestion.

*2. A competitively optimal strategy is determined for stationary boundary conditions. Does this inhibit such a modelling approach to be applicable for global change simulations (transiently changing boundary conditions)? See also comment 11.*

This study is not for directly application at global scale, but the succession patterns can be used to understand transient responses of vegetation to climate changes and the model formulations are being incorporated into an Earth system modeling framework (NASA GISS ModelE). Our analysis can help to understand model behavior at global scales.

*3. Allocation and the plant-level C and N budgets, assume fixed tissue C:N ratios and flexible allocation to sapwood to make C and N budgets work. **This does not correspond to the known controls on sapwood area and may induce unwanted model behaviour.** However, it is difficult to think understand to what degree this affects the results and conclusions. See also comment 5.*

The allocation between fine roots plus leaves and wood does not affect the differentiation between sapwood and heartwood. We have a detailed description of the conversion from sapwood to heartwood in the appendix of Weng et al. 2015. The new tissues of wood are always counted as sapwood, and the oldest tissues are converted to heartwood. Actually, the differentiation between sapwood and heartwood does not affect model function in this model because we don't have plant hydraulics yet in this model. We will update allocation scheme to make it easier to understand.

*4. Total N that is circulating in the system is manipulated for the different simulations, with zero N loss and inputs. This sounds like a rather pragmatic than realistic solution. In reality, losses are never zero, and N levels are manipulated in experiments by fertilisation. Why is it not implemented like this? Would you expect any systematic differences between your implementation and one with non-zero losses and manipulated inputs?*

The processes of nitrogen inputs and losses are complex, involving nitrogen deposition, fixation, soil nitrogen mineralization, root uptake efficiency, microbial immobilization, leaching, denitrification, etc.. However, at equilibrium state, the total ecosystem nitrogen is determined by relative rates of nitrogen input and nitrogen output. For example, a high nitrogen input and low output lead to high total ecosystem nitrogen level. In this study, we pack all these effects by setting different total N levels to represent implicitly relative nitrogen input and loss rates. For example, a high total ecosystem nitrogen level represents the ecosystems and edaphic conditions with relative high nitrogen input and low loss rates. By packing these processes into different total N levels, we can focus our study on allocation.

We did both nitrogen closed (with zero nitrogen input and loss) and nitrogen open (with non-zero nitrogen input and output) runs in a previous study (Weng et al. 2017, Global Change Biology). In both types of runs, the nitrogen level determines the competition of PFTs and the competitively optimal player since plants can only "feel" the nitrogen availability, and don't "care" how this availability is set (e.g., either by setting the total N or through complex input and output processes).

We will add a discussion about the meaning of the different nitrogen levels if invited for submitting a revised version.

*5. The abstract mentions "opposite fractional allocation to fine roots and wood" in competitive and fixed-allocation runs.* ***Relatively more allocation to fine roots at high N levels in fixed allocation runs sounds like a result that runs counter to the common understanding of the controls on fine root allocation*** *(e.g. Poorter et al., 2012 New Phytologist, doi: 10.1111/j.1469-8137.2011.03952.x), and appears confusing in view of the fact that the model is based on a fixed root:leaf area ratio.* ***I suspect that the increase in relative root allocation at high N levels in the fixed-allocation run is a result of the excess-C-sapwood allocation in this model.*** *See also my comment 3. If this is indeed the case, I would recommend not to present results in the abstract that are contingent on this, arguably unrealistic, model behaviour. I guess the authors don't want to challenge the common understanding of fine root allocation controls with their results. It would suffice to point out that in the competition runs, relative fine root allocation de- creases with increasing N levels; and present this in the light of the common modelling approach whereby the root:leaf area ratios (and effective relative allocation ratios) are fixed.*

Thanks for the suggestion. High allocation to fine roots is resulted from "**excess-C-sapwood allocation**" in the basal allocation setting (numerical solution of parameter $r_{D/S}$). The model solves two parameters in these model runs: $r_{D/S}$ (at each daily step for both single PFT runs and multi PFTs runs) and $\varphi_{RL}$ (after more than a thousand years model runs with multiple PFTs).

Our results are consistent with Poorter et al. 2012 well (Fig. 1:f of Poorter et al. 2012), because their data should be compared with our multi-PFT, long-term simulation results (i.e., competitively optimal strategy in our Fig. 3). For a single PFT (fixed strategy and short-term responses to nitrogen addition), it should be compared with fertilization experiments (e.g., Lee & Jose 2003, Forest Ecology & Management).

[Figure]

Nutrient availability (rel. scale)

leaves (LMF; red line), stems (SMF; brown line) and roots (RMF; blue line)

We will revise the abstract according to reviewer's suggestions, the data analysis of Poorter et al. 2012 and other nitrogen fertilization experiments.

*6. Presentation of results for a fixed φ: In my understanding, the essence of this research is to predict how stand-level relative allocation changes in response to N availability and CO2. The approach to predict it is to derive a competitively optimal allocation strategy at the individual-level. In view of the main aim (essence) of this research, I would expect as a (first) figure something that shows competitively optimal φ for each N/CO2 level, derived from the competition runs.*

Thanks for the suggestion. We will add a new figure to show competitively optimal $\varphi$ for each N and $CO_2$ level.

*7. A question: Does each point in Fig. 3 show the wood/root/foliage NPP (fraction) for the competitively optimal φ at the respective N and CO2 levels (for the competition runs)? Comparing this to values from fixed-allocation runs with a pre-specified φ is maybe not the most interesting. This leads to the next point. . .*

The left is absolute, and right is fraction. Please see the figure with unit:

[Figure]

*8. [see Addendum to this comment] In view of my comment 1 (dynamic adaptation of allocation also within species), readers may want to know **if an optimality criterion that is defined for some instantaneous individual-level metric (like foliage NPP, or height growth, etc.) leads to the (qualitatively) same predictions as an optimality criterion that is funded in an evolutionarily stable strategy (ESS, like the one applied here)**. It might be interesting to evaluate the fixed-allocation runs to determine which φ maximises some optimality criterion (like foliage NPP, or height growth, etc.). A (first) figure could then compare this individual-level derived optimal φ to the ESS-derived optimal φ. Furthermore, points in Fig. 3 for the fixed-allocation runs could then be taken to represent the φ level that maximises the optimality criterion for each N/CO2 level respectively. This would also enable a direct comparison of the two optimality approaches (ESS vs. instantaneous). I understand if this suggestion is beyond the scope of this paper, or not feasible. Anyways, I would be interested to learn more about such a comparison. Addendum: I see now (upon reading the Discussion, l. 425), that this point is*

*addressed by Supplementary figures S1 and S2. Maybe this is too much of my own personal interest, but I think to generally enhance this point in the presentation of the results (possibly with additional/modified figures as suggested above) would improve the manuscript.*

I will explain the differences between optimal vs. competitively optimal if invited to submit a revised version. Basically, optimal strategy is not necessarily the competitively optimal. From Figures 1 and 2, you can see that low root allocation is optimal in many N levels. However, they cannot outcompete high root allocation strategies if invaded by them. That means, in the environment created by the optimal strategy individuals, another strategy can out compete the resident, though it may have low NPP or fitness in monoculture. I will add a figure in supplementary materials to show this.

*9. A description of how the competitively optimal allocation is determined (description of the algorithm), would be helpful. May be added before current Section 2.2.*

I will add a description following this suggestion.

*10. The differences in predictions based on different allocation schemes are interesting, but the missing comparison to observational data prevents conclusions to be drawn about which is more realistic or leads to better model performance. The question is: What is the key observation that can be used to test predictions? And of course: What is the key prediction that authors want to test? In my view, it is a viable option to remain with theoretical predictions, not actually using data from observations. However, this may require an extended discussion in the light of generally observed patterns with additional references to the literature. The challenge remains that the overproliferation in root growth predicted by the competitively-optimal allocation scheme may not directly be testable. How much is over-proliferation in reality? What would be a suitable observation to evaluate this prediction?*

I will add a paragraph of discussion to discuss the value of this study in experimental perspectives.

*11. Regarding implications for Earth system modelling: From reading this manuscript, it's not entirely clear whether the approach for determining the competitively-optimal allocation strategy is applicable for typical Earth system simulations, where boundary conditions change transiently. In my understanding, the approach chosen here determines a system steady state, formed by a monospecific stand with a certain allocation strategy, that cannot be invaded by any species with a different allocation strategy. As explained in the manuscript, this requires the model to be run into steady state for 1200 simulation years. How would this be implemented for a typical Earth system simulation setup? I think it would be very informative to complement Section 4.3 with a discussion on this point.*

It is helpful for understanding model behavior and track competitively optimal strategy.

We are exploring one more layer of complexity, adaptive dynamics (or successional dynamics), on the top of the pool-flux dynamics and demographic dynamics. For a robust DGVM in Earth system models, it is necessary to analyze these processes and their effects on model performance in detail. We will discuss it in detail if invited for submitting a revised version.

*12. The description of how the present model version differs from model versions used in Weng et al., 2016, 2015, could be made clearer.*

I will  put more technical details of this model into a supplementary material.

**SPECIFIC POINTS**
*l.96: Should be a gap in 'trait (s)'?*

Done as suggested. But, we will remove this equation following Reviewer 1's suggestion.

*Eq. 2: To solve the model, ds/dt has to be set to zero, I guess. Shouldn't this be reflected in Eq. 2? Or how exactly is the competitively optimal strategy determined?*

This equation is used to show the temporal dynamics of a plant trait in evolution, which, conceptually, is the numerical shifts of a plant trait approaching its ESS in simulations.

For analytically solve this equation, one needs to build up a fitness equation, such as:

$$Fitness(s_I|s_R) = Benefit(s_I|s_R) - Cost_{I,R}(s_I|s_R) - Cost_I(s_I)$$

where, $s_I$ is invader's trait and $s_R$ is residence. Let the derivative of the right side to be zero and according to the definition of ESS $s_I = s_R$. Then, it is possible to solve it analytically if you are lucky.

We will remove this equation following Reviewer 1's suggestion. We agree with the Reviewer 1 that this equation is not very helpful here. We will add a paragraph to explain the processes of approaching ESS by succession in VDMs. The numerical simulations are complex enough, we don't want to complicate the paper with more discussions of this equation.

*l.148: Apart from variations in across-species allocation patterns (e.g., oak species tending to invest more into roots), there are also clear patterns in within species and within-individuals (flexible adaptation) variations in allocation when subjected to shifts in resource availability. In my understanding, such fast allocation responses are not captured by the modelling approach here. This should be clarified. Connects to Comment 1 above.*

They have to follow the same rules, otherwise they will be replaced by those who follow the rules. Actually, we used only one PFT, which only differs in fine root – leaf area ratio ($\varphi$). If plants have enough plasticity in allocation, they will approach the ESS much quicker. We will make it clear in a revised manuscript.

*l.159: The simulation experiments are described in the abstract and intro to be done along a "nitrogen availability gradient". How did you manipulate N? This is described at a later stage, but could already be made clear here.*

We will clarify the setting of ecosystem N.

*Section 2.1: A separate paragraph on how CO2 assimilation is simulated, would be helpful.*

We will add a brief description of the photosynthesis model, which is the same as it in LM3-PPA and LM3.

*Eq. 3: To be consistent with Fig. 1, I would suggest to use the symbol X_FR as the pool size (or C_FR in this case), distinguish community- and individual-level variables for example using a bar over the variable for the community-level, and use a separate letter for the parameter 'Root_0' (e.g., K_FR).*

Done as suggested.

*Eq. 4: Should be clarified that this is the community-level total root biomass (if I am correct). Clarification is needed to understand Eq. 4.*

Clarified. $C_{FR}$, total is the total roots in soil, with unit of kgC/m2 and CFR,tree is the total root carbon of a tree (in a cohort with identical trees) with unit of kgC/tree. This equation transforms nitrogen uptake rate from gN m-2 hour-1 to gN tree-1 hour-1.

*Eq. 5: D is not defined. Diameter?*

Yes, it is diameter. defined.

*l. 215: Add bracket: ". . . targets for leaf, fine root, and sapwood cross-sectional area (L*_k, FR*_k, and A*_SW)" here for a better overview of the description.*

Done as suggested.

*Eq. 5/6: How is D incremented? The way the system is described now, the tree doesn't grow in D or am I missing something?*

We add a new equation to show diameter growth.

l. 238: Can you add f_1 and f_2 to the description in this sentence? E.g., "capped by a larger fraction of NSC (f_1)"?

done as suggested.

*l. 241: Are Eq. 7 and its parameters f_1 and f_2 identical throughout the year? The description here suggests that something is different during leaf flush versus the period of "normal growth". Or maybe I'm just misunderstanding it the way it's formulated now.*

They are constant over time. Generally, "$f_1$NSC" defines the maximum NSC availability at the beginning of a growing season when $(L+FR)<<(L^*+FR^*)$; "$f_2$NSC" defines the normal growth of stems after $(L+FR)$ is close to $(L^*+FR^*)$ (i.e., normal growth during the growing season;

"[(L*+FR*) -(L+FR)]" are the carbon for the new leaves and fine roots due to stem growth in last time step.

The term "$f_1$NSC" also prevents overspending of NSC when NSC is very low in some bad years.

*l. 246: Units of 0.15? g C?*

It's the fraction of NSC. Clarified.

*l. 254: Since sapwood production requires N as well (although relatively less than production of other tissues), and "excess C" sounds like this is the amount of NSC left, after NSN is used up (hence zero), I would assume that some iteration is necessary to perfectly match the use of NSC and NSN in the allocation procedure. How is this solved? Either more detail should be given here, or the description should be modified to avoid misunderstanding.*

It can be analytically solved. In the codes, we numerically solve it by iterating at daily step and allowing the wood C:N ratio to variate from day to day.

We will clarify it in a revised version.

*l. 260: I would welcome a summarising sentence on the mechanisms determining C:N stoichiometry. The connection between sapwood allocation and the NSC:NSN budget implies that a plant that acquires relatively little N in comparison to assimilated C (in other words: an "N-limited tree") would produce relatively more sapwood. Does this mean that Eq. 6 (the A_SW sub-equation) is "over"-satisfied? What are the implications of this in the model? Does it affect the relationship between height growth vs. crown area expansion?*

According to the allometry equation (Eq. 5 in the main text), tree height and crown area are functions of diameter. So, how to define sapwood does not affect anything in the current version of the model. The $A_{SW}$ sub-equation Eq. 6 is used to separate sapwood from the whole trunk. As mentioned in the responses to General comment 2, it does not affect anything of the model. We did this because the model needs this variable. (We really need it when incorporating plant hydraulics.).

We will update the whole section with a new description of the model's allocation scheme. The partitioning of carbon and nitrogen into the plant pools is constrained by allometric equations, targets of leaves and fine roots, and the C:N ratios of these plant pools. The plant growth (and therefore allocation) is simulated at two steps: 1) calculating the amount of carbon and nitrogen that will be used for plant growth at this time step (which can be hourly, daily, weekly, etc.); 2) allocating the available carbon and nitrogen to leaves, fine roots, seeds, and sapwood following rules of first principles. We let the plant growth follow the rules below as they are in the first version (Weng et al. 2015, Biogeoscieces. carbon only model) and second version of LM3-PPA (Weng et al. 2017, Global Change Biology. carbon and nitrogen). Overall, this is a flexible allocation scheme and still follows the major assumptions in the previous versions of LM3-PPA (Weng, et al., 2015, 2017). It prioritizes the allocation to leaves and fine roots, maintains a minimum growth rate of stems at nitrogen limitation, keeps the constant area ratio of fine roots to leaves, and optimizes resource usage by relocating carbon and nitrogen to wood tissues when

nitrogen is not sufficient for full growth of leaves and fine roots. When nitrogen is limited, leaves and fine roots are lower than their targets, reducing photosynthesis and carbon supply.

*Section 2.1: A description of how the competitively optimal allocation is determined (description of the algorithm) would be helpful. May be added before current Section 2.2.*

We will add a description of "competitively optimal" in Section 2.2.

*l. 292: "Full factorial" suggests that all combinations of treatment factors are applied to force the runs. But here, this is a mix of a treatment factor (N levels) and model parameter (phi). I suggest to rephrase this.*

We will rephrase it as suggested.

*Table 1: If i=(114.5, . . ., 552) g N m-2, then 4.5-0.5\*i is a negative number. Is this correct? Maybe N levels in units of kg N m-2 are used here?*

Here, *i* takes the value of 1, 2, 3, 4, …, 8, following the order of the nitrogen levels from 114.5 to 552 gN m$^{-2}$. We will clarify it.

*l. 362-364: The decrease in fractional allocation to fine roots with elevated CO2 at high N levels is surprising. May it be a result of the excess-C sapwood allocation approach implemented in this model? May warrant a brief discussion of this aspect.*

For competition runs, it's because of the competition (competitive φ).

*l. **386-388: In my reading, this is a main result and should be shown in** a separate figure, shown at the very beginning of the results section.*

We will move  this paragraph in 390~406 to the beginning of the results section and the figure S1 to the main text as Figure 2.

*l. 418-419: See my comment 5.*

We will rephrase this sentence.

*l.425: See my comment 8. This is an interesting point, but is dealt with rather briefly here. How is "maximising growth rate" implemented exactly? NPP? NPP of a specific pool? "Allocating very little" is vague. The crucial aspect is that for a given N level and uptake half-saturation constant, the plants allocate much less to fine roots in the best- performing (by what measure?) fixed-allocation run than in the competitively optimal- allocation run.*

We will rephrase this sentence and make it clear. The measure is the height growth in a competitive environment.

*l. 430-433: Sentence is hard to follow. Is the height at transition into the canopy (reaching critical height) increasing or decreasing with increasing N?*

We will rephrase this sentence: "Changes in the height at which understory trees transition to the canopy from low nitrogen to high nitrogen indicate a shift from the importance of competition for soil nitrogen to the importance of competition for light as ecosystem nitrogen increases." to:

"Increases in the critical height, which is the height of the shortest tree in canopy layer, from low nitrogen to high nitrogen indicates a shift from the importance of competition for soil nitrogen to the importance of competition for light as ecosystem nitrogen increases."

*l. 493 ("succession"): Discussing competitively optimal strategy shifts during succession confused me here. I understood, that the competitively optimal strategy is determined for a \*steady-state\*, and (based on my understanding from reading previous papers of this group of authors) an ESS is determined from competition upon invasion into a mono-specific stand. But now* **I realise that the algorithm for determining competitive optimality has never been described in the present paper. A gap that should be filled (see also comments above).**

We will add a brief description of "competitively optimal strategy" in model description.

*l. 569: I had a great laugh when I read the short description of that repository on github ("BiomeESS: for simulating multiple plant forms, on-going, unpublished, with ridiculous processes and many bugs.") Maybe the author wants to change that upon publication of this manuscript (and if necessary resolve some known bugs). If not, I appreciate the honesty.*

Thanks for taking a look! We have updated the codes and will update the description upon accept of this paper.

---

## Author Response (AR1)

Dear Dr. Zaehle,

We thank you and the reviewers for their comments and suggestions. These comments are very helpful for improving the presentation of our study. We have re-run the model after fixing a problem that leads to high wood C:N ratio (350 ~ 800) in the codes, and made a thorough revision of the paper.

We mainly did the following:

1. We added a new description of the allocation scheme of our model with analytical solution of allocation at each step.

2. We fixed a numerical error in the codes that leads to high wood C:N ratio and updated all the figures and results. The major patterns in new model runs are very similar with those in previous version.

3. We added discussion about the reality of simulated results in comparison with data from meta-analysis and FACE experiments. We also discussed emerging common patterns of allocation modeling in comparison with other models and implications of our research for ecosystem modeling and Earth system modeling studies.

4. We compiled a detailed description of our model and included it as Supplementary Information I. It includes photosynthesis model, respiration, demographic processes (reproduction, growth, and mortality), population dynamics, phenology, and soil biogeochemical processes. Most of them have been published in previous papers about LM3-PPA (Weng et al. 2015, Weng et al. 2017). We hope it make the readers of this paper easier to get the details of this model.

5. We also made a series of sensitivity runs with different parameters of soil nitrogen mineralization and alternative assumptions of extra carbon allocation at nitrogen limitation (suggested by editor) and found the simulated patterns are robust. We only included these results in this response letter.

Please see the detailed responses to reviewers' comments and revisions enclosed in this letter. Review comments are in italics and our responses are in normal.

Best,

Ensheng Weng on behalf of coauthors Ray Dybzinski, Caroline E. Farrior, and Stephen W. Pacala

**Response to Referee #1**

*The current paper aims to compare the predictions of biomass allocation within a vegetation demographic model (VDM) **with explicit competition versus a model without competition**, under elevated CO$_2$ across a nitrogen availability gradient. To this end, the authors use a derivation of an existing VDM, where the only process that varies is the biomass allocation. The authors then present comparisons between **the two model versions** at equilibrium for one site. The topic of predicting vegetation allocation pattern for different nutrient limitation states is an important one, and one that existing vegetation model often have trouble with. The approach of having one model with two different process representations is also very valuable as it can pinpoint model differences to the exact processes in question.*

*However, the value of this study is largely obscured by the way the model is presented and discussed, making it very difficult for the reader to link between model assumptions, results and model implications.*

Thanks for the comments. We have revised the manuscript following the comments and suggestions of all the three reviewers.

**Major comments**

*It is unclear to me if this analysis actually shows **a difference between a model with and one without competition or simply a difference between a model with fixed and one with flexible allocation**. As the authors themselves point out in the introduction, pool-based vegetation models often incorporate a flexible allocation scheme based on nutrient and water availability. It is currently unclear if a model with such a scheme would perform differently from the competition model included here.*

It is a model with monoculture PFT (no competition) vs. polyculture PFTs (with competition). We have clarified the model runs and changed the notations of the figures (from "fixed vs. comp." to "mono. vs. multiple").

*One of the key assumptions of the model **is the order of allocation** (l 245): first a fixed fraction allocated to the sapwood, then allocation to the leaves and roots, then if there is available C and N left, to sapwood and fruit. And, most importantly, any carbon left in excess because of N limitation is allocated to sapwood. This last step could have some interesting implications for light competition under N limitation, and can maybe explain the different wood allocation patterns observed for the competition models. This model assumption needs to be justified and discussed.*

We agree that our description of allocation scheme is confusing, and leads to misunderstanding of the simulation experiments. We used four steps to describe the technical implementation of the ideas of the allocation model in the codes. However, we failed to convey the major principles of the allocation scheme itself.

The allocation scheme is robust and has been used in many Perfect Plasticity Approximation model-based studies. The carbon assimilated by leaves via photosynthesis enters into the non-structural carbon (NSC) pool first and then is used for respiration, growth, and reproduction. The nitrogen absorbed by roots enters into the non-structural nitrogen (NSN) pool and then is allocated to plant pools (*i.e.*, leaves, fine roots, seeds, and sapwood) following plant growth. The partitioning of carbon and nitrogen into the plant pools is constrained by allometric equations, targets of leaves and fine roots, and the target C:N ratios of these plant pools. The plant growth (and therefore allocation) is simulated at two steps: 1) calculating the amount of carbon and nitrogen that can be used for plant growth at this time step (daily in this study); 2) allocating the available carbon and nitrogen to leaves, fine roots, seeds, and sapwood following a couple of rules based on allometric scaling and functional relationship. We let the plant growth follow the rules below as they are in the first version (Weng et al. 2015, Biogeosciences. carbon only model) and second version of LM3-PPA (Weng et al. 2017, Global Change Biology. Carbon-nitrogen model):

1. Plants maintain their leaves and fine roots as close as possible to their targets as defined by allometry equations during the growing season.

2. The ratio of fine root area to leaf area is constant (cf. pipe model) when there is no disturbance to abruptly change leaves and fine roots.

3. Plants must keep a certain level of carbon storage (i.e., NSC) for respiration (*i.e.*, they don't kill themselves by using up their NSC for tissue growth) and external risks.

4. C:N ratios of plant tissues must be close to their target C:N ratios, though they can have daily variations due to numerical issues in matching carbon and nitrogen allocation in daily growth.

5. Plants are able to use available carbon and nitrogen in the most efficient way from the perspective of competition.

We updated the whole section with a new description of the model (**Allocation and plant growth**).

[revised manuscript text omitted]
, and optimizes resource usage by maximizing leaf and fine root growth when nitrogen is abundant and increasing allocation to wood tissues when nitrogen is limited. When nitrogen is limited, plants allocate a larger portion of NPP to stems and thus lower down leaves and fine roots, reducing photosynthesis and carbon supply."

And, we also discussed this allocation scheme from its first principles to competition mechanisms (Section **4.1 Modeling of allocation and competition and their effects on model predictions**)

"In our model, the allocation of carbon and nitrogen within an individual tree is based on allometric scaling, functional relationships, and optimization of resource usage. Basically, the allometric scaling relationships define the maximum leaf and fine root growth at a given tree size and the functional relationships (pipe model) define the ratios of leaf area to sapwood cross-sectional area and fine root surface area. These rules are commonly used in ecosystem models (Franklin et al., 2012) and have been shown to generate reasonable predictions (De Kauwe et al., 2014; Valentine and Mäkelä, 2012). Overall, these rules lead to the priority of allocation to leaves and fine roots but allow for structurally-unlimited stem growth when resources (carbon and nitrogen in this study) are available (i.e., the remainder goes to stems after leaf and fine root growth).

We define a maximum leaf and fine root allocation, $f_{LFR,max}$, to limit the maximum allocation to leaves and fine roots to maintain a relatively stable growth rate of wood in years of low productivity. The simulated wood growth patterns agree with real wood growth in temperate trees (Cuny et al., 2012; Michelot et al., 2012). Trees need to grow new wood tissues continuously (especially early in the growing season) to maintain their functions (Plomion et al., 2001). This parameter does not change the fact that leaves and fine roots are the priority. Since allocation ratios to stems are around 0.4~0.7 in temperate forests (Curtis et al., 2002; Litton et al., 2007), with a value of 0.85, $f_{LFR,max}$ only seldom affects the overall carbon allocation ratios of leaves, fine roots, and stems, and still maintains wood grow in years of low productivity. If $f_{LFR,max} = 1$ (i.e., the highest priority for leaf and fine root growth), simulated trunk radial growth would have unreasonably high interannual variation because leaf and fine root growth would use all carbon to approach to their targets, leaving nothing for stems in some years of low productivity."

*While I understand that this is a theoretical study, and such studies are valuable and note **every study needs to show a comparison with data, there is a complete lack of model reality checking**. Especially when the two model versions show contrasting allocation patterns, there should be a way to determine what the reality is. There is a wealth of data from FACE experiments, N addition experiments, long-term measurements, soil N gradients etc. I believe it would be very interesting to have a section in the discussion comparing the current model predictions with general observed trends.*

We agree with this comment. We have added three paragraphs in Discussion for reality checking of the model and discussing our simulation results.

"Our competitively-optimal predictions are generally consistent with observations of forest ecosystem production and allocation. For example, high nitrogen environments (i.e., productive environments) favor high wood allocation and low root allocation (Litton et al., 2007; Poorter et al., 2012) because the woody tissues are an unlimited sink for surplus carbon. Low nitrogen availability limits plant $CO_2$ responses (Norby et al. 2010) in the competition runs (polyculture) because of high root allocation. Our model predicts increased root allocation at all nitrogen levels in response to elevated [$CO_2$] in the competition runs. Data from free air $CO_2$

enhancement (FACE) forest experiments largely agree (Drake et al., 2011; Iversen et al., 2012; Jackson et al., 2009; Lukac et al., 2003; Nie et al., 2013; Pritchard et al., 2008; Smith et al., 2013). However, in ORNL-FACE, the increases in root production due to elevated $CO_2$ increase and then declined after 8 years of $CO_2$ enhancement (Iversen, 2010; Norby and Zak, 2011). Though there are no direct data available for quantitatively validating the patterns predicted by our model, especially for the long-term, competitive runs, a detailed modeling analysis can help to understand the varied patterns in the experiments and shed light on the modeling of allocation. ”

*As it is, the discussion mostly contains comparisons with previous models from the same model family, which while I think is probably relevant to the authors for model development purposes, is of little interest to the general audience.*

We have summarized the universal rules of this modeling approach and added a paragraph in discussion to compare with other model predictions.

“In our model, the allocation of carbon and nitrogen within an individual tree is based on allometric scaling, functional relationships, and optimization of resource usage. Basically, the allometric scaling relationships define the maximum leaf and fine root growth at a given tree size and the functional relationships (pipe model) define the ratios of leaf area to sapwood cross-sectional area and fine root surface area. These rules are commonly used in ecosystem models (Franklin et al., 2012) and have been shown to generate reasonable predictions (De Kauwe et al., 2014; Valentine and Mäkelä, 2012). Overall, these rules lead to the priority of allocation to leaves and fine roots but allow for structurally-unlimited stem growth when resources (carbon and nitrogen in this study) are available (i.e., the remainder goes to stems after leaf and fine root growth).”

We also discussed the emerging common principles for modeling allocation:

“As shown in model inter-comparison studies, the mechanisms of modeling allocation differ very much, leading to high variation in their predictions (e.g., De Kauwe et al. 2014). Calibrating model parameters to fit data may not increase model predictive skill because data are often also highly variable. Franklin et al. (2012) suggest that in order to build realistic and predictive allocation models, we should correctly identify and implement fundamental principles. Our model predicts similar patterns to those of Valentine and Mäkelä (2012), which are very different in their details but share fundamental principles, including 1) evolutionary- or competitive-optimization, 2) capped leaves and fine roots, 3) structurally unlimited stem allocation (i.e., for optimizing carbon use), and 4) height-structure competition for light and root-mass-based competition for soil resources. The principles 2 and 3 are commonly used in models (De Kauwe et al., 2014; Jiang et al., 2019). However, the different rules of implementing them (e.g., allometric equation, functional relationships, etc. ) lead to highly varied predictions (as shown in De Kauwe et al., 2014), though the formulations may be very similar. In competitivelyoptimal models, such as this study and also Valentine and Mäkelä (2012), the competition processes generate similar emergent patterns by selecting those that can survive in competition, regardless the details of those differences."

**Detailed comments**

*L 52 I'm not sure there are any ESM's that just simulate the nitrogen cycle, this sentence might need rephrasing*

We rephrased this sentence to "that simulate ecosystem biogeochemical cycles as lumped pools and fluxes …".

*L85 the last sentence in this paragraph ("Competitively-optimal...") does not follow directly from the rest of the paragraph, there seems to be a logic jump. What is competitively-optimal? How does such a model result in allocation strategies?*

We reorganized these three paragraphs and add a couple of sentences (in a paragraph) to explain "competitively optimal strategy".

" To predict transient changes in vegetation structure and composition in response to climate change, vegetation demographic models (VDMs) that are able to simulate transient population dynamics are incorporated into ESMs (Fisher et al., 2018; Scheiter and Higgins, 2009). Generally, VDMs explicitly simulate demographic processes, such as plant reproduction, growth, and mortality, to generate the dynamics of populations (Fig. 1: B). To speed computations and minimize complexity, groups of individuals are usually modeled as cohorts. With multiple cohorts and PFTs, VDMs can bring plant functional diversity and adaptive dynamics into ESMs when explicitly simulating individual-based competition for different resources and vegetation succession and thus predict dominant plant traits changes with environmental conditions and ecosystem development (Scheiter et al., 2013; Scheiter and Higgins, 2009; Weng et al., 2015).

The combinations of plant traits represent the competition strategies at different stages of ecosystem development. Evolutionarily, a strategy that can outcompete all other strategies in the environment created by itself will be dominant. This strategy is called an evolutionarily stable strategy or a competitively-optimal strategy (McGill and Brown, 2007). In VDMs, competitively-optimal strategies can therefore be reasonably predicted based on the costs and benefits of different strategies (i.e., combinations of plant traits) through their effects on demographic processes (i.e., fitness) and ecosystem biogeochemical cycles (Fig. 1:C) (e.g., Farrior et al., 2015; Weng et al., 2015)."

*L99 There is a lot of information packed into this equation which is not appropriately explained. Also I am not sure if this equation is relevant to the rest of the paper.*

We removed this equation and added a figure (Fig. 1) to show the three levels of model processes in VDMs.

[Figure]

*L111 The turnover of vegetation carbon pools is generally not only driven by mortality but also tissue senescence*

We rephrased this sentence to include senescence of leaves and turnover of fine roots.

"In addition, the turnover of vegetation carbon pools becomes a function of allocation, leaf longevity, fine root turnover, and tree mortality rates, which change with vegetation succession and the most competitive plant traits."

*L 207 Are the C:N ratios of all pools considered fixed?*

Only leaves and fine roots are strictly fixed. Wood C:N can be variable in numerically solving the allocation patterns for convenience, but the allocation scheme makes it only variates in a very small range.

However, we had a logical bug in our previous codes, making wood C/N ratio not constrained (see the figure below).

[Figure]

We fixed that problem and re-ran all the tests, and updated all the figures with new simulations. For the wood C:/N ratio, see the figure below:

[Figure]

*L238, eq. 7 It would help here if the first term and the second term in the minimum function were explained in words - I think it is start of growing season available NSC and during growing season available NSC?*

We have re-written this section and in revised manuscript, it is Eq. 6:

"the available NSC ($G_C$) is the summation of a small fraction ($f_1$) of the total NSC in an individual plant and the differences between the targets of leaf and fine roots and their current biomass capped by a larger fraction ($f_2$) of NSC (Eq. 6.1). The available NSN ($G_N$) is analogous to that of the NSC and meets approximately the stoichiometrical requirement of plant tissues (Eq. 6.2).

$$G_C = \min \left( f_1 NSC + L^* + FR^* - L - FR, f_2 NSC \right) \qquad \text{(Eq. 6.1)}$$

$$G_N = \min \left( f_1 NSN + N_L^* + N_{FR}^* - N_L - N_{FR}, f_2 NSN, \right) \qquad \text{(Eq. 6.2)}$$

where $L^*$ and $FR^*$ are the targets of leaves and fine roots, respectively (see Eq. 3); $L$ and $FR$ are current leaf and fine roots biomass, respectively; $N_L^*$ and $N_{FR}^*$ are nitrogen of leaves and fine roots at their targets according to their target C:N ratios. The parameter $f_2$ gives the daily availability of NSC during periods of leaf flush at the beginning of a growing season and $f_1$ normal growth of stems after plant leaves and fine roots approach their targets. Usually, parameter $f_1$ is much greater than $f_2$. We let $f_1$=0.02 and $f_2$= 1/(365x3) in this study."

*L245 I'm not sure I understand why step 1 is needed given eq 6c*

We have re-written this section and we have a parameter $f_{LFR,max}$ in the equations for partitioning available carbon and nitrogen into new tissues (i.e., allocation, Eq. 7). Step 1 means 1-$f_{LFR,max}$, where $f_{LFR,max}$ is the maximum fraction of available carbon used for leaf and fine root growth. We added a paragraph in Discussion to explain parameter $f_{LFR,max}$:

"We define a maximum leaf and fine root allocation, $f_{LFR,max}$, to limit the maximum allocation to leaves and fine roots to maintain a relatively stable growth rate of wood in years of low productivity. The simulated wood growth patterns agree with real wood growth in temperate trees (Cuny et al., 2012; Michelot et al., 2012). Trees need to grow new wood tissues continuously (especially early in the growing season) to maintain their functions (Plomion et al., 2001). This parameter does not change the fact that leaves and fine roots are the priority. Since allocation ratios to stems are around 0.4~0.7 in temperate forests (Curtis et al., 2002; Litton et al., 2007), with a value of 0.85, $f_{LFR,max}$ only seldom affects the overall carbon allocation ratios of leaves, fine roots, and stems, and still maintains wood grow in years of low productivity. If $f_{LFR,max}$ = 1 (i.e., the highest priority for leaf and fine root growth), simulated trunk radial growth would have unreasonably high interannual variation because leaf and fine root growth would use all carbon to approach to their targets, leaving nothing for stems in some years of low productivity. "

*L254 Does step 4 here imply that the sapwood has variable C:N? Can this increase indefinitely under N limitation?*

Yes, sapwood has variable C:N ratio during the numerical iteration. However, it does not lead to indefinite increase in wood C:N because of reduced GPP and relatively stable supply of mineral nitrogen. However, we had a bug in the old codes, making the high equilibrium wood C:N ratio close to 900 (target is 350). We fixed it. Actually, the allocation equations can be analytically solved (please see Eqs. 8 and 9 in the main text, and also copied in response to the major comments ).

*L279 Is there a justification for the range of soil N availability?*

We set this range according to the soil nitrogen content of Harvard Forest from Compton and Boone, 2000. We have updated the manuscript with description of soil nitrogen and references.

"In forest sites, soil carbon is around 8 kgC m$^{-2}$ and nitrogen 300 gN m$^{-2}$ (Compton and Boone, 2000). "

*L355 Generally, I would say 'hump-shaped' is a curve that goes up then down, which is not the case here.*

We rephrased this sentence as "Fine root NPP does not significantly change with total nitrogen in polyculture runs".

*L445 Are there no observational studies showing this behaviour?*

This study (Dybzinski et al. 2019) is experimental. We added another observational study from Oyewole et al., 2017.

*L482 Are there no measurements in the literature of fine root C:N ratios?*

There are many measurements of root C:N. However, here, we were trying to talk about its ESS responses to ecosystem nitrogen. We removed this sentence in the revised manuscript since it is not necessary.

**Response to Referee #2**

*This study presents a modeling comparison where a single model was altered with fixed allocation and competition-driven allocation scheme along a nitrogen availability gradient and under ambient and elevated $CO_2$. The competition scheme that the author considered are nutrient availability and light availability. The authors found that competition-driven allocation scheme predicted different fractional allocation to fine root and wood as compared to fixed-allocation scheme. While the results are generally supported by their study, I do have several issues that I would like to bring to author's attention.*

The terms "fixed allocation scheme" is really confusing. It's a "fixed scheme" of allocation, while "allocation" is flexible. We will change to "mono vs. poly" in the revised version.

**Major comments:**

*It appears that the allocation assumptions made in L 254 – 256 are key to their predicted results. In particular, it appears to me that the exact order of **step 1 and step 2** may have a profound effect on the competition dynamics. I wonder what will happen if plant prioritize NSC allocation to leaf and root first, and chuck the remaining C to wood next? In the current text, I think the author did not provide sufficient discussion or justification to these potentially fundamental assumptions. Moreover, what happen if the extra C under step 4 is respired rather than allocated to wood? This could potentially match with some existing model treatment with the extra C, which deserves some discussion.*

We replace the whole section with a new description from the perspective of mathematics. It is in the section of "**Allocation and plant growth**" and copied in the response to the major comments of reviewer #1. We don't copy the whole section here for saving space.

*Furthermore, while the results indicate a reversed fractional allocation pattern to fine- root and wood under competition-driven allocation scheme, there is no "data" to actually prove that this new allocation scheme is an improvement to the fixed allocation scheme. Many models already consider "dynamic" allocation based on nutrient avail- ability and water, but the author did not make any comparison against those model behaviors. I'd suggest the authors at least to bridge their modeling results with some observations to make a more convincing argument that their scheme has some advantages.*

We added two paragraphs to discuss the reality of our simulations and bridge our modelling approach to modeling community.

"Our competitively-optimal predictions are generally consistent with observations of forest ecosystem production and allocation. For example, high nitrogen environments (i.e., productive environments) favor high wood allocation and low root allocation (Litton et al., 2007; Poorter et al., 2012) because the woody tissues are an unlimited sink for surplus carbon. Low nitrogen availability limits plant $CO_2$ responses (Norby et al. 2010) in the competition runs (polyculture) because of high root allocation. Our model predicts increased root allocation at all nitrogen levels in response to elevated [$CO_2$] in the competition runs. Data from free air $CO_2$ enhancement (FACE) forest experiments largely agree (Drake et al., 2011; Iversen et al., 2012; Jackson et al., 2009; Lukac et al., 2003; Nie et al., 2013; Pritchard et al., 2008; Smith et al., 2013). However, in ORNL-FACE, the increases in root production due to elevated $CO_2$ increase and then declined after 8 years of $CO_2$ enhancement (Iversen, 2010; Norby and Zak, 2011). Though there are no direct data available for quantitatively validating the patterns predicted by our model, especially for the long-term, competitive runs, a detailed modeling analysis can help to understand the varied patterns in the experiments and shed light on the modeling of allocation"

"As shown in model inter-comparison studies, the mechanisms of modeling allocation differ very much, leading to high variation in their predictions (e.g., De Kauwe et al. 2014). Calibrating model parameters to fit data may not increase model predictive skill because data are often also highly variable. Franklin et al. (2012) suggest that in order to build realistic and predictive allocation models, we should correctly identify and implement fundamental principles. Our model predicts similar patterns to those of Valentine and Mäkelä (2012), which are very different in their details but share fundamental principles, including 1) evolutionary- or competitive-optimization, 2) capped leaves and fine roots, 3) structurally unlimited stem allocation (i.e., for optimizing carbon use), and 4) height-structure competition for light and root-mass-based competition for soil resources. The principles 2 and 3 are commonly used in models (De Kauwe et al., 2014; Jiang et al., 2019). However, the different rules of implementing them (e.g., allometric equation, functional relationships, etc. ) lead to highly varied predictions (as shown in De Kauwe et al., 2014), though the formulations may be very similar. In competitively-optimal models, such as this study and also Valentine and Mäkelä (2012), the competition processes generate similar emergent patterns by selecting those that can survive in competition, regardless the details of those differences. "

*Moreover, the author highlighted that competition for light and nutrient drives successional dynamics (e.g. L 46, 83, 105-107, etc.), which left me with the impression that successional dynamics is a key component of the paper. But in fact, it surprises me that the authors only included results on successional dynamics in the supplementary materials, and there's little discussion around this topic. I'd suggest tightening up Figure S4 and S5, and move them into the main text, with more thorough discussions around them.*

We added a succession pattern figure (below) into the paper (Fig. 4 in revised manuscript).

[Figure]

**Minor comments:**

*L 24-26: question: does this mean fixed allocation performs similarly in predicted NPP when compared to those based on competitive-allocation strategy? So the change in allocation pattern does not result in any changes in predicted NPP?*

Yes, though NPP changes a little bit. Changes in allocation pattern mainly affect the relative allocation between stems and fine roots. Leaves' NPP is similar at the same nitrogen level.

*L35-36. It's a bit unclear what the author trying to illustrate here.*

We rephrased this sentence as "competition leads to higher plant biomass response to elevated [$CO_2$] with increasing nitrogen availability".

*L38: "ecosystem-level predictions" of what? You indicated earlier that the predicted NPP was similar, right?*

Yes, it's ecosystem carbon storage. Clarified in revised manuscript: "significantly different ecosystem-level predictions of carbon storage than those that use fixed strategies".

*L230. The symbol $\varphi_{RL}$ was described here, which appears to be too late. Suggest to define it in its first appearance.*

We have moved it to the equation where it is used first time and reorganized this section.

*Figure 2. The author showed how competition runs compared differently to the fixed allocation runs, based on $\varphi RL =4$. Since you are talking about succession and competition, it remains unclear what is the community response?*

We added the successional figures as Fig. 4 (copied above).

*Figure 3. Missing unit on y-axis.*

Added.

**Response to Referee #3 (Dr. Benjamin Stocker)**

*This paper describes a model and its prediction for competitively optimal allocation (ratio of root to foliage surface area) and how it varies with a range of nitrogen levels and two CO2 levels. The same (or a similar version of the same? See point 12 below) model has been described and applied previously to investigate optimal phenological strategies in Weng et al. (2016, GCB, doi: 10.1111/gcb.13542) and a carbon only version was presented in Weng et al. (2015, BG, doi:10.5194/bg-12-2655-2015).*

Thanks for the summary.

*The present manuscript addresses allocation as a single variable parameter, although multiple traits affect plant functioning in the face of N availability and CO2 concentrations. However, the focus on allocation is justified, in my opinion, for two reasons: First, allocation warrants particular attention as it is a key process that is known to be responsive to changes in above and belowground resource availabilities and affects the carbon cycling in ecosystems when an allocation shifts occur between long-lived wood and short-lived foliage or fine roots. Second, most vegetation models simulate allocation either based on fixed parameters, or based on empirical relationships. However, as pointed out in the manuscript (l. 528), predicting allocation from first principles is key to realistically and robustly simulating vegetation changes in response to multiple changing environmental factors. The present model embodies a promising way forward to resolve this challenge, determining a competitively optimal allocation strategy, based on height-structured competition for light in the form of a Vegetation Demographics Modelling approach (cohort-based, not average individual-based as is common in Dynamic Vegetation Models). In this respect, the present model takes a pioneering path, that only less than a handful (or even less?) other models can follow.*

Thanks for the comments.

*I see two main weaknesses of the present study. **First, predictions are not tested against observational data**. What data needs to be used as a test (see comment 10)? However, I don't expect much additional insight from a comparison to observational data at this stage, and consider the theoretical predictions to deserve to be presented as "naked predictions". However, for a paper that deals with just the theoretical side of the problem, some aspects may warrant additional attention (additional figures for results and extended discussion) in the manuscript (see comments 6, 8, 10, 11). **Second, the modelling approach with regards to the excess-C-sapwood allocation** (see l.245) raised some questions for me. Is it compatible with our understanding for what controls sapwood area (or what determines the Huber value, defined as the ratio of sapwood area: leaf area)? I worry that this model formulation may cause unrealistic*

*model behaviour in some instances. Anyways, the conclusions need to be drawn carefully with regards to this aspect (see comment 5 below).*

I agree with Dr. Stocker. We have updated the description of allocation scheme (copied in response to a major comment of reviewer #1) and discussed the comparison of our simulations with observations and simulations of other models.

Specifically for the Huber value (sapwood area/leaf area), it is kept constant for a PFT, following Eq. 3 ($A_{SW}^*(D) = \alpha_{CSA} \cdot l^* \cdot A_{CR}(D)$) in the main text of the revised manuscript, where Huber value is a PFT-specific parameter ($\alpha_{CSA}$). Since the allometry equations are correct (Eq. 2 in the main text) and the model doesn't have processes of plant hydraulics, Huber value is just used to differentiate sapwood and heartwood and does not affect the functions of the trees. We revised the description of this equation in the manuscript and also copied below:

"We set *targets* for leaf ($L^*$), fine root ($FR^*$), and sapwood cross-sectional area ($A_{SW}^*$) that govern plant allocation of non-structural carbon and nitrogen during growth. These *targets* are related by the following equations based on the assumption of the pipe model (Shinozaki, Kichiro et al., 1964):

$$L^*(D, p) = l^* \cdot A_{CR}(D) \cdot LMA \cdot p(t)$$
$$FR^*(D) = \varphi_{RL} \cdot l^* \cdot \frac{A_{CR}(D)}{SRA} \qquad\qquad \text{(Eq. 3)}$$
$$A_{SW}^*(D) = \alpha_{CSA} \cdot l^* \cdot A_{CR}(D)$$

where $L^*$(D, p), $FR^*(D)$, and $A_{SW}^*(D)$ are the targets of leaf mass (kg C/tree), fine root biomass (kg C/tree), and sapwood cross sectional area (m$^2$/tree), respectively, at tree diameter $D$; $l^*$ is the target leaf area per unit crown area of a given PFT; $A_{CR}(D)$ is the crown area of a tree with diameter $D$; LMA is PFT-specific leaf mass per unit area; and $p(t)$ is a PFT-specific function ranging from zero to one that governs leaf phenology (Weng et al., 2015); $\varphi_{RL}$ is the target ratio of total root surface area to the total leaf area; SRA is specific root area; and $_{CSA}$ is an empirical constant (the ratio of sapwood cross-sectional area to target leaf area). The phenology function $p$(t) takes values 0 (non-growing season) or 1 (growing season) following the phenology model of LM3-PPA (Weng et al., 2015). The onset of a growing season is controlled by two variables, growing degree days (GDD), and a weighted mean daily temperature ($T_{pheno}$), while the end of a growing season is controlled by $T_{pheno}$."

The "excess-C-sapwood allocation" is a numerical step to adjust the growth of leaves, fine roots, wood, and seeds according to available carbon and nitrogen. We have replaced the whole section with a new description of plant growth and allocation. And we also compiled a detailed description of the model in supplementary information I. Please see section "C. Plant growth and carbon allocation", where we have described the conversion from sapwood to heartwood in detail.

Copied below:

"**Conversion from sapwood to heartwood**

As trees grow, sapwood (SW) is transformed to heartwood (HW). This unidirectional process does not affect the size of the woody biomass C pool. We assume that if the actual sapwood cross-sectional area $A_{SW}$ is larger than its target value, $A_{SW}^*(D)$, the excess portion of sapwood biomass is converted to heartwood. Thus, to determine the amount of sapwood converted to heartwood in a given time step ($dHW$), we simply calculate the difference between $SW$ and the target sapwood C ($SW^*$) needed to balance $L^*$ and $FR^*$:

$$dHW = \max(0, SW - SW^*) \tag{C19}$$

Using the equation for total tree biomass (main text Eq. 4), the target biomass of sapwood is:

$$SW^* = 0.25\pi\Lambda\rho_W\alpha_Z(D^{2+\theta_Z} - D_{HW}^{2+\theta_Z}) \tag{C20}$$

where $D$ is the diameter of the trunk and $D_{HW}$ is the heartwood diameter, which is given by:

$$D_{HW} = 2\sqrt{A_{HW}/\pi} \tag{C21}$$

where $A_{HW}$ is the cross-sectional area of heartwood. Assuming $A_{SW}$ is at its target value,

$$A_{HW} = A_t - A_{SW}^* \tag{C22}$$

The cross-sectional area of a trunk ($A_t$) is:

$$A_t = \pi\left(\frac{D}{2}\right)^2 \tag{C23}$$

And, according to Eq A2.1 and Eq A2.3, the target cross sectional area of sapwood is defined as:

$$A_{SW}^* = \alpha_{CSA}l^*A_{CR}(D) = \alpha_{CSA}l^*\alpha_C D^{\theta_C} \tag{C24}$$

"

*In the "SPECIFIC POINTS" described below, I am suggesting some modifications in the description of the model and to improve readability, and some changes in the presentation to distill the most relevant points and most interesting insights from this research. GENERAL POINTS may warrant particular attention. The paper is generally written very well and the presentation of results is clear and clean. If these points can be addressed satisfactorily, I may recommend a revised version of this manuscript for publication in Biogeosciences.*

*GENERAL POINTS*

*1. Dynamic adaptation (adaptive plastic responses) of allocation occurs also within species. The present model embodies the assumption that species follow a fixed allocation strategy, and changes in allocation are induced by replacement of species with different allocation strategies.* **Please add a discussion for the assumption of fixed within-species allocation may affect results.**

Our allocation scheme itself is flexible and can change with environment even with one fixed scheme of allocation (i.e., fixed $\varphi_{RL}$) by numerically maximizing carbon usage at nitrogen limitation. The key step in solving this set of equations is to solve $r_{D/S}$ in each growth step (daily in this simulator). The parameter $r_{D/S}$ changes with relative nitrogen availability. When there is no nitrogen limitation, $r_{D/S}$ equals to 1 and the allocation follows the conditions defined by Eqs. 7.1~7.5 (carbon only model). When $r_{D/S}$ equals to 0, $G_N$ does not meet the nitrogen demand even if all the $G_C$ is allocated to wood tissues and the model has to return the excessive carbon to the NSC pool (this is a very rare case because of low carbon input long before $r_{D/S}$ approaches to 0 due to nitrogen limitation. However, our codes must be able to deal with all possible cases.). When $r_{D/S}$ is in between ($0 < r_{D/S} < 1$), the leaves and fine roots cannot reach to their targets after this step of growth (i.e., plants maintain a low LAI in low nitrogen environments). At low nitrogen availability, the parameter $r_{D/S}$ keeps a low LAI, a relatively constant NPP/GPP ratio, and a relatively stable NSC for each single PFT.

We have updated the section in the revised manuscript to give a detailed description mathematically (**Allocation and plant growth**).

We have added two paragraphs to discuss this allocation scheme following this suggestion in Discussion.

"In our model, the allocation of carbon and nitrogen within an individual tree is based on allometric scaling, functional relationships, and optimization of resource usage. Basically, the allometric scaling relationships define the maximum leaf and fine root growth at a given tree size and the functional relationships (pipe model) define the ratios of leaf area to sapwood cross-sectional area and fine root surface area. These rules are commonly used in ecosystem models (Franklin et al., 2012) and have been shown to generate reasonable predictions (De Kauwe et al., 2014; Valentine and Mäkelä, 2012). Overall, these rules lead to the priority of allocation to leaves and fine roots but allow for structurally-unlimited stem growth when resources (carbon and nitrogen in this study) are available (i.e., the remainder goes to stems after leaf and fine root growth).

We define a maximum leaf and fine root allocation, $f_{LFR,max}$, to limit the maximum allocation to leaves and fine roots to maintain a relatively stable growth rate of wood in years of low productivity. The simulated wood growth patterns agree with real wood growth in temperate trees (Cuny et al., 2012; Michelot et al., 2012). Trees need to grow new wood tissues continuously (especially early in the growing season) to maintain their functions (Plomion et al., 2001). This parameter does not change the fact that leaves and fine roots are the priority. Since allocation ratios to stems are around 0.4~0.7 in temperate forests (Curtis et al., 2002; Litton et al., 2007), with a value of 0.85, $f_{LFR,max}$ only seldom affects the overall carbon allocation ratios of leaves, fine roots, and stems, and still maintains wood grow in years of low productivity. If $f_{LFR,max} = 1$ (i.e., the highest priority for leaf and fine root growth), simulated trunk radial growth would have unreasonably high interannual variation because leaf and fine root growth would use all carbon to approach to their targets, leaving nothing for stems in some years of low productivity."

*2. A competitively optimal strategy is determined for stationary boundary conditions. Does this inhibit such a modelling approach to be applicable for global change simulations (transiently changing boundary conditions)? See also comment 11.*

This study is not for directly application at global scale, but the succession patterns can be used to understand transient responses of vegetation to climate changes and the model formulations are being incorporated into an Earth system modeling framework (NASA GISS ModelE). Our analysis can help to understand model behavior at global scales.

We also added a paragraph in Section **4.3 Implications for Earth system modeling** to discuss the emerging principles of allocation modeling:

"As shown in model inter-comparison studies, the mechanisms of modeling allocation differ very much, leading to high variation in their predictions (e.g., De Kauwe et al. 2014). Calibrating model parameters to fit data may not increase model predictive skill because data are often also highly variable. Franklin et al. (2012) suggest that in order to build realistic and predictive allocation models, we should correctly identify and implement fundamental principles. Our model predicts similar patterns to those of Valentine and Mäkelä (2012), which are very different in their details but share fundamental principles, including 1) evolutionary- or competitive-optimization, 2) capped leaves and fine roots, 3) structurally unlimited stem allocation (i.e., for optimizing carbon use), and 4) height-structure competition for light and root-mass-based competition for soil resources. The principles 2 and 3 are commonly used in models (De Kauwe et al., 2014; Jiang et al., 2019). However, the different rules of implementing them (e.g., allometric equation, functional relationships, etc. ) lead to highly varied predictions (as shown in De Kauwe et al., 2014), though the formulations may be very similar. In competitively-optimal models, such as this study and also Valentine and Mäkelä (2012), the competition processes generate similar emergent patterns by selecting those that can survive in competition, regardless the details of those differences."

*3. Allocation and the plant-level C and N budgets, assume fixed tissue C:N ratios and flexible allocation to sapwood to make C and N budgets work. **This does not correspond to the known controls on sapwood area and may induce unwanted model behaviour.** However, it is difficult to think understand to what degree this affects the results and conclusions. See also comment 5.*

The allocation between fine roots plus leaves and wood does not affect the differentiation between sapwood and heartwood. We have a detailed description of the conversion from sapwood to heartwood in the appendix of Weng et al. 2015. The new tissues of wood are always counted as sapwood, and the oldest tissues are converted to heartwood. Actually, the differentiation between sapwood and heartwood does not affect model function in this model because we don't have plant hydraulics yet in this model. We included a supplementary material to describe the conversion from sapwood to heartwood. This section is copied in the response to General Points 1.

*4. Total N that is circulating in the system is manipulated for the different simulations, with zero N loss and inputs. This sounds like a rather pragmatic than realistic solution. In reality, losses are never zero, and N levels are manipulated in experiments by fertilisation. Why is it not implemented like this? Would you expect any systematic differences between your implementation and one with non-zero losses and manipulated inputs?*

The processes of nitrogen inputs and losses are complex, involving nitrogen deposition, fixation, soil nitrogen mineralization, root uptake efficiency, microbial immobilization, leaching, denitrification, etc.. However, at equilibrium state, the total ecosystem nitrogen is determined by relative rates of nitrogen input and nitrogen output. For example, a high nitrogen input and low output lead to high total ecosystem nitrogen level. In this study, we pack all these effects by setting different total N levels to represent implicitly relative nitrogen input and loss rates. For example, a high total ecosystem nitrogen level represents the ecosystems and edaphic conditions with relative high nitrogen input and low loss rates. By packing these processes into different total N levels, we can focus our study on allocation.

We did both nitrogen closed (with zero nitrogen input and loss) and nitrogen open (with non-zero nitrogen input and output) runs in a previous study (Weng et al. 2017, Global Change Biology). In both types of runs, the nitrogen level determines the competition of PFTs and the competitively optimal player since plants can only "feel" the nitrogen availability, and don't "care" how this availability is set (e.g., either by setting the total N or through complex input and output processes).

We explained the meaning of the different nitrogen levels in the section of simulation experiments.

"In all the simulation experiments, we assume the ecosystem has no nitrogen inputs and no outputs for convenience since we already have eight total nitrogen levels to represent the consequences of different nitrogen input and output processes at equilibrium state."

*5. The abstract mentions "opposite fractional allocation to fine roots and wood" in competitive and fixed-allocation runs. **Relatively more allocation to fine roots at high N levels in fixed allocation runs sounds like a result that runs counter to the common understanding of the controls on fine root allocation** (e.g. Poorter et al., 2012 New Phytologist, doi: 10.1111/j.1469-8137.2011.03952.x), and appears confusing in view of the fact that the model is based on a fixed*

*root:leaf area ratio.* ***I suspect that the increase in relative root allocation at high N levels in the fixed-allocation run is a result of the excess-C-sapwood allocation in this model.*** *See also my comment 3. If this is indeed the case, I would recommend not to present results in the abstract that are contingent on this, arguably unrealistic, model behaviour. I guess the authors don't want to challenge the common understanding of fine root allocation controls with their results. It would suffice to point out that in the competition runs, relative fine root allocation de- creases with increasing N levels; and present this in the light of the common modelling approach whereby the root:leaf area ratios (and effective relative allocation ratios) are fixed.*

Thanks for the suggestion. High allocation to fine roots is resulted from "**excess-C-sapwood allocation**" in the allocation scheme for the individual growth (numerical solution of parameter $r_{D/S}$). The model solves two parameters in these model runs: $r_{D/S}$ (at each daily step for both single PFT runs and multi PFTs runs) and $\varphi_{RL}$ (after more than a thousand years model runs with multiple PFTs).

Our results are consistent with Poorter et al. 2012 well (Fig. 1:f of Poorter et al. 2012), because their data should be compared with our multi-PFT, long-term simulation results (i.e., competitively optimal strategy in our Fig. 3). For a single PFT (fixed strategy and short-term responses to nitrogen addition), it should be compared with fertilization experiments (e.g., Lee & Jose 2003, Forest Ecology & Management).

[Figure]

leaves (LMF; red line), stems (SMF; brown line) and roots (RMF; blue line)

We have added a paragraph in discussion to validate our simulation results.

"As shown in model inter-comparison studies, the mechanisms of modeling allocation differ very much, leading to high variation in their predictions (e.g., De Kauwe et al. 2014). Calibrating model parameters to fit data may not increase model predictive skill because data are often also highly variable. Franklin et al. (2012) suggest that in order to build realistic and predictive allocation models, we should correctly identify and implement fundamental principles. Our model predicts similar patterns to those of Valentine and Mäkelä (2012), which are very different in their details but share fundamental principles, including 1) evolutionary- or competitive-optimization, 2) capped leaves and fine roots, 3) structurally unlimited stem allocation (i.e., for optimizing carbon use), and 4) height-structure competition for light and root-mass-based competition for soil resources. The principles 2 and 3 are commonly used in models (De Kauwe et al., 2014; Jiang et al., 2019). However, the different rules of implementing them (e.g., allometric equation, functional relationships, etc. ) lead to highly varied predictions (as shown in De Kauwe et al., 2014), though the formulations may be very similar. In competitively-optimal models, such as this study and also Valentine and Mäkelä (2012), the competition processes generate similar emergent patterns by selecting those that can survive in competition, regardless the details of those differences."

*6. Presentation of results for a fixed φ: In my understanding, the essence of this research is to predict how stand-level relative allocation changes in response to N availability and CO2. The approach to predict it is to derive a competitively optimal allocation strategy at the individual-level. In view of the main aim (essence) of this research, I would expect as a (first) figure something that shows competitively optimal φ for each N/CO2 level, derived from the competition runs.*

Thanks for the suggestion. We added a panel into Fig. 5 in revised manuscript (original Fig. 3) to show the winning strategy in polyculture runs, which shows the closest $\varphi_{RL}$ to the competitively optimal. We also added two figures in supplementary material II (Fig. S4 and S5) to show the winning $\varphi$ for each N and $CO_2$ level in polyculture runs II.

[Figure]

*7. A question: Does each point in Fig. 3 show the wood/root/foliage NPP (fraction) for the competitively optimal φ at the respective N and CO2 levels (for the competition runs)? Comparing this to values from fixed-allocation runs with a pre-specified φ is maybe not the most interesting. This leads to the next point. . .*

The left is absolute, and right is fraction. We have updated this figure (Fig. 6 in revised manuscript)

[Figure]

*8. [see Addendum to this comment] In view of my comment 1 (dynamic adaptation of allocation also within species), readers may want to know **if an optimality criterion that is defined for some instantaneous individual-level metric (like foliage NPP, or height growth, etc.) leads to the (qualitatively) same predictions as an optimality criterion that is funded in an evolutionarily stable strategy (ESS, like the one applied here)**. It might be interesting to evaluate the fixed-allocation runs to determine which φ maximises some optimality criterion (like foliage NPP, or height growth, etc.). A (first) figure could then compare this individual-level derived optimal φ to the ESS-derived optimal φ. Furthermore, points in Fig. 3 for the fixed-allocation runs could then be taken to represent the φ level that maximises the optimality criterion for each N/CO2 level respectively. This would also enable a direct comparison of the*

*two optimality approaches (ESS vs. instantaneous). I understand if this suggestion is beyond the scope of this paper, or not feasible. Anyways, I would be interested to learn more about such a comparison. Addendum: I see now (upon reading the Discussion, l. 425), that this point is addressed by Supplementary figures S1 and S2. Maybe this is too much of my own personal interest, but I think to generally enhance this point in the presentation of the results (possibly with additional/modified figures as suggested above) would improve the manuscript.*

We explained the differences between optimal vs. competitively optimal in introduction following reviewer's suggestion. Basically, optimal strategy is not necessarily the competitively optimal. From Figures 1 and 2, you can see that low root allocation is optimal in many N levels. However, they cannot outcompete high root allocation strategies if invaded by them. That means, in the environment created by the optimal strategy individuals, another strategy can out compete the resident, though it may have low NPP or fitness in monoculture. The revision is copied below:

"The competitively optimal strategy is the one that can successfully exclude all others in the processes of competition and succession, but it is not necessarily the one that maximizes production in monoculture. For example, each $\varphi_{RL}$ creates an environment of light profile and soil nitrogen in its monoculture. Other $\varphi_{RL}$ PFTs may have higher fitness in this environment than the one who creates it. Only the competitively dominant strategy has the highest fitness in the environment it creates (Fig. 1: C)."

*9. A description of how the competitively optimal allocation is determined (description of the algorithm), would be helpful. May be added before current Section 2.2.*

We have added a description of competitively optimal strategy following this suggestion.

"The process of choosing a context-dependent competitively dominant $\varphi_{RL}$ will take place after finding the fitness of each $\varphi_{RL}$ in monoculture and in competition with other PFTs (*i.e.*, different values of $\varphi_{RL}$). The competitively optimal strategy is the one that can successfully exclude all others in the processes of competition and succession, but it is not necessarily the one that maximizes production in monoculture. For example, each $\varphi_{RL}$ creates an environment of light profile and soil nitrogen in its monoculture. Other $\varphi_{RL}$ PFTs may have higher fitness in this environment than the one who creates it. Only the competitively dominant strategy has the highest fitness in the environment it creates (Fig. 1: C). "

*10. The differences in predictions based on different allocation schemes are interesting, but the missing comparison to observational data prevents conclusions to be drawn about which is more realistic or leads to better model performance. The question is: What is the key observation that can be used to test predictions? And of course: What is the key prediction that authors want to test? In my view, it is a viable option to remain with theoretical predictions, not actually using*

*data from observations. However, this may require an extended discussion in the light of generally observed patterns with additional references to the literature. The challenge remains that the overproliferation in root growth predicted by the competitively-optimal allocation scheme may not directly be testable.* ***How much is over-proliferation in reality? What would be a suitable observation to evaluate this prediction?***

We added a paragraph in discussion to discuss the experimental results related to root overproliferation and proposed our expectation of what we want to get from new experiments.

"Root overproliferation is still controversial in experiments. For example, Gersani et al. (2001) and O'Brien (2005) found competing plants generate more roots than those planted isolated for pea and soybeans, respectively; whereas, McNickle and Brown (2014) found root growth follows the availability of soil nutrients and individuals growth with competitors have the same root growth as that predicted by the changed nutrient availability. Roots are far more adaptive and complex than those simulated in models at modifying their growth patterns in response to soil nutrient and water dynamics (Hodge, 2009). The root growth strategies in response to competition also vary with species (Belter and Cahill, 2015). The mechanisms of self-recognition of inter- and intra- roots also can lead to varied behavior of root growth (Chen et al., 2012). However, all of the aforementioned studies considered only *plastic* root overproliferation, where individuals produce more roots in the presence of other individuals than they do in isolation, analogous to stem elongation of crowded seedlings (Dudley and Schmitt, 1996). A portion of root overproliferation may also be *fixed*, analogous to trees that still grow tall even when grown in isolation. Dybzinski et al. (2019) showed that plant community nitrogen uptake rate was independent of fine root mass in seedlings of numerous species, suggesting a high degree of fixed fine root overproliferation. To improve root competition models, more detailed experiments that control root growth should be conducted to quantify the marginal benefits of roots in isolated, monoculture, and polyculture environments. "

*11. Regarding implications for Earth system modelling: From reading this manuscript, it's not entirely clear whether the approach for determining the competitively-optimal allocation strategy is applicable for typical Earth system simulations, where boundary conditions change transiently. In my understanding, the approach chosen here determines a system steady state, formed by a monospecific stand with a certain allocation strategy, that cannot be invaded by any species with a different allocation strategy. As explained in the manuscript, this requires the model to be run into steady state for 1200 simulation years. How would this be implemented for a typical Earth system simulation setup? I think it would be very informative to complement Section 4.3 with a discussion on this point.*

It is helpful for understanding model behavior and track competitively optimal strategy.

We are exploring one more layer of complexity, adaptive dynamics (or successional dynamics), on the top of the pool-flux dynamics and demographic dynamics. For a robust DGVM in Earth system models, it is necessary to analyze these processes and their effects on model performance in detail.

We added a paragraph in "**4.3 Implications for Earth system modeling**" to discuss the emerging common principles of allocation modeling:

"As shown in model inter-comparison studies, the mechanisms of modeling allocation differ very much, leading to high variation in their predictions (e.g., De Kauwe et al. 2014). Calibrating model parameters to fit data may not increase model predictive skill because data are often also highly variable. Franklin et al. (2012) suggest that in order to build realistic and predictive allocation models, we should correctly identify and implement fundamental principles. Our model predicts similar patterns to those of Valentine and Mäkelä (2012), which are very different in their details but share fundamental principles, including 1) evolutionary- or competitive-optimization, 2) capped leaves and fine roots, 3) structurally unlimited stem allocation (i.e., for optimizing carbon use), and 4) height-structure competition for light and root-mass-based competition for soil resources. The principles 2 and 3 are commonly used in models (De Kauwe et al., 2014; Jiang et al., 2019). However, the different rules of implementing them (e.g., allometric equation, functional relationships, etc. ) lead to highly varied predictions (as shown in De Kauwe et al., 2014), though the formulations may be very similar. In competitively-optimal models, such as this study and also Valentine and Mäkelä (2012), the competition processes generate similar emergent patterns by selecting those that can survive in competition, regardless the details of those differences."

*12. The description of how the present model version differs from model versions used in Weng et al., 2016, 2015, could be made clearer.*

We have compiled a full description of the model in a supplementary material I.

**SPECIFIC POINTS**

*l.96: Should be a gap in 'trait (s)'?*

We removed this equation following Reviewer 1's suggestion.

*Eq. 2: To solve the model, ds/dt has to be set to zero, I guess. Shouldn't this be reflected in Eq. 2? Or how exactly is the competitively optimal strategy determined?*

This equation is used to show the temporal dynamics of a plant trait in evolution, which, conceptually, is the numerical shifts of a plant trait approaching its ESS in simulations.

For analytically solve this equation, one needs to build up a fitness equation, such as:

$$Fitness(s_I|s_R) = Benefit(s_I|s_R) - Cost_{I,R}(s_I|s_R) - Cost_I(s_I)$$

where, $s_I$ is invader's trait and $s_R$ is residence. Let the derivative of the right side to be zero and according to the definition of ESS $s_I=s_R$. Then, it is possible to solve it analytically if you are lucky.

We removed this equation following Reviewer 1's suggestion. We agree with the Reviewer 1 that this equation is not very helpful here. We will add a paragraph to explain the processes of approaching ESS by succession in VDMs. Since the numerical simulations are complex, we don't want to complicate the paper with more discussions of this equation.

*l.148: Apart from variations in across-species allocation patterns (e.g., oak species tending to invest more into roots), there are also clear patterns in within species and within-individuals (flexible adaptation) variations in allocation when subjected to shifts in resource availability. In my understanding, such fast allocation responses are not captured by the modelling approach here. This should be clarified. Connects to Comment 1 above.*

They have to follow the same rules, otherwise they will be replaced by those who follow the rules. Actually, we used only one PFT, which only differs in fine root – leaf area ratio ($\varphi_{RL}$). If plants have enough plasticity in allocation, they will approach the ESS much quicker. We will make it clear in a revised manuscript.

*l.159: The simulation experiments are described in the abstract and intro to be done along a "nitrogen availability gradient". How did you manipulate N? This is described at a later stage, but could already be made clear here.*

We clarified it in the description of Simulation experiments, but did not mention those settings here because they are not part of the universal feature of the model.

"We set two atmospheric $CO_2$ concentration ($[CO_2]$) levels: 380 ppm and 580 ppm, and eight ecosystem total nitrogen levels (ranging from 114.5 gN m$^{-2}$ to 552 gN m$^{-2}$ at the interval of 62.5 gN m$^{-2}$) by assigning the initial content of the slow SOM pool for our simulation experiments (Table 1). This range covers the soil nitrogen content at Harvard Forest (Compton and Boone, 2000; Melillo et al., 2011). The nitrogen cycles through the plant and soil pools and is redistributed among them via plant demographic processes, soil carbon transfers, and plant uptake. In all the simulation experiments, we assume the ecosystem has no nitrogen inputs and no outputs for convenience since we already have eight total nitrogen levels to represent the results of different nitrogen input and output processes at equilibrium state. "

*Section 2.1: A separate paragraph on how CO2 assimilation is simulated, would be helpful.*

We added a brief description of the photosynthesis model, which is the same as it in LM3-PPA and LM3.

"Plant growth and reproduction are driven by the carbon assimilation of leaves via photosynthesis, which is in turn dependent on water and nitrogen uptake by fine roots. The photosynthesis model is the same as it in LM3-PPA (Weng et al., 2015), which is a simplified version of Leuning model (Leuning et al., 1995). This model first calculates photosynthesis rate, stomatal conductance, and water demand of the leaves of each tree (cohort) in the absence of soil water limitation. Then, it calculates available water supply, and reduce the demand-based assimilation and stomatal conductance accordingly if water supply is less than water demand. Assimilated carbon enters into the NSC pool and is subsequently used for respiration, growth, and reproduction. (Please see Supplementary Information I-A for detail)."

*Eq. 3: To be consistent with Fig. 1, I would suggest to use the symbol X_FR as the pool size (or C_FR in this case), distinguish community- and individual-level variables for example using a bar over the variable for the community-level, and use a separate letter for the parameter 'Root_0' (e.g., K_FR).*

Done as suggested. (In revised manuscript, it Eq. 4)

*Eq. 4: Should be clarified that this is the community-level total root biomass (if I am correct). Clarification is needed to understand Eq. 4.*

Clarified. $C_{FR}$, total is the total roots in soil, with unit of kgC/m2 and CFR,tree is the total root carbon of a tree (in a cohort with identical trees) with unit of kgC/tree. This equation transforms nitrogen uptake rate from gN m-2 hour-1 to gN tree-1 hour-1. We also added a tale (Table 1) in the revised manuscript for the major parameters of this model.

*Eq. 5: D is not defined. Diameter?*

Yes, it is diameter. defined.

*l. 215: Add bracket: ". . . targets for leaf, fine root, and sapwood cross-sectional area (L\*_k, FR\*_k, and A\*_SW)" here for a better overview of the description.*

Done as suggested.

*Eq. 5/6: How is D incremented? The way the system is described now, the tree doesn't grow in D or am I missing something?*

We add a new equation to show diameter growth.

l. 238: Can you add f_1 and f_2 to the description in this sentence? E.g., "capped by a larger fraction of NSC (f_1)"?

Done as suggested. We switched the definition of f1 and f2 for convenience.

*l. 241: Are Eq. 7 and its parameters f_1 and f_2 identical throughout the year? The description here suggests that something is different during leaf flush versus the period of "normal growth". Or maybe I'm just misunderstanding it the way it's formulated now.*

They are constant over time. Generally, "$f_1$NSC" defines the maximum NSC availability at the beginning of a growing season when (L+FR)<<(L$^*$+FR$^*$); "$f_2$NSC"defines the normal growth of stems after (L+FR) is close to (L$^*$+FR$^*$) (i.e., normal growth during the growing season; "[(L$^*$+FR$^*$) -(L+FR)]" are the carbon for the new leaves and fine roots due to stem growth in last time step.

The term "$f_1$NSC" also prevents overspending of NSC when NSC is very low in some bad years.

*l. 246: Units of 0.15? g C?*

It's the fraction of NSC. Clarified. In revised manuscript, we define a parameter to cap the maximum fraction of available carbon allocated to leaves and fine roots, $f_{LFR,max}$, as:

- 0.15 =0.85.

*l. 254: Since sapwood production requires N as well (although relatively less than production of other tissues), and "excess C" sounds like this is the amount of NSC left, after NSN is used up (hence zero), I would assume that some iteration is necessary to perfectly match the use of NSC and NSN in the allocation procedure. How is this solved? Either more detail should be given here, or the description should be modified to avoid misunderstanding.*

It can be analytically solved. In the codes, we numerically solve it by iterating at daily step and allowing the wood C:N ratio to variate from day to day. We clarified the whole allocation section it in the revised manuscript and added the analytical solution.

"The parameter $r_{S/D}$ controls the allocation of $G_C$ and $G_N$ to the four plant pools (Eq. 7.1). It can be analytically solved (Eqs. 8 and 9).

$$r_{S/D} = Min\left[1, Max\left(0, \frac{G_N - G_C/CN_W}{N_{demand} - G_C/CN_W}\right)\right] \; , \tag{Eq. 8}$$

where, $N_{demand}$ is the potential N demand for plant growth at $r_{S/D}$=1 (i.e., no nitrogen limitation).

$$N_{demand} = \frac{\gamma\sigma\left[FR + Min\left(\frac{L^* + FR^* - L - FR,}{f_{LFR,max}\,G_C}\right)\right] - \varphi_{RL}L}{(\gamma\sigma + \varphi_{RL})CN_L} +$$

$$\frac{\varphi_{RL}\left[L + Min\left(\frac{L^* + FR^* - L - FR,}{f_{LFR,max}\,G_C}\right)\right] - \gamma\sigma L}{(\gamma\sigma + \varphi_{RL})CN_{FR}} + \frac{v\left[G_C - Min\left(\frac{L^* + FR^* - L - FR,}{f_{LFR,max}\,G_C}\right)\right]}{CN_F} + \qquad\text{(Eq. 9)}$$

$$\frac{(1-v)\left[G_C - Min\left(\frac{L^* + FR^* - L - FR,}{f_{LFR,max}\,G_C}\right)\right]}{CN_W}.$$

When $G_N \geq N_{demand}$ ($r_{S/D} = 1$), there is no nitrogen limitation, and all the $G_C$ will be used for plant growth and the allocation follows the rules of the carbon only model (Eqs 7.4~7.6 as $r_{S/D} = 1$). The excessive nitrogen ($G_N - N_{demand}$) will be returned to the NSN pool. When $G_C/CN_{W,0} < G_N < N_{demand}$ (i.e., $0 < r_{S/D} < 1$), all $G_C$ and $G_N$ will be used in new tissue growth; however, the leaves and fine roots cannot reach their targets at this step. When $G_N \leq G_C/CN_{W,0}$ ($r_{S/D} = 0$), all the $G_N$ will be allocated to sapwood and the excessive carbon ($G_C - G_N CN_{W,0}$) will be returned to NSC pool. This is a very rare case since a low $G_N$ leads to low leaf growth, reducing $G_C$ before the case $G_N < G_C/CN_{W,0}$ happens. Therefore, in most cases, Eq. 7.1 is: $G_C = G_W + G_L + G_{FR} + G_F$."

*l. 260: I would welcome a summarising sentence on the mechanisms determining C:N stoichiometry. The connection between sapwood allocation and the NSC:NSN budget implies that a plant that acquires relatively little N in comparison to assimilated C (in other words: an "N-limited tree") would produce relatively more sapwood. Does this mean that Eq. 6 (the A_SW sub-equation) is "over"-satisfied? What are the implications of this in the model? Does it affect the relationship between height growth vs. crown area expansion?*

According to the allometry equation (Eq. 5 in the main text), tree height and crown area are functions of diameter. So, how to define sapwood does not affect anything in the current version of the model. The $A_{SW}$ sub-equation Eq. 6 is used to separate sapwood from the whole trunk. As mentioned in the responses to General comment 2, it does not affect anything of the model. We did this because the model needs this variable. (We really need it when incorporating plant hydraulics.).

We have updated the whole section with a new description of the model's allocation scheme (Pages 10~16, and also copied in response to Reviewer #1's major comments). The partitioning of carbon and nitrogen into the plant pools is constrained by allometric equations, targets of leaves and fine roots, and the C:N ratios of these plant pools. The plant growth (and therefore allocation) is simulated at two steps: 1) calculating the amount of carbon and nitrogen that will be used for plant growth at this time step (which can be hourly, daily, weekly, etc.); 2) allocating the available carbon and nitrogen to leaves, fine roots, seeds, and sapwood following rules of first principles. We let the plant growth follow the rules below as they are in the first version (Weng et al. 2015, Biogeoscieces. carbon only model) and second version of LM3-PPA (Weng et al. 2017, Global Change Biology. carbon and nitrogen). Overall, this is a flexible allocation scheme and still follows the major assumptions in the previous versions of LM3-PPA (Weng, et al., 2015, 2017). It prioritizes the allocation to leaves and fine roots, maintains a minimum growth rate of stems at nitrogen limitation, keeps the constant area ratio of fine roots to leaves, and optimizes resource usage by relocating carbon and nitrogen to wood tissues when nitrogen is not sufficient for full growth of leaves and fine roots. When nitrogen is limited, leaves and fine roots are lower than their targets, reducing photosynthesis and carbon supply.

*Section 2.1: A description of how the competitively optimal allocation is determined (description of the algorithm) would be helpful. May be added before current Section 2.2.*

We have added a description of "competitively optimal" in the end of Section 2.1.

"The process of choosing a context-dependent competitively dominant $\varphi_{RL}$ will take place after finding the fitness of each $\varphi_{RL}$ in monoculture and in competition with other PFTs (*i.e.*, different values of $\varphi_{RL}$). The competitively optimal strategy is the one that can successfully exclude all others in the processes of competition and succession, but it is not necessarily the one that maximizes production in monoculture. For example, each $\varphi_{RL}$ creates an environment of light profile and soil nitrogen in its monoculture. Other $\varphi_{RL}$ PFTs may have higher fitness in this environment than the one who creates it. Only the competitively dominant strategy has the highest fitness in the environment it creates (Fig. 1: C). "

*l. 292: "Full factorial" suggests that all combinations of treatment factors are applied to force the runs. But here, this is a mix of a treatment factor (N levels) and model parameter (phi). I suggest to rephrase this.*

We removed the word "factorial" and clarified the combinations of PFTs, N levels, and CO2 concentrations of the monoculture runs.

*Table 1: If i=(114.5, . . ., 552) g N m-2, then 4.5-0.5\*i is a negative number. Is this correct? Maybe N levels in units of kg N m-2 are used here?*

Here, *i* takes the value of 1, 2, 3, 4, …, 8, following the order of the nitrogen levels from 114.5 to 552 gN m$^{-2}$. We clarified it.

"For each nitrogen level, we set eight PFTs with $\varphi_{RL}$ that varied in a range 3.5 (e.g., $x \sim x+3.5$) at the interval of 0.5, starting with the highest $\varphi_{RL}$ of 8.0 at the lowest N level (114.5 gN m$^{-2}$) and decreasing 0.5 per level of increase in ecosystem total N. Let $i$=1, 2, …, 8 denote the eight N levels from 114.5 to 552 gN m$^{-2}$, the $\varphi_{RL}$ of the eight PFTs at each level are (5.0-0.5$i$, 5.5-0.5$i$, …, 8.5-0.5$i$) (Table 1). For example, at the nitrogen of 114.5 gN m$^{-2}$ ($i$ = 1), the $\varphi_{RL}$ of the eight PFTs are 4.5, 5.0, …, 8.0 and at 177 gN m$^{-2}$ ($i$ = 2), they are 4.0, 4.5, …, 7.5. "

*l. 362-364: The decrease in fractional allocation to fine roots with elevated CO2 at high N levels is surprising. May it be a result of the excess-C sapwood allocation approach implemented in this model? May warrant a brief discussion of this aspect.*

In monoculture runs, the fractional allocation to fine roots decreases with elevated CO2 at all N levels because of high nitrogen limitation due to high carbon assimilation (photosynthesis) at elevated CO2. It is consistent with field observations that high production forests have high wood allocation (Litton et al. 2007). In polyculture runs, it only happens in high N levels, because the differences in competitive φRL between the two CO2 concentrations become small (Fig. 4:a), while GPP increases are high, which leads to high fractional allocation to wood.

*l. **386-388: In my reading, this is a main result and should be shown in** a separate figure, shown at the very beginning of the results section.*

We moved this paragraph in 390~406 to the beginning of the results section and the figure S1 to the main text as Figure 3.

*l. 418-419: See my comment 5.*

We removed "but decreases – the opposite qualitative response – under fixed strategy".

*l.425: See my comment 8. This is an interesting point, but is dealt with rather briefly here. How is "maximising growth rate" implemented exactly? NPP? NPP of a specific pool? "Allocating very little" is vague. The crucial aspect is that for a given N level and uptake half-saturation constant, the plants allocate much less to fine roots in the best- performing (by what measure?) fixed-allocation run than in the competitively optimal- allocation run.*

We removed this sentence because the pattern not so strong in the new simulations, and we discussed the competitively optimal strategy by outcompeting other strategies.

*l. 430-433: Sentence is hard to follow. Is the height at transition into the canopy (reaching critical height) increasing or decreasing with increasing N?*

We rephrased this sentence: "Changes in the height at which understory trees transition to the canopy from low nitrogen to high nitrogen indicate a shift from the importance of competition for soil nitrogen to the importance of competition for light as ecosystem nitrogen increases." to:

"Increases in the critical height, which is the height of the shortest tree in canopy layer, from low nitrogen to high nitrogen indicates a shift from the importance of competition for soil nitrogen to the importance of competition for light as ecosystem nitrogen increases."

*l. 493 ("succession"): Discussing competitively optimal strategy shifts during succession confused me here. I understood, that the competitively optimal strategy is determined for a \*steady-state\*, and (based on my understanding from reading previous papers of this group of authors) an ESS is determined from competition upon invasion into a mono-specific stand. But now **I realise that the algorithm for determining competitive optimality has never been described in the present paper. A gap that should be filled (see also comments above).***

We have added a brief description of "competitively optimal strategy" in model description. In the simulations, the strategy closest to the competitively optimal is obtained by the polyculture runs (i.e., the one who survives 1800 years model run in competition with others).

*l. 569: I had a great laugh when I read the short description of that repository on github ("BiomeESS: for simulating multiple plant forms, on-going, unpublished, with ridiculous processes and many bugs.") Maybe the author wants to change that upon publication of this manuscript (and if necessary resolve some known bugs). If not, I appreciate the honesty.*

Thanks for taking a look! We have set a new branch for the version used in this paper and we will update the description upon accept of this paper. We also included a detailed description of the model as supplementary material of this paper.

**Additional sensitivity tests**

1. Extra carbon returned to non-structural pool when available nitrogen cannot meet the demand

[Figure]

[Figure]

[Figure]

**2: Higher soil nitrogen mineralization rate**

[revised manuscript text omitted]

---

## Author Response (AR2)

Dear Dr. Zaehle,

Thanks for your suggestions on addressing reviewers' concerns. We have revised our manuscript thoroughly according to the comments from Dr. De Kauwe and reviewer #1.

We have added more evidences and experimental results to support our modeling approach and simulation results in section Discussion, described the details of modeling water dynamics in this model (BiomeE) and added the figures of simulated transpiration and soil water dynamics in supplementary information II, discussed possible effects of changes in temperature and water on model predictions. For addressing reviewer #1's concerns on "the fate of excessive carbon", we discussed three possible options and explained why we chose this one. We also discussed the uncertainty of our model in predicting short-term physiological responses and long-term ecological responses based on the fundamental assumptions of this model.

Please see the detailed responses to reviewers' comments and revised manuscript with tracking changes enclosed in this letter. Review comments are in italics and our responses are in normal. The line numbers of manuscript are referred to the tracking change version enclosed in this letter.

Your sincerely,

Ensheng Weng, Ray Dybzinski, Caroline E. Farrior, and Stephen W. Pacala

*Submitted on 09 Jul 2019*
*Referee #4: Martin De Kauwe, mdekauwe@gmail.com*

*In this study, Weng et al. explore the important question of how competition alters the responses of the vegetation to elevated CO2. They simulated forest responses to eCO2 along a N availability gradient using fixed and "competitively-optimal" allocation strategies. For such an important plant response (i.e. change in allocation) to global change, the detail afforded in most global models is troubling, so a study like this is very timely. Overall, I found this study interesting and one that I would like to see published in Biogeosciences. Nevertheless, I think this study will require some further revisions, particularly focusing issues of clarity (hopefully I've helped outline a few places).*

*My main suggestion would be to think a bit more about the discussion ...*

*- **What do the authors want the reader to take from this study?** For example, the authors open their discussion by saying: "Our model predicts increased root allocation at all nitrogen levels in response to elevated [CO2] in the competition runs." This is fine, but why not tell the reader why this happens mechanistically? What about your approach leads to this? Is it simply a consequence of what you assumed, or something more emergent? Also, what magnitude of change do you predict? And how does this vary with N availability? This seems more insightful than root allocation increased and this is broadly what you see in FACE experiments.*

We have rewritten this paragraph and re-organized the section of Discussion following Dr. De Kauwe's comments. The first two paragraphs (Lines 517~532) are copied below:

"Our simulations show that the responses of individual plants to elevated [$CO_2$] can be significantly changed by explicit inclusion of competition processes. Here, the major tradeoff for light- and N-limited trees is the relative allocation between stems and fine roots (Dybzinski et al. 2011). Although the wood allocation (and thus carbon sequestration potential) of every PFT used in this study increases under elevated [$CO_2$] at all nitrogen levels (e.g. Fig. 6e dashed lines), only those PFTs that allocate more to fine roots (with lower carbon sequestration potential) can survive competition under elevated [$CO_2$] (Fig. 6c solid lines). Put together, explicit inclusion of competition processes reduces the expected increase in biomass (and thus carbon sequestration potential) under elevated $[CO_2]$ compared with simulations that do not include competition processes (Fig. 7b).

Since there is a lack of direct observations or experiments to quantitatively validate the long-term patterns predicted by our model, we did not calibrate it to fit observations at Harvard Forest. In the following section, we analyze the model processes in detail and validate our modeling approach by comparing the general patterns from observations and experiments with model predictions. These comparisons also shed light on the modeling of allocation and vegetation responses to elevated $[CO_2]$."

*- Exploring this further, in the discussion about increased "fine-root overproliferation" being an emergent outcome of your simulations, could you talk a bit more about how this happens? As I understand it, you use a saturating N uptake function of root mass. In my experience, this does what it says on the tin, so there is only limited benefit in terms of increasing N with greater root investment. So, how does this differ in this study? One logical way would be if root allocation was very low to begin with, is this true here? I would suggest that the saturating root function is consistent with some of the FACE results, i.e. there is a benefit in increased N uptake, but this saturates. So, this leads me to ask how this leads to such a strong response in your experiments, over such a long time period ... This is interesting and worthy of discussion.*

The fine-root overproliferation is resulted from competition with other individuals, instead of the saturation uptake rate of roots. It is like an arm race: when your neighbor increases its investment in roots, you must follow. Otherwise, your neighbor will get more resources, grow faster, and overtop you. Even if fine roots are saturated at the stand level (i.e., adding new roots does not increase the total N uptake rate of all the trees in this site), individual plants still benefit from increased root investment, because this gives them more nitrogen than their neighbors. Yet, these competitive games lead to a tragedy of the commons in the form of lowered allocation to stems by the competitive-dominant strategies when they are resident.

We reorganized this paragraph to make explanations clearer (Lines: 663~677):

"The allocation strategy that maximizes site vegetation biomass allocates very little to fine roots (Figs. 3 and S1). In contrast, the competitively optimal strategy allocates more carbon to fine roots, termed "fine-root overproliferation" in the literature (Gersani et al., 2001; McNickle and

Dybzinski, 2013; O'Brien et al., 2005). It is the result of a competitive "arms race": while increasing fine root area under elevated [CO2] does not result in more nitrogen for an individual, failing to do so would cede some of that individual's nitrogen to its neighbors. Because most nitrogen uptake is via mass flow and diffusion (Oyewole et al., 2017) and because both of these mechanisms depend on sink strength, individuals with relatively greater fine root mass than their neighbors take a greater share of nitrogen, as was recently demonstrated empirically (Dybzinski et al., 2019; Kulmatiski et al., 2017). Thus, fine roots may overproliferate for competitive reasons relative to lower optimal fine root mass in the hypothetical absence of an evolutionary history of competition (Craine, 2006; McNickle and Dybzinski, 2013). This may also explain why root C:N ratio is highly variable (Dybzinski et al., 2015; Luo et al., 2006; Nie et al., 2013): a high density of fine roots in soil may be more important than the high absorption ability of a single root in competing for soil nitrogen in the usually low mineral nitrogen soils."

*- **It would be good to talk about competition for water and explore how both this and climate might change your model predictions.** I make this point below so I won't repeat it. Particularly when you make the link to the shift to competition for light (paragraph ln 725 onwards). **Those cited studies that your model result are consistent with, don't as I recall**, consider an explicit role for water either ...*

We have added a paragraph discussing water effects in section "**4.3 Model complexity and uncertainty**" (Lines: 768~783):

"Water is also a critical factor affecting allocation and its responses to elevated [$CO_2$]. Low soil moisture usually leads to high allocation to roots (Poorter et al., 2012). Elevated [$CO_2$] can reduce transpiration (as found in our study as well, Fig S7) and therefore increase soil moisture, resulting in increases in allocation to stems and aboveground biomass (Walker et al., 2019). A game-theoretic modeling study using the PPA framework shows that the competitively optimal allocation strategy shifts to high wood allocation at elevated [$CO_2$] in environments with water limitation (Farrior et al., 2015). This is opposite to the elevated [$CO_2$] effects on allocation in nitrogen-limited environments as simulated in this study. Fine root allocation is more responsive to nitrogen changes than it to soil moisture changes (Canham et al., 1996; Poorter et al., 2012). Poorter et al. (2012) attribute the mechanisms to the optimal strategies in response to the relative stable nitrogen supply and stochastic water input in soil. The vertical distribution of roots and the contributions of roots in different layers to water and nitrogen uptake also suggest that the uptake of soil nutrients are dominant in shaping root system architecture (Chapman et al., 2012; Morris et al., 2017), though root growth and turnover are flexible and sensitive to nitrogen and water supply (Deak and Malamy, 2005; Linkohr et al., 2002; Pregitzer et al., 1993)."

Since it would require additional simulations across two dimensions of environmental variables to determine how water limitation influences the effects of $CO_2$ in this model, we prefer leave this work for future. We have investigated the effects of water limitation in the absence of nitrogen limitation (Weng et al. 2015). In that study, the results show changes in $\varphi_{RL}$ ranging from 0.5~0.9.

As for the comment "***Those cited studies that your model result are consistent with, don't as I recall***", we reorganized the validation section and discussed the various responses in FACE experiments and the rationale of model development (Lines 603~621).
"The literature on experimental responses of plant community to elevated $[CO_2]$ shows that the responses vary with site characteristics, forest composition, stand age, plant physiological responses, and soil microbial feedbacks. For example, in Duke Free Air $CO_2$ Enhancement (FACE) experiment, where the major trees are loblolly pine (*Pinus taeda*), increases in root production at elevated $[CO_2]$ stimulated increased nitrogen supply that allowed the forest to sustain higher productivity (Drake et al., 2011). However, in Oak Ridge FACE, where the major trees are sweetgum (*Liquidambar styraciflua*), increased fine-root production under elevated [CO2] did not result in increased net nitrogen mineralization and increases in root production declined after eight years of $CO_2$ enhancement (Iversen, 2010; Norby and Zak, 2011). In EucFACE, where the major trees are *Eucalyptus tereticornis* and the soil is infertile, trees significantly increased their root exudation under limited nutrient supplies but had no significant increase in biomass in response to elevated [CO2] (Jiang et al., 2019a). The BangorFACE experiment (Smith et al., 2013) found that interspecific competition (*Alnus glutinosa, Betula pendula and Fagus sylvatica*) resulted in greater increases in root biomass at elevated $[CO_2]$. Leaf area index (LAI) responses to elevated $[CO_2]$ are also highly varied. As summarized by Norby and Zak (2011), low LAI (in this case, open canopy) sites showed significant increases in LAI and high LAI (in this case, closed canopy) sites showed low increases or even decreases in

LAI. They concluded that LAI in closed-canopy forests is not responsive to elevated [$CO_2$] (Norby et al., 2003; Norby and Zak, 2011)."

*- Line 791: This argument is completely true, but it also stands to reason that such approaches also need to be tested against data too! Just because something has the potential to predict more variable responses to climate, does not mean the predictions are more sound! This point is developed on line 848 by calling for an improvement in model validation/benchmarking. I don't follow this argument, to be honest. There is surely plenty of data available with which you could test core elements of the predictions of your model? For example, you could use the BAAD allometry databases, or similar, you need not just focus on CO2. Moreover, asserting that because your model predicts different responses over > 1000 years than those from short term experimental responses, and so, little can (may) be learned by tested against such data is ill thought through in my opinion. You are never going to have the types of data your model will need to "validate" it. The point of manipulation experiments, or comparisons across natural gradients (e.g. N availability, aridity, temperature), is to test core elements of (what should be emergent) model behaviour. In doing so, you are or trying to ensure that the underlying principles are sound. There are a number of studies that also have competition experiments (e.g. BIOCON, PHACE, etc) admittedly in grassland ecosystems, but there are data. It is, of course, true that simply assuming the response in a short term manipulation experiment is the "truth" would be fanciful, but these are one of our best ways to ground models in data. With this paragraph, why not think a bit more creatively about what kinds of existing datasets could be used to test elements of your model predictions? You will never have the data to replicate this experiment, so one either discusses the state of data, or one appreciates quite how much data we actually have and try to make use of it.*

We agree with Martin that we should have tested our model with data. Since this research was designed to explore the rules of allocation and roles of allocation schemes in the competition outputs of a vegetation dynamics model with full demographic processes, we didn't run experiments in FACE sites and didn't tune parameters according observations. We analyzed BAAD data and the data from Luyssaert et al. (2007), but found they could not be helpful in presenting the patterns along with nitrogen gradient. However, we analyzed our results with the meta-analyses of Poorter et al. (2012), Litton et al. (2003), and many FACE results (Duke, ORNL, EucFACE, BangorFACE, etc.)

We reorganized the discussion section with more results from field observations, data synthesis, FACE experiments throughout this section. Particularly, we added more evidences in sections "**4.1 Mechanisms of game-theoretic allocation modeling and simulation results validation**" and "**4.3 Model complexity and uncertainty**".

We copy the validation paragraph in section "4.1" below (Lines: 592~602. Please see more in these two sections):

"Generally, the predictions from competitively-optimal allocation strategies predicted by our model can be found in large scale forest censuses and site-level experiments, such as: 1) high nitrogen environments (i.e., productive environments) favor high wood allocation and low root allocation (Litton et al., 2003; Poorter et al., 2012); 2) elevated [CO2] increases root allocation (Drake et al., 2011; Iversen, 2010; Jackson et al., 2009; Nie et al., 2013; Smith et al., 2013); 3) low nitrogen availability limits vegetation biomass responses to elevated [CO2] as a result of high root allocation or root exudation (Jiang et al., 2019a; Norby and Zak, 2011); and 4) increases in vegetation biomass at elevated [CO2] are largely due to high wood allocation (Norby and Zak, 2011; Walker et al., 2019). These predictions emerged from the fundamental assumptions of our model without tuning parameters to fit the data, providing some confidence in the robustness of our approach."

We removed the "*completely true*" argument ("To make predictions of carbon cycle responses to the novel conditions of climate change, we must understand what determines the most competitive strategy, how the most competitive strategy changes with conditions, and how the most competitive strategy impacts the carbon cycle.") because our detailed discussion has shown these. We also removed the arguments related to "model benchmarking". It is too far from our results. We have discussed what this model can and cannot do, and explained why in sections "**4.1 Mechanisms of game-theoretic allocation modeling and simulation results validation**" and "**4.4 Common principles for allocation modeling and implications**".

*Specific things:*

*----------------*

*- In the methods, I do not really follow the simplification from LM3-PPA to BiomeE. After reading section 2.1, I'm really unclear what the key differences are, all that is presented for guidance is: "simplified the processes of energy transfer and soil water dynamics". This could mean a wide range of things! Does that mean otherwise the models are the same? So what is gained by this simplification? Does the model perform similarly? Could this be shown?*

As a model embedded in the land model of GFDL's ESMs, LM3-PPA has the modules that calculate the energy balances of raindrops, leaves/vegetation, and soil in detail, particularly for providing boundary conditions of land surface to atmosphere. For example, it requires a raindrop's temperature, energy content, mixture with leaves and soil. These calculations take lot computation time, making the model runs very slow. We simplified these processes with the soil water dynamics module used in TECO (Weng & Luo 2008).

We added a section "**F. Root Water Uptake and Soil Water Dynamic**s" in supplementary material I (model description). The root water uptake processes is the same as it in LM3-PPA (Eqs. F1~F9). The soil water dynamics and the energy budget associated with water fluxes are simplified using the algorithms in the TECO model. This model performs the same as LM3-PPA (the version used in Weng et al., 2017) does in plant growth and soil organic matter decomposition because the codes for these processes are almost identical.

We added three figures in supplementary information (II) to show water budget and soil water dynamics. Since this is study is not to explore the questions about water-carbon cycle interactions, we didn't particularly tune parameters to fit observations at Harvard Forest.

[Figure]

**Figure S7 Responses of transpiration to elevated [CO2] in monoculture and in polyculture runs.** The open symbols with dashed lines represent monoculture runs (panels a and b, only $\varphi_{RL}$=4 shown here.). The closed symbols with solid line represent polyculture runs (panels c and d). The relative changes of transpiration at eCO$_2$ are calculated as: 100x (Transp$_{eCO2}$-Trans$_{aCO2}$)/Transp$_{aCO2}$.

[Figure]

**Figure S8 Mean annual transpiration in monoculture and in polyculture runs at equilibrium state.** Panel a is for ambient $CO_2$ and panel b is elevated $CO_2$. The open symbols with dashed lines represent monoculture runs The closed symbols with solid line represent polyculture runs (blue-closed circles are for transpiration at $aCO_2$ and red-closed diamonds $eCO_2$).

[Figure]

**Figure S9 Water fluxes and soil water dynamics.** Two years' daily water dynamics at ambient and elevated [CO2], respectively, are shown in this figure, including daily precipitation (Precp), transpiration (Transp), soil surface evaporation (Evap), runoff, soil water content (vol./vol.) in layers 1 (0~0.05 m), 2 (0.05~0.5), and 3 (0.5~1.5 m).

*- Following on from this...the description of how water stress affects productivity is completely unclear to me, even after reading the text on lines 212-215. From digging through the supplementary, it seems like individuals could have different levels of water stress, but do they? Do you assume different slope terms "m" in your Leuning stomatal model? Do individuals have different rooting depths?*

We added the details of root water uptake and soil water dynamics modeling in SI – 1 (model description). We assume the roots follow the same vertical distribution (Eq. F9) and the slope "m" is also the same for individuals that differ in their sizes. However, the individuals with different sizes may still have different water stress when soil water is limiting because these individuals have different absolute roots in each soil layer and their water demand is dependent on the radiation they are getting (i.e., the canopy layers they are in). As shown in the figures in SI-2, lower soil layers have low water variation.

*- Following up on this point, where would an interested reader find the equations? Does Weng et al. (2017) contain all the equations? If so, can the authors more clearly indicate this at the top? My understanding is that the code is freely available, why not tell the reader of this in the methods? I know if I was reading this paper in my free time that would immediately make me more interested...*

The equations can be found in Weng et al. 2015 and Weng & Luo 2008. We have added a section (F) in the supplementary material I: model description. We also move the codes availability description to the Method section (Lines 170~172).

"BiomeE is derived from the version of LM3-PPA used in Weng et al. (2017), and its code is available at Github (https://github.com/wengensheng/BiomeESS)."

*- Eqn 3 ... could the authors provide rough ranges for the targets that emerge from these equations? I would have found this very helpful as I was reading the paper. I'm anticipating that the authors will respond by saying the range could be huge given the possible combinations, so consider this an optional request. I just wanted to get a sense of how much each target varied by and over what kinds of numbers.*

The range is huge, as Martin said, because tree size can vary from centimeters to tens of meters, but it follows the same allometry equations. The idea of targets of leaves and fine roots is to define a fully developed tree based on the structural relationships between tree diameter, height, and crown area, and the functional relationships between leaves, fine roots, and sapwood. In this model, the structural relationships define the tree sizes in height and spread (i.e., crown area), and the functional relationships define how many leaves and fine roots can potentially be attached to this tree.

*- where does the empirical constant representing the ratio of sapwood cross-sectional area to target leaf area come from? Is this based on measurements in any way? It presumably comes by given that leaf area and sapwood cross-sectional area are measured.*

It is described in Weng et al. 2015. This parameter is estimated from the observed ratios of cross sectional area of sapwood to leaves (i.e., Huber values, McDowell et al. 2002) times crown leaf area index. However, since we don't have a plant hydraulic model here, it is just used to separate wood into sapwood and heartwood for reasonably fitting these two pools that have been defined in LM3, and this parameter does not affect any plant physiological processes, though it is very important for plant hydraulics.

*- In instances where the plant doesn't have the resources to grow, if I follow the text, then C and N are returned to the storage pools for later. How large do these pools get? How much respiration takes place? In other models applied to eCO2 experiments (e.g. CABLE, CLM), the inability to grow in response to eCO2 led to a need to up-regulate respiration to make things balance (Zaehle et al. 2014, New Phyt). There is arguably very little experimental support for this kind of behaviour, in fact the data from the EucFACE experiment would show no support at all (paper in press).* **This could be a worthwhile thing to comment on in the discussion of the manuscript. Does the model assumptions lead to large builds up of these stores? If it doesn't, then can the models make a mechanistic link to explain how they achieve this seems more realistic behaviour compared to other models applied to eCO2 experiments...**

The case of carbon returning to NSC pool due to short of N is very rare, because leaf growth has been slowed down long before it happens and therefore reducing carbon supply.

Three cases: 1) N_supply >= N', C_supply is allocated according to the carbon-only scheme (full growth of leaves and fine roots as defined by their targets);

2) C_supply/CN_Wood <=N_supply< N', allocation follows the equations (7);

3) N_supply< C_supply/CN_Wood, part of C in C_supply return to NSC pool. However, the case (2) has reduced leaf growth and therefore C_supply (i.e., negative feedback by reducing leaves) before the condition "N_supply< C_supply/CN_Wood" is met.

We slightly revised this section to make it clearer (lines: 274~293).

"The parameter $r_{S/D}$ controls the allocation of $G_C$ and $G_N$ to the four plant pools (Eq. 7.1). It can be analytically solved (Eqs. 8 and 9).

$$r_{S/D} = Min\left[1, Max\left(0, \frac{G_N - G_C/CN_W}{N' - G_C/CN_W}\right)\right], \qquad \text{(Eq. 8)}$$

where, $N'$ is defined as the potential nitrogen demand for plant growth at $r_{S/D}=1$ (i.e., no nitrogen limitation).

$$N' \equiv \frac{\gamma\sigma\left[FR+Min\left(\frac{L^*+FR^*-L-FR,}{f_{LFR,max}\,G_C}\right)\right]-\varphi_{RL}L}{(\gamma\sigma+\varphi_{RL})CN_L} + \frac{\varphi_{RL}\left[L+Min\left(\frac{L^*+FR^*-L-FR,}{f_{LFR,max}\,G_C}\right)\right]-\gamma\sigma L}{(\gamma\sigma+\varphi_{RL})CN_{FR}} +$$

$$\frac{v\left[G_C-Min\left(\frac{L^*+FR^*-L-FR,}{f_{LFR,max}\,G_C}\right)\right]}{CN_F} + \frac{(1-v)\left[G_C-Min\left(\frac{L^*+FR^*-L-FR,}{f_{LFR,max}\,G_C}\right)\right]}{CN_W}.$$

(Eq. 9)

When $G_N \geq N'$ ($r_{S/D} = 1$), there is no nitrogen limitation, and all the $G_C$ will be used for plant growth and the allocation follows the rules of the carbon only model (Eqs 7.4~7.6 as $r_{S/D} = 1$). The excessive nitrogen ($G_N-N'$) will be returned to the NSN pool (as if they were never taken out). When $G_C/CN_{W,0}<G_N< N'$ (i.e., $0< r_{S/D} < 1$), all $G_C$ and $G_N$ will be used in new tissue growth; however, the leaves and fine roots cannot reach their targets at this step (i.e. they are down-regulated). When $G_N \leq G_C/CN_{W,0}$ ($r_{S/D} = 0$), all the $G_N$ will be allocated to sapwood and the excessive carbon ($G_C-G_N CN_{W,0}$) will be returned to NSC pool. This is a very rare case since a low $G_N$ leads to low leaf growth, reducing $G_C$ before the case $G_N<G_C/CN_{W,0}$ happens. Therefore, in most cases, Eq. 7.1 is: $G_C = G_W + G_L + G_{FR} + G_F$. **Overall, this strategy down-regulates leaf production under low nitrogen conditions while making use of assimilated carbon in height-structured competition for light**. "

This allocation scheme does not lead to high non-structural carbon (NSC) accumulation at low nitrogen environments (see the figure below and also in SI-II).

[Figure]

**Figure S10 Non-structural carbon storage in monoculture and in polyculture runs at equilibrium state.** Panel a is for ambient [$CO_2$] and panel b is elevated [$CO_2$]. The open symbols with dashed lines represent monoculture runs The closed symbols with solid line represent polyculture runs (blue-closed circles are for transpiration at $aCO_2$ and red-closed diamonds $eCO_2$).

As for respiration, we did not increase respiration at the short of nitrogen, but reduced leaf and root growth to reduce carbon supply and therefore nitrogen demand (i.e., negative feedback). The idea of this allocation scheme is to down-regulate photosynthesis by reducing LAI and make the most efficient use of available resources (particularly assimilated carbon). Increasing wood growth and reducing leaves and root growth is an optimal strategy at the limitation of nitrogen. This strategy would not compensate the decreases in plant growth induced by limiting nitrogen (i.e., no overgrowth of wood at limited nitrogen). It also does not waste the assimilated carbon by respiration (which is shown not true in many studies, as the study mentioned by Martin, the research of Jiang et al. 2019. We cited this paper according to the version in bioRxiv).

We added a paragraph in section "4.1 Mechanisms of game-theoretic allocation modeling and simulation results validation" discussing this strategy (Lines: 622~631):

"The nature of developing a model with generic assumptions and balanced processes reduces its capability to predict all of these responses. For example, plants have a variety of physiological mechanisms to deal with excessive carbon supply when plant demand (i.e., "sink") is relatively low (Fatichi et al., 2019; Körner, 2006), such as down-regulating leaf photosynthesis rate by the accumulated assimilates (Goldschmidt and Huber, 1992) or respiring excessive carbohydrates to regenerate substrates for photosynthesis (Atkin and Macherel, 2009). But these mechanisms are short-term physiological responses (minutes to hours, sometimes days) for plants in situations of temporary nitrogen shortage, high irradiation, or drought stress. It is not "economically" sustainable in an infertile environment to maintain highly productive leaves but to often suppress their photosynthesis or respire a large portion of their assimilated carbon."

*- I note that the other reviewers mentioned it and it is a theme I've noticed across a few of the papers from this set of authors...**there are datasets that are \*freely\* available to test the behaviour of this and other models from this group. I don't immediately see the what is stopping these authors testing their approaches on eCO2 data?** Is it because those studies don't have competition (not true of all FACE sites), but then please say so. The lead author was involved in a number of these studies and so would have access to all the data required. I realise they've added a further paragraph about the broad responses being consistent, but I find this a bit unsatisfactory to be honest. For years, modelling groups have been able to pass off general statements that their models were consistent with eCO2 experiments when they were explicitly tested, this clearly wasn't the case! Despite my reservations on this issue, this isn't a sticking point for me, the authors designed their experiment and it is not my place to tell them the paper I would have written (even if I might just have done that :P). It would be great in future work if the authors found a way to make use of the experimental data.*

Thanks for these suggestions. We do have access to the data of Oak Ridge and Duke FACE experiments. We really want to use these data to calibrate our model quantitatively at different sites and explore the key parameters (mechanisms) leading to different responses to elevated

[CO2] in these sits. I have joined a FACE-MIP proposal led by Dr. Walker for model inter-comparison with demographic vegetation models. Hopefully, we can do it together with Dr. De Kauwe in the near future. In the revised text (Lines: 645~660), we clearly described what this model cannot do and hopefully to be solved in the future with careful calibration of this model with data from those FACE sites.

"Since the purpose of this study is to explore long-term ecological strategies in different but relatively stable environments, we did not include these processes, especially since they present additional challenges in balancing the complexity of the tradeoffs between modeled demographic processes and plant traits. However, the lack of these processes does limit the predictions of instantaneous responses to variation in environmental conditions or resource supply and possibly of some long-term vegetation characteristics as well. For example, our model predicts reduced LAI under nitrogen limitation (Fig. S11) based on first principles, but it is incidentally the only mechanism that reduces the whole-canopy photosynthesis rate in our model. There are mechanisms that increase nitrogen use efficiency at the expense of carbon by increasing LMA and therefore leaf longevity to maintain high LAI and high canopy-level photosynthesis rates (Aerts, 1995, 1999; Aerts and Chapin, 1999; Givnish, 2002). We did not include these mechanisms in our simulations, although they are well-developed in this model (Weng et al. 2017), because we wished to focus on the strategy of allocation. The clear descriptions of our model's assumptions, its traceable processes, and inclusion of the tradeoffs involved in aboveground and belowground competition provide a useful benchmark from which to incorporate additional mechanisms and tradeoffs. "

*- With the competition angle (this could be me not quite following), you effectively have 8 PFTs competing? But you've only tested one fairly specific ecosystem (i.e. the meteorology found at Harvard forest). Presumably, your results would vary with climate? If I've followed, then I'm somewhat surprised this wasn't also a consideration? At the very least, can this be explored as a discussion point? Temperature and changing water availability (if properly parameterised, see earlier question), could conceivably change your conclusions...*

The overall pattern would be the same, though the quantity would change. We did a likely research in Weng et al. 2017 for the strategy of LMA at three sites (Oak Ridge, Harvard forest, and a Northern Old Black Spruce site in Manitoba, Canada) but not in this study. We added two paragraphs to discuss possible responses in different climates, combined with water effects (Lines: 748~783).

"We conducted simulations only at one site for the purpose of exploring the general patterns of competitively optimal allocation strategies and their responses to elevated [CO2] at different nitrogen availabilities. We can speculate about shifts in the competitively optimal allocation strategy in different forest biomes by considering the effects of temperature on soil nitrogen supply via the SOM's decomposition rate and its positive effect on net nitrogen mineralization. For example, the SOM decomposition rate is usually high in warm regions and low in cold regions (Davidson and Janssens, 2006) assuming there are no water limitations and SOM is equilibrated with carbon input. According to our model, allocation to roots is high in low nitrogen supply conditions (cold regions) and low in high nitrogen supply conditions (warm regions). This pattern can be found from temperate to boreal forest zones (Cairns et al., 1997; Gower et al., 2001; Reich et al., 2014; Zadworny et al., 2016). Temperature also alters NPP, i.e., carbon supply: as temperature goes down, NPP decreases and nitrogen demand decreases, alleviating nitrogen limitation and leading to shifts of allocation to stems. So, the differences in temperature effects on photosynthesis and SOM decomposition will determine competitive allocation strategy. Since SOM decomposition is more sensitive to temperature than gross primary production is at long-temporal and large-spatial scales (Beer et al., 2010; Carey et al., 2016; Crowther et al., 2016), our model suggests that allocation will shift to wood in a warming world. Whether the carbon stored in that wood is enough to offset the carbon released from increasing soil respiration is a critical question.

Water is also a critical factor affecting allocation and its responses to elevated [CO2]. Low soil moisture usually leads to high allocation to roots (Poorter et al., 2012). Elevated CO2 can reduce transpiration (as found in our study as well, Fig S7) and therefore increase soil moisture, resulting in increases in allocation to stems and aboveground biomass (Walker et al., 2019). A game-theoretic modeling study using the PPA framework shows that the competitively optimal allocation strategy shifts to high wood allocation at elevated [CO2]  in environments with water limitation (Farrior et al., 2015). This is opposite to the elevated [CO2] effects on allocation in nitrogen-limited environments as simulated in this study. Fine root allocation is more responsive to nitrogen changes than it to soil moisture changes (Canham et al., 1996; Poorter et al., 2012).

Poorter et al. (2012) attribute the mechanisms to the optimal strategies in response to the relative stable nitrogen supply and stochastic water input in soil. The vertical distribution of roots and the contributions of roots in different layers to water and nitrogen uptake also suggest that the uptake of soil nutrients are dominant in shaping root system architecture (Chapman et al., 2012; Morris et al., 2017), though root growth and turnover are flexible and sensitive to nitrogen and water supply (Deak and Malamy, 2005; Linkohr et al., 2002; Pregitzer et al., 1993)."

*- With Fig 3, would it be useful to make the allocation changes relative? It is a little hard to see the changes because of the span of different fractions on the c and d panels. Similarly, instead of showing one of either GPP or NPP, why not show the response ratio?*

We tried a couple of ways to show the relative changes and response ratio and found this is the best way to show the spreads of allocation along the gradient of nitrogen and root: leaf area ratio (RL). It is difficult to use a case (e.g., RL=6 and N = 114.5) as base to calculate relative changes and response ratio because they vary too much when RL changes from 6 to 1 and N from 114.5 to 552. The purpose of this figure is to show the predictions for monoculture runs are consistent with our predictions of the rules of allocation.

We changed the span of panels c and d in this figure, making it the same with it in the Fig. S1 (monoculture runs at elevate [CO]). The relative changes in response to elevated [CO2] can be found in Figures 5~7 (with the case of RL=4).

*- Is there a reason you don't show a figure more like Fig 3 for the polyculture simulations? You seem to jump straight to the changes for basal area, I was certainly expecting a similar plot first to orientate myself.*

For polyculture runs, there is only one PFT left at equilibrium state (this PFT outcompetes all others). So, the lines in each panel (variable) of Fig.3 become one for the polyculture runs. We used Fig. 4 to show how this happens through succession by showing the temporal changes in basal area of each PFT. For polyculture runs, all the panels (variables) in Fig. 1 are shown in Figs. 5, 6 and 7 with comparison with a monoculture case (RL=4) and responses to elevated [$CO_2$]. Fig. 7 c and d are specifically for comparison with all cases of the polyculture runs (Fig. 3f).

*- In fig 4g, I don't follow why the orange (RL=2) ends up being succeeded by (RL=3) after 600 years? This seems pretty abrupt and I don't see it commented on. Could the mechanism be explained further in the text? In every other panel, there seems to be a clear winner and then that is it. Similarly, across all panels, the transition between the dominance of one strategy and replacement by another looks quite abrupt. My expectation was that this would be more gradual than these plots are showing? Could the authors explain why I've got this concept so wrong! Or perhaps it is the compression of time on the x-axis that makes it seem like this visually?*

A successful invasion is slow because the invader (RL=3) only has a little advantage over the resident (RL=2) and it must start from a very low density of population. I zoomed in this figure at two levels (time and tree density) to show the details of changes in basal area and tree density (See the figure below).

[Figure]

The winning PFT must wait for an environment that will be generated by a dominant PFT that favors them. In our simulations, it took almost 200 years and the density of the winning PFT was very low after 200 years waiting (around 7 individuals/ha). Then, these individuals had to grow slowly in dark understory, though a little bit faster than the seedlings of current dominant PFT (RL=2). At the same time, the individuals of current dominant PFTs in understory kept replacing the trees in the top layer. After the winning PFT approached the top layer, they could generate seeds to increase its density finally. These new seedlings also need time to grow and approach the top layer with mortality of top layer trees. However, once there were enough individuals of the winning PFT in the top layer and plenty of new seedlings were generated, individuals of RL=2 will have no chance to get to the top layer and the replacing processes will be accelerated. We only showed 4 nitrogen levels in this figure (Fig. 4). There are some more such cases (slow invasion) shown in the supplementary material II Figs. S2 and S3, especially in the case of eCO$_2$ and N=427 g/m$^2$, where PFT4 takes around 1200 years to win.

We added a brief explanation of the succession processes in Lines 430~433.

"In some situations (e.g., Fig. 4g and Figs. S2 and S3), it takes a long time for the most competitive PFTs to out-compete the previously dominant PFTs because of the sequential replacement of dominant PFTs during the course of succession and the slow growth rate of trees in understory."

*Small things:*
* * *
*- This could be my ignorance of the difference (or lack of) but when the authors refer to "vegetation demographic models" do they simply mean dynamic vegetation model (DVMs) or DGVMs (with "global" thrown in)? If they do, would it make sense to maintain the far more common (ubiquitous?) usage? I'm usually not pedantic over such things, but to be honest I didn't see the need to redefine a very common catch-all term. It is up to the authors what they do with this point.*

We follow the community's term (e.g., Fisher et al. 2018). I agree with Martin that this is a type of dynamic vegetation model (DVM) and this term should be replaced by DGVM eventually and become a description of model mechanisms.

*- How many ESMs actually have VDMs in them? It would be good to cite a few if there are, I can't think of many off the top of my head! Aren't most run offline, rather than interactively with climate? The papers cited are certainly not examples of DGVMs embedded within ESMs.*
There are a couple of teams working on incorporating demographic processes into DGVMs (e.g., FATES of DOE, LM4 of NOAA GFDL, and Ent of NASA GISS). However, no one is successful in their coupled ESMs (as I know). We replaced "ESMs" with "the system" in this sentence to avoid confusion (Lines 84~89):
"With multiple cohorts and PFTs, VDMs can bring plant functional diversity and adaptive dynamics into the system when explicitly simulating individual-based competition for different resources and vegetation succession and thus predict dominant plant traits changes with environmental conditions and ecosystem development (Scheiter et al., 2013; Scheiter and Higgins, 2009; Weng et al., 2015)"
The citations of this sentence are all standalone vegetation models.

*- In the methods when the authors refer to "monoculture" runs as having allocation schemes as "analogous to the fixed allocation", I have a bit of trouble with this description. To me, this says fixed fraction, i.e. X, Y and Z% to difference plant pools. But, in which DGVM is that true? Some land surface models perhaps, but DGVMs? I feel like there is a lot of space for interpretation by the reader in with the authors mean here and the "see above" would send the reader back to ~line 60 from line 150. Why not be explicit in the methods exactly what is meant? I think the clarity will only help the readability of the paper.*

We removed this phrase because it cannot be explained well in one sentence and it is not the place to compare the single PFT's allocation strategy with other models'. Here, we actually meant it is a fixed strategy, instead of fixed allocation (or fixed fraction), as defined in De

Kauwe et al. 2014, based on allometry, functional relationships, and abiotic conditions. In this paper, there are two steps in determining allocation scheme: individual strategy and competitively optimal strategy as a result of competition. We have added a paragraph to discuss our modeling approach (Lines: 827~839).

"In competitively-optimal models, such as this study and also Valentine and Mäkelä (2012), the competition processes generate similar emergent patterns by selecting those that can survive in competition, regardless the details of those differences. The competition processes also make the details of allocation settings for a single PFT and their direct responses to elevated CO2 less important, because competition processes will select out the most competitive strategy from diverse strategies in response to changes in [CO2] and nitrogen. Our study and Valentine and Mäkelä (2012), posit a fundamental tradeoff between light competition and nitrogen competition via allocation based on insights gained from simpler models (e.g., Dybzinski et al., 2015; Mäkelä et al., 2008) for predicting allocation as an emergent property of competition. One advantage of building a model in this way is that the vegetation dynamics are predicted from first principles, rather than based on the correlations between vegetation properties and environmental conditions. With these first principles, the models can produce reasonable predictions, though the details of physiological and demographic processes vary among models."

*- The CN target of leaves seems pretty high? Where do these targets come from? Table 1 would be great with an additional column with "references". If the value isn't literature based then that column should be left empty.*

There are two measurements in GLOPNET data: 58.8 and 74.1. So, ours is a little bit higher than them (76.5). This value is from the eq. 2 of Weng et al. 2017 (leaf N per unit area = A+B*LMA, where parameters A and B are obtained by fitting the GLOPNET data. We didn't specifically tune these parameters in this study. We added a column for references in Table 1 following Martin's suggestion,

*- "This range covers the soil nitrogen content at Harvard Forest" - in space (across the forest?)? In time (i.e. over what time periods?)? Could the authors attempt to characterise what this range reflects in terms of N availability in the wider context of availability found globally? I suspect*

*this would be helpful for the general reader, I don't personally have an intuitive SOM value in my head and I would have found this helpful.*

We changed this sentence as (Lines 339~342):

"This range covers the soil nitrogen contents across the plots  at Harvard Forest with different species compositions and land use history (200~300 gN m$^{-2}$)  (Compton and Boone, 2000; Melillo et al., 2011), and represents the range from infertile to fertile soils in temperate forests (Post et al., 1985; Yang et al., 2011)."

*- Line 344: You said that the PFT was based on an evergreen needle-leaved tree, but you're modelling a deciduous ecosystem? Is this to avoid phenology issues, then why not pick a different ecosystem!?*

Harvard forest is a mixed forest with evergreen and deciduous trees. the reason we chose "Evergreen" is to simplify model processes by skipping phenology, which could complicate our simulations.

*- Line 547 - presumably you meant to replace "significantly" and forgot to, please check.*

Corrected. Thanks!

*Martin De Kauwe*

Thank you, Martin. Your comments are so helpful!

*Submitted on 18 Jul 2019*

*Anonymous Referee #1*

*This paper is a revision of a previous discussion paper looking at the interaction of competition and dynamic biomass allocation in response to N availability and elevated CO2 within a vegetation model. While the authors have expanded and clarified the methods, as well as clarifying their terms and extending the discussion, **I find that there is one fundamental issue raised by all three reviewers which has not been addressed satisfactorily, namely the allocation of carbon to sapwood under nitrogen limitation.** The brief text description has been replaced by a series of detailed equations (Eq. 7), but the conceptual problem remains. Under N limitation, so a low r_S/D, the C allocated to leaves and roots decreases (Eq. 7.4), while the C allocated to sapwood increases (Eq. 7.6). This is not merely a 'numerical step' as the authors claim in one of their replies but a fundamental assumption of the model. The implications of this assumption are that under N limitation, the tree size increases, with the C:N ratio of wood increasing and the leaf area moving away from its target value.*

This is actually a strategy of down-regulating LAI to reduce photosynthesis rate (carbon supply) at limited nitrogen supply. Woody tissue is an economic and efficient place to accept the extra carbon. We clarified it in the main text (Lines: 291~293).

"this strategy down-regulates leaf production under low nitrogen conditions while making use of assimilated carbon in height-structured competition for light."

The tree size does not increase at limited nitrogen (see the figure blow for simulated basal area, which can be an index of tree size).  Please also see detailed responses to the last question.

*The effects of this assumption can be seen clearly for the monoculture model version, with increased allocation to sapwood but decreased allocation to roots under elevated CO2 (Fig. 6), as well as a sharp decrease in tree height at high N availability (Fig. S6) The implied variation in tree height must also have some implications for the PPA and competition part of the model. This is in contradiction with observations, where, while a stoichiometric change is observed under nutrient limitation, this is limited in range and often accompanied by a reduction in growth e.g (Norby et al., 2010).*

Overall, the critical height (an index of PPA model that separates canopy and understory layers, defined as the height of the shortest tree in canopy layer) increases with N levels. Only in one case, it decreases because of the oscillation of size structure. This problem can be attenuated by tuning parameters to fit the actual growth and mortality rates (as we did in Weng et al. 2015 with a U-shaped mortality curve with tree size), but cannot be fully resolved within a patch.

The tree size is increasing generally as N increases. Please see the figure of biomass changes vs. ecosystem total nitrogen. Biomass increases with nitrogen at equilibrium state. The biomass per unit ground it proportional to tree diameter because each single tree's biomass is proportional to $D^{2.5}$ and crown area is proportional to $D^{1.5}$, so biomass per unit crown area is proportional to $D^{2.5}/D^{1.5} = D$ approximately.

Please see the basal area figure above (an index of the size of all trees, proportional to $D^{2.0}/D^{1.5} = D^{0.5}$ approximately) at equilibrium in different nitrogen levels below:

[Figure]

*While prioritising the growth of leaves and roots under resource limitation is supported by other studies, the allocation of excess C to wood is not. The question of the fate of excess C under nutrient limitation is of course still a very important problem (Fatichi et al., 2014), and proposed solutions cover increased storage, root exudates or ecosystem respiration but I cannot think of any studies that support unlimited wood growth.*

Our allocation scheme is a leaf (and root, because of the fixed R/L) priority strategy. We have made it clear in the revised manuscript.

(Lines: 302~304) "This allocation scheme prioritizes the allocation to leaves and fine roots, maintains a minimum growth rate of stems, and keeps the constant area ratio of fine roots to leaves. Based on these allocation rules, the average allocation of carbon and nitrogen to leaves, fine roots, and wood over a growing season are governed by the targets for the leaf area per unit crown area (i.e., crown leaf area index, $l^*$) and fine root area per unit leaf area ($\varphi_{RL}$)."

(Lines: 291~293) "Overall, this strategy down-regulates leaf production under low nitrogen conditions while making use of assimilated carbon in height-structured competition for light."

Following editor's suggestions, "*including one or two paragraphs in the discussion that thoroughly discusses the implications of the evolutionary principles behind the allocation assumption in PPA, and whether or not these assumptions adequately reflect the short-term vegetation responses to instantaneous environmental changes rather than gradual changes of vegetation dynamics under gradually changing environmental boundary conditions*", we have added three paragraphs to discuss the fate of excess C and explain why we choose this one based on the differences between physiological responses and long-term ecological strategies (Lines 622~660).

"The nature of developing a model with generic assumptions and balanced processes reduces its capability to predict all of these responses. For example, plants have a variety of physiological mechanisms to deal with excessive carbon supply when plant demand (i.e., "sink") is relatively low (Fatichi et al., 2019; Körner, 2006), such as down-regulating leaf photosynthesis rate by the accumulated assimilates (Goldschmidt and Huber, 1992) or respiring excessive carbohydrates to regenerate substrates for photosynthesis (Atkin and Macherel, 2009). But these mechanisms are short-term physiological responses (minutes to hours, sometimes days) for plants in situations of temporary nitrogen shortage, high irradiation, or drought stress. It is not "economically" sustainable in an infertile environment to maintain highly productive leaves but to often suppress their photosynthesis or respire a large portion of their assimilated carbon.

Root exudation is a critical process for plants. It can stimulate soil organic matter decomposition and nitrogen mineralization to facilitate soil nitrogen supply at the expense of carbon (Cheng, 2009; Cheng et al., 2014; Drake et al., 2011; Phillips et al., 2011). The process of root exudation has been adopted by many models to couple with microbial processes in the determination of soil organic matter decomposition (Sulman et al., 2014; Wieder et al., 2014, 2015). Some carbon-only models, e.g., LM3 (Shevliakova et al., 2009), the parent model of this one, and TECO (Luo et al., 2001), incorporate root exudation to put extra carbon into the soil in order to avoid down-regulating canopy photosynthesis or overestimating vegetation biomass, both of which had been tuned against data. However, in a demographic competition model like this one, when the microbial activities are not fully coupled and the nitrogen in soil is assumed fully accessible by roots of all individuals, individual plants cannot reap a reward from root exudation as they do in nature. Therefore, root exudation is not a competitive strategy in the system defined by the assumptions of this model.

Since the purpose of this study is to explore long-term ecological strategies in different but relatively stable environments, we did not include these processes, especially since they present additional challenges in balancing the complexity of the tradeoffs between modeled demographic processes and plant traits. However, the lack of these processes does limit the predictions of instantaneous responses to variation in environmental conditions or resource supply and possibly of some long-term vegetation characteristics as well. For example, our model predicts reduced LAI under nitrogen limitation (Fig. S11) based on first principles, but it is incidentally the only mechanism that reduces the whole-canopy photosynthesis rate in our model. There are mechanisms that increase nitrogen use efficiency at the expense of carbon by increasing LMA and therefore leaf longevity to maintain high LAI and high canopy-level photosynthesis rates (Aerts, 1995, 1999; Aerts and Chapin, 1999; Givnish, 2002). We did not include these mechanisms in our simulations, although they are well-developed in this model (Weng et al. 2017), because we wished to focus on the strategy of allocation. The clear descriptions of our model's assumptions, its traceable processes, and inclusion of the tradeoffs involved in aboveground and belowground competition provide a useful benchmark from which to incorporate additional mechanisms and tradeoffs."

*Given that all reviewers have asked the authors to justify and discuss their wood allocation assumption and I do not find that they have done so in a satisfactory manner, I cannot recommend this paper for publication.*

Hope our explanations and revisions in the main text addressed the concerns of you and the reader of this paper who raised the same concerns.

[revised manuscript text omitted]